# A quantitative theory of gamma synchronization in macaque V1

Eric Lowet[1][†][‡][*], Mark J Roberts[1], Alina Peter[2], Bart Gips[3], Peter De Weerd[1,4]

[1]Faculty of Psychology and Neuroscience, Maastricht University, Maastricht, Netherlands; [2]Ernst Strüngmann Institute (ESI) for Neuroscience in Cooperation with Max Planck Society, Frankfurt, Germany; [3]Donders Institute for Brain, Cognition and Behaviour, Radboud University Nijmegen, Nijmegen, Netherlands; [4]Maastricht Centre for Systems Biology, Maastricht University, Maastricht, Netherlands

**Abstract** Gamma-band synchronization coordinates brief periods of excitability in oscillating neuronal populations to optimize information transmission during sensation and cognition. Commonly, a stable, shared frequency over time is considered a condition for functional neural synchronization. Here, we demonstrate the opposite: instantaneous frequency modulations are critical to regulate phase relations and synchronization. In monkey visual area V1, nearby local populations driven by different visual stimulation showed different gamma frequencies. When similar enough, these frequencies continually attracted and repulsed each other, which enabled preferred phase relations to be maintained in periods of minimized frequency difference. Crucially, the precise dynamics of frequencies and phases across a wide range of stimulus conditions was predicted from a physics theory that describes how weakly coupled oscillators influence each other's phase relations. Hence, the fundamental mathematical principle of synchronization through instantaneous frequency modulations applies to gamma in V1 and is likely generalizable to other brain regions and rhythms.
DOI: https://doi.org/10.7554/eLife.26642.001

*For correspondence:
elowet@mailfence.com

Present address: [†]McGovern Institute, Massachusetts Institute of Technology, Cambridge, United States; [‡]Boston University, Boston, United States

Competing interests: The authors declare that no competing interests exist.

## Introduction

Synchronization, the ability of oscillators to mutually adapt their rhythms (*Pikovsky et al., 2002*; *Winfree, 1967*), is a ubiquitous natural phenomenon. Neural synchronization in the gamma-range has been reported both in subcortical structures (*Akam et al., 2012*; *Steriade et al., 1993*; *Zhou et al., 2016*) and in cortical areas (*Fries, 2015*; *Gray and Singer, 1989*; *Gregoriou et al., 2009*). Gamma rhythms emerge in activated neural circuits in which fast-spiking inhibitory neurons play a central role (*Cardin et al., 2009*; *Tiesinga and Sejnowski, 2009*; *Traub et al., 1996*). A prime example is the emergence of gamma rhythms in the early visual cortex during visual stimulus processing (e.g. *Brunet et al., 2015*; *Gail et al., 2000*; *Gray and Singer, 1989*; *Hermes et al., 2015*; *Ray and Maunsell, 2010*; *Roberts et al., 2013*). Gamma synchronization has been related to the formation of neural assemblies within (*Gail et al., 2000*; *Gray and Singer, 1989*; *Havenith et al., 2011*; *Vinck et al., 2010*) and across brain areas (*Bosman et al., 2012*; *Gregoriou et al., 2009*; *Grothe et al., 2012*; *Jia et al., 2013a*; *Roberts et al., 2013*; *Sirota et al., 2008*; *Zhou et al., 2016*).

The precise temporal coordination of presynaptic spikes increases their effectiveness on postsynaptic targets (*Fries et al., 2001*; *Tiesinga et al., 2004*) and can thereby modulate the effectiveness of neural communication (*Börgers et al., 2005*; *Cannon et al., 2014*; *Womelsdorf et al., 2007*), as shown between V1 and V4 during visual attention (*Bosman et al., 2012*; *Grothe et al., 2012*). Temporal coordination in terms of spike timing (phase code) might be an efficient and robust mechanism for information coding (*Havenith et al., 2011*; *Jensen et al., 2014*; *Maris et al., 2016*; *Tiesinga and*

*Sejnowski, 2009*; *Vinck et al., 2010*). Further, gamma rhythmic inhibition might increase coding efficiency through sparsening (*Chalk et al., 2015*; *Jadi and Sejnowski, 2014*; *Vinck and Bosman, 2016*) and normalization (*Gieselmann and Thiele, 2008*; *Ray et al., 2013*) of neural activity. These network consequences of gamma have led to influential hypotheses about the function of gamma in sensation and cognition (*Buehlmann and Deco, 2010*; *Buzsáki and Wang, 2012*; *Eckhorn et al., 2001*; *Fries, 2015*; *Gray and Singer, 1989*; *Maris et al., 2016*; *Miller and Buschman, 2013*), including a role in perceptual grouping (*Eckhorn et al., 2001*; *Engel et al., 1999*; *Gray and Singer, 1989*) and in visual attention (*Bosman et al., 2012*; *Fries, 2015*; *Gregoriou et al., 2009*; *Miller and Buschman, 2013*).

Surprisingly, in spite of important scientific advances, it is not well understood how gamma rhythms synchronize and what the underlying principles of synchronization are. For example, recent experimental observations of large variability in gamma oscillation frequency have raised doubts about the robustness and functionality of gamma synchronization in the brain. It has been observed that frequency fluctuates strongly over time (*Atallah and Scanziani, 2009*; *Burns et al., 2010*; *2011*) and that different cortical locations can express different preferred frequencies at a single moment in time (*Bosman et al., 2012*; *Ray and Maunsell, 2010*). That these observations have led to doubts on the functionality of gamma synchronization reveals a stationary view of synchronization, which assumes that the underlying oscillatory dynamics are stable at a fixed phase-relation and shared frequency. This is also reflected in the widespread use of stationary methods to assess gamma synchronization, of which spectral coherence is a prime example (*Carter et al., 1973*). From a dynamic systems perspective, however, synchronization is primarily a non-stationary process (*Izhikevich, 2007*; *Izhikevich and Kuramoto, 2006*; *Kopell and Ermentrout, 2002*; *Pikovsky et al., 2002*; *Winfree, 1967*), because oscillators mutually adjust their rhythms through phase shifts (i.e. through changes in the instantaneous frequency).

Here, by using a combination of theoretical and experimental techniques, we studied the dynamical principles of gamma synchronization in monkey visual area V1. We simultaneously recorded gamma-rhythmic neural activity at different V1 cortical locations and studied their synchronization properties while using local stimulus contrast (*Ray and Maunsell, 2010*; *Roberts et al., 2013*) to modulate the frequency difference averaged over time (detuning). Strikingly, even when the mean frequencies did not match (detuning > 0), we often observed that gamma rhythms synchronized. This was achieved by continuously varying their instantaneous frequency difference, which permitted the temporary maintenance of a preferred phase relationship during reoccurring periods of minimized instantaneous frequency difference. The interplay between the detuning and the amount of instantaneous frequency modulations regulated the phase-locking strength and the preferred phase-relation between V1 locations. Furthermore, to achieve a principled understanding of our observations, we applied the theoretical framework of weakly coupled oscillators to our data (*Ermentrout and Kleinfeld, 2001*; *Hoppensteadt and Izhikevich, 1998*; *Kopell and Ermentrout, 2002*; *Kuramoto, 1991*; *Pikovsky et al., 2002*). We found that a single differential equation accounted well for the non-stationary frequency modulations and further allowed for precise predictions of how the phase-locking and the phase-relation between gamma rhythms changed across conditions.

## Results

### Frequency differences regulate the phase synchronization process between local monkey V1 gamma rhythms

We first asked how synchronization within V1 was influenced by mean frequency differences, and by the distance between recording sites. To this aim, we recorded simultaneously from two to three laminar probes (each with 16 recording contacts spaced along the recording shaft, see Supplementary Materials for alignment procedure) in cortical area V1 of two macaques (M1 and M2) (*Figure 1A*). We used distances between probes of 1–6 mm, matching approximately the extent of V1 horizontal connectivity (*Stettler et al., 2002*). Notably, horizontal connectivity strength declines strongly with distance between cortical locations (*Stettler et al., 2002*), so that increasing interprobe distance indexes decreasing horizontal connectivity strength. The monkeys fixated centrally while a full-screen static square-wave grating with spatially varying contrast was shown (*Figure 1B*).

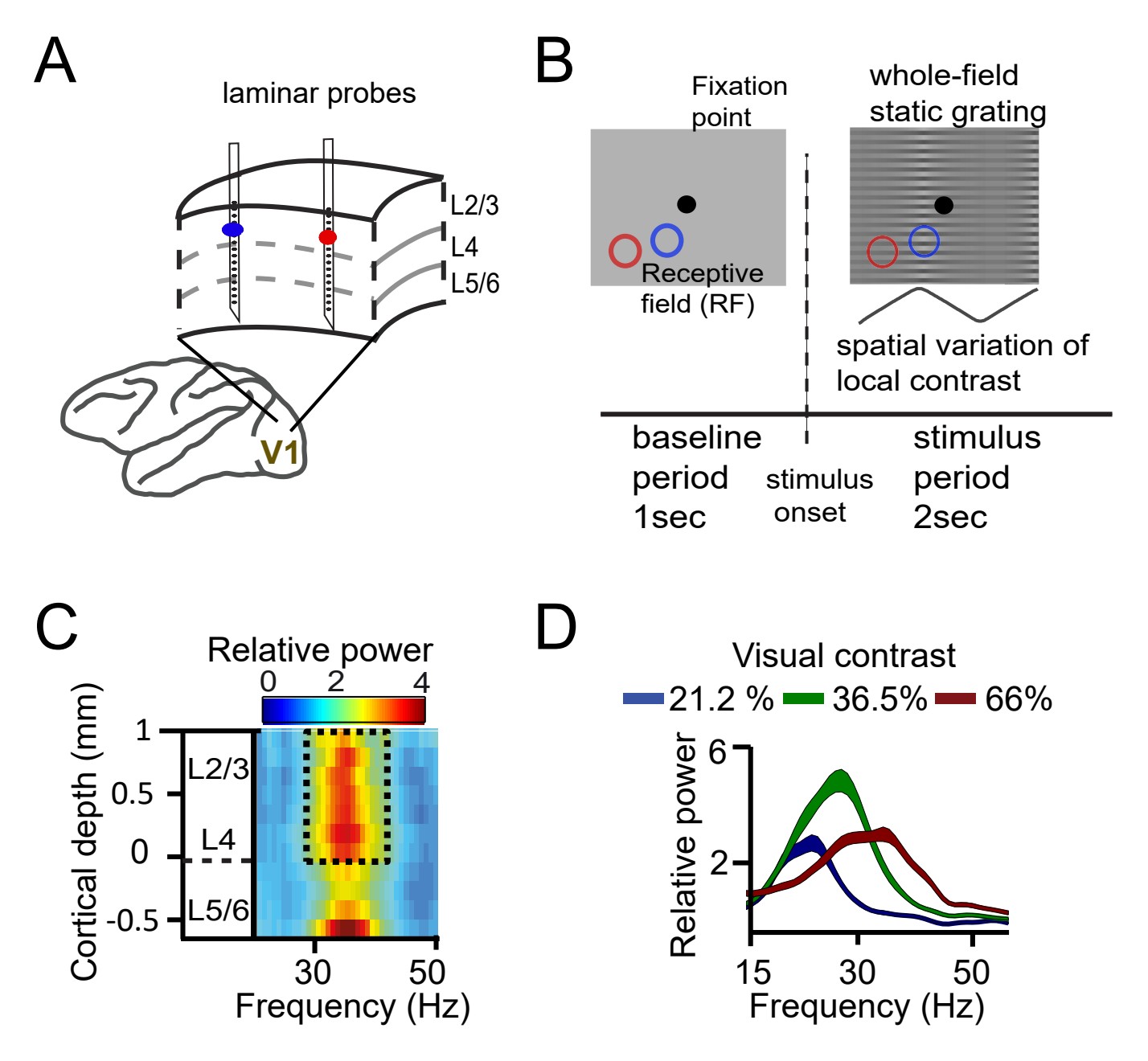

**Figure 1.** Experimental setup and contrast-dependent V1 gamma frequencies. (**A**) Schematic rendering of recording location. Two to three laminar probes were inserted with 1–6 mm separation in cortical area V1. (**B**) The visual paradigm consisted of a 1 s baseline period with a gray background and 2 s visual stimulation with a full-screen static grating characterized by spatially varying local contrast. During both periods the monkeys maintained their gaze on a fixation point (controlled by eye tracking). For analysis, the stimulation period (0.2–2 s) was used, not including the first 200 ms to avoid stimulus-evoked transients. Two receptive fields (RF) from different probes are shown on the grating stimulus (blue and red circles). The aim was to modulate (detune) the local frequencies of gamma rhythms using local contrast differences. (**C**) Spectral power relative to baseline as a function of V1 cortical depth (36.5% contrast, population average, M1). Data for gamma analysis are taken from granular and superficial layers (dashed box) unless stated otherwise. (**D**) Local contrast modulated gamma frequency (population average, M1) as shown in the power spectral profile for three of the five contrast values employed. Width of shaded area represents SE.

DOI: https://doi.org/10.7554/eLife.26642.002

The following figure supplements are available for figure 1:

**Figure supplement 1.** Cortical depth alignment and analysis.

DOI: https://doi.org/10.7554/eLife.26642.003

*Figure 1 continued*

**Figure supplement 2.** Effect of contrast and eccentricity on macaque V1 gamma frequency.

DOI: https://doi.org/10.7554/eLife.26642.004

The local contrast varied periodically over visual space such that different contrasts were presented to different cortical locations. The magnitude of contrast difference (ranging from 0% to ~43%, see Table S1) was manipulated by varying the sign and amplitude of the spatial variation in contrast. The stimulus gratings induced gamma power in layers 2–4 and in the deepest layer (*Figure 1C*, *Figure 1—figure supplement 1*; *van Kerkoerle et al., 2014*; *Xing et al., 2012*). The gamma frequency increased systematically with higher local contrast (linear regression, single contact level, M1: $R^2 = 0.38$, M2: $R^2 = 0.27$, both $p < 10^{-10}$, *Figure 1D*, *Figure 1—figure supplement 2*). The range of the frequency shift in our data (~5 Hz) was smaller than in *Roberts et al., 2013*, reflecting a narrower contrast range used here; from ~20% to ~60% (*Table 1*). The tight relationship between contrast and gamma frequency allowed us to induce different mean frequencies in nearby cortical locations separated by as little as 1–6 mm (e.g., *Figure 2A*).

The close positions of recording sites may have led to a contribution of volume condition to synchronization measures. The LFP, despite being local in comparison to extracranial electrical field measure like EEG, still might integrate signals over a scale of up to 1 cm horizontally (*Kajikawa and Schroeder, 2011*; *Lindén et al., 2011*; *Xing et al., 2009*), which may affect the interpretability of layer-dependent analysis (*Kajikawa and Schroeder, 2015*). Using laminar probes enabled us to reduce the influence of volume conduction by calculating current-source density (CSD), as the second spatial derivative of LFP signals measured along each probe (*Mitzdorf and Singer, 1977*; *Schroeder et al., 1991a*; *Vaknin et al., 1988*). The success in reducing volume conduction using CSD favors its use over LFP for spectral analysis at high spatial resolution. Next, we used a singular spectrum decomposition technique (SSD, (*Bonizzi et al., 2014*)) to extract gamma components from the CSD. From these single-trial gamma signals, we estimated the instantaneous frequency and phase at individual recording sites and the instantaneous phase difference between sites. In the example shown of a single pair of recording sites (*Figure 2A*), the stimulus induced a gamma frequency of 36 Hz at one probe and 32 Hz at the other, because different contrasts appeared in the respective receptive fields. As shown in the raw trace of the instantaneous phase differences in *Figure 2B*, the gamma phase difference was not constant over time, but continuously exhibited modulations and shifts. Sometimes, the phase difference changed slowly and at other times, it changed faster. The change of phase difference over time is called 'phase precession' (*Pikovsky et al., 2002*). Note that this should be distinguished from a phenomenon of the same name: the precession of preferred spiking phase in the theta cycle observed in rats moving through hippocampal place fields (*Skaggs et al., 1996*). In the present study, the 'rate' or 'speed' of precession is expressed as the instantaneous frequency difference in Hz (*Figure 2C–D*). We found that the observed modulations in phase difference were not random as would be expected if different frequencies precluded synchronization. Instead, the instantaneous phase difference was related to the instantaneous frequency difference. In *Figure 2E*, we plotted the instantaneous frequency difference

**Table 1.** Range of contrast difference conditions used for the experimental task for monkeys M1 and M2.
The top sub-table shows the contrast difference conditions (in %) used for M1, and the bottom sub-table shows the values for M2.

| | Contrast difference condition (monkey M1) | | | | | | | | |
|---|---|---|---|---|---|---|---|---|---|
| **Range** | **44.7** | **35.9** | **24.8** | **13.3** | **0** | **−13.3** | **−24.8** | **−35.9** | **−44.7** |
| RF 1 | 66 | 58.6 | 51.7 | 44.3 | 36.5 | 31 | 27 | 22.7 | 21.2 |
| RF 2 | 21.2 | 22.7 | 27 | 31 | 36.5 | 44.3 | 51.7 | 58.6 | 66 |
| | Contrast difference condition (monkey M2) | | | | | | | | |
| Range | 42.7 | 34 | 24.5 | 13.6 | 0 | −13.6 | −24.5 | −34 | −42.7 |
| RF 1 | 62.7 | 57.5 | 52.2 | 46 | 39.2 | 32.5 | 27.7 | 23.5 | 20 |
| RF 2 | 20 | 23.5 | 27.7 | 32.5 | 39.2 | 46 | 52.2 | 57.5 | 62.7 |

DOI: https://doi.org/10.7554/eLife.26642.005

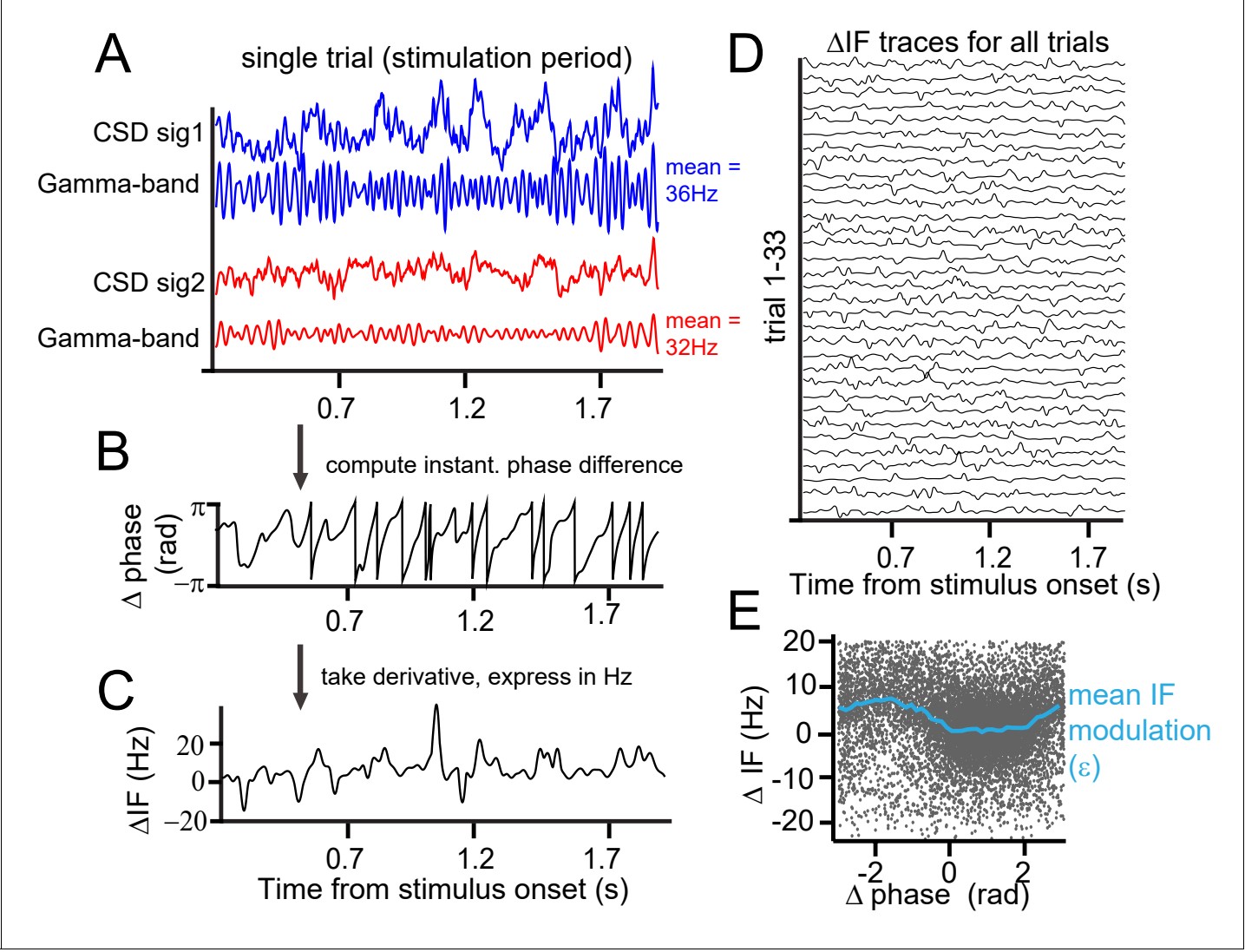

**Figure 2.** Instantaneous frequency modulation. (**A**) Example CSD (blue and red) traces recorded during visual stimulation from which gamma-band components were extracted using singular-spectrum decomposition. (**B**) We computed the phase difference (black) between signals 1 and 2 by computing the circular difference of their instantaneous phases. The instantaneous phase was derived by applying the Hilbert transform on the gamma-band components. (**C**) Taking the derivative, and scaling the result as the instantaneous frequency difference (ΔIF) gives the rate of phase precession. Notice the modulations over time. Note further that ΔIF variations are the result of IF variations occurring simultaneously at the two contact points that together constitute a contact pair. (**D**) Shows the ΔIF traces for all trials in a single session and stimulus condition for a single contact pair. (**E**) The ΔIF points (N = trial number*samples = 33*1800 = 59400) are plotted as a function of phase difference. A clear modulation of ΔIF values (blue line represents the mean) with phase difference can be observed showing that ΔIF modulations are not random. This means that phase precession depends on the momentary phase-difference (phase-relation) between contrasts. It is worth noting that the ΔIF values tend to be positive, which is related to the sign of the contrast difference and resulting detuning. If for the same pair the contrast difference had been reversed, ΔIF values would have tended to be negative.

DOI: https://doi.org/10.7554/eLife.26642.006

as a function of the instantaneous phase difference. Each black point represents one momentary observation and the blue line the average using binning of 0.25 rad width. The plot shows that the instantaneous frequency difference (ΔIF) tends to be lower at certain phase differences than at others. We observed that the average frequency difference was close to 0 for phases differences between 0 and 2 radians, but was much higher at other phase relationships. The key to understanding how this dynamic relationship leads to synchronization is that phase relationships associated with lower frequency differences are maintained longer over time (slower precession) than phase

relationships associated with higher frequency differences. This can be readily appreciated by the higher density of dots in *Figure 2E* between 0 and 2 radians.

Three key examples from our results nicely illustrate the dynamics of the relationship between instantaneous frequency difference and phase difference (*Figure 3*). These examples were derived from our experimental design, in which we varied the cortical distance between probes in a pair (varying horizontal connectivity strength), and in which we systematically varied for each pair the contrast difference (9 levels), and hence the mean gamma frequency difference. We show in these examples positive frequency differences for illustration, but negative differences were also present for single contact pairs in our data, depending on the sign of the contrast difference (see *Figure 3—figure supplement 1*). In the first example (*Figure 3* column 1), we show two cortical locations separated by a relatively large distance of ~5 mm, presented with a visual contrast difference of 17% (*Figure 3A*). This yielded an overall mean frequency difference of 5 Hz (*Figure 3B*). If this frequency difference were constant, the phase difference would advance at a phase precession rate of $2\pi$ every 200 ms, which would preclude synchronization. However, the frequency difference was not constant. Instead, the instantaneous frequency difference changed as a function of phase difference (*Figure 3B*, *Figure 3—figure supplement 1*) with a modulation amplitude of ~1 Hz (approximately (max-min)/2; see Appendix). At the smallest frequency difference (4 Hz, yellow point), the phase precession was slowest ($2\pi$ every 250 ms). As a result, the probability distribution of phase differences over time (*Figure 3C*) was non-uniform giving a phase-locking value (*Lachaux et al., 1999*) (PLV) of 0.11. The peak of the distribution, the 'preferred phase', was at 1.3 rad, in line with the minimum of the instantaneous frequency modulation shape. In the second example, we chose a pair with a similar frequency difference of 4.8 Hz but a reduced distance (~2.5 mm, *Figure 3D*). The instantaneous frequency modulation was larger with a modulation amplitude of 1.8 Hz (*Figure 3E*) and a minimum around 3 Hz at the preferred phase. Because a lower minimum frequency difference corresponds to slower phase precession at the preferred phase than in the previous example, the preferred phase was maintained for longer. This resulted in a narrower phase difference distribution, indicating higher synchrony (PLV = 0.32, *Figure 3F*). The peak of the distribution was centered at a smaller phase difference (0.78 rad). In the third example, the cortical distance remained the same as in *Figure 3D* but the frequency difference was reduced (2.8 Hz) by eliminating the contrast difference (*Figure 3G* and *Figure 1—figure supplement 2*). Compared to example 2, the magnitude of the instantaneous frequency modulation did not change (modulation amplitude 1.8 Hz, *Figure 3H*), but showed a lower mean difference and a minimum close to zero (1 Hz, *Figure 3H*). Thus, the associated phase difference (0.48 rad) could be maintained for even longer periods and the phase difference probability distribution became even more pronounced and narrower (PLV = 0.51, *Figure 3I*). The three examples illustrate how the mean frequency difference and cortical distance (a proxy of the strength of horizontal interactions) determine the dynamic relationship between the instantaneous frequency difference and the phase difference during synchronization. In the following sections, we will show how these observations were characteristic of the whole dataset comprising 805 recorded across-probe contact pairs in monkey M1 and 882 pairs in monkey M2.

Before framing the relationship between instantaneous frequency modulations and the phase difference distribution in a mathematical manner (next section), we illustrate that relationship by an analogy with two cyclists on a circular circuit. Their speed is calculated as the number of circuits they complete in a given time, hence speed is analogous to frequency. Phase is equivalent to position on the track and phase difference is equivalent to the distance between the cyclists. Phase locking therefore is analogous to the amount of time they spend at a consistent distance from each other. The phase precession rate is analogous to the speed with which the distance between the cyclists changes. If the cyclists maintain constant speeds as they go around the track, the distance between them will vary at a constant rate, and they will only maintain a consistent distance (phase difference) if they both cycle at the same speed. Hence with stable instantaneous speed (i.e. frequency), phase locking is either absent or complete. If the cyclists do vary their speed, more complex patterns become possible. If they vary their speeds independently, the phase-difference distribution will be flat. However, suppose the slower cyclist can travel faster in the slipstream of the faster cyclist, then the amount of time the cyclists travel close to each other will be greater than the time they spend far away. They might cycle around the whole circuit this way (complete phase locking), but more likely the faster cyclist will get away and the cyclists will travel at their natural speed until they come together again allowing the slower cyclist to speed up. Hence, all phase differences are represented,

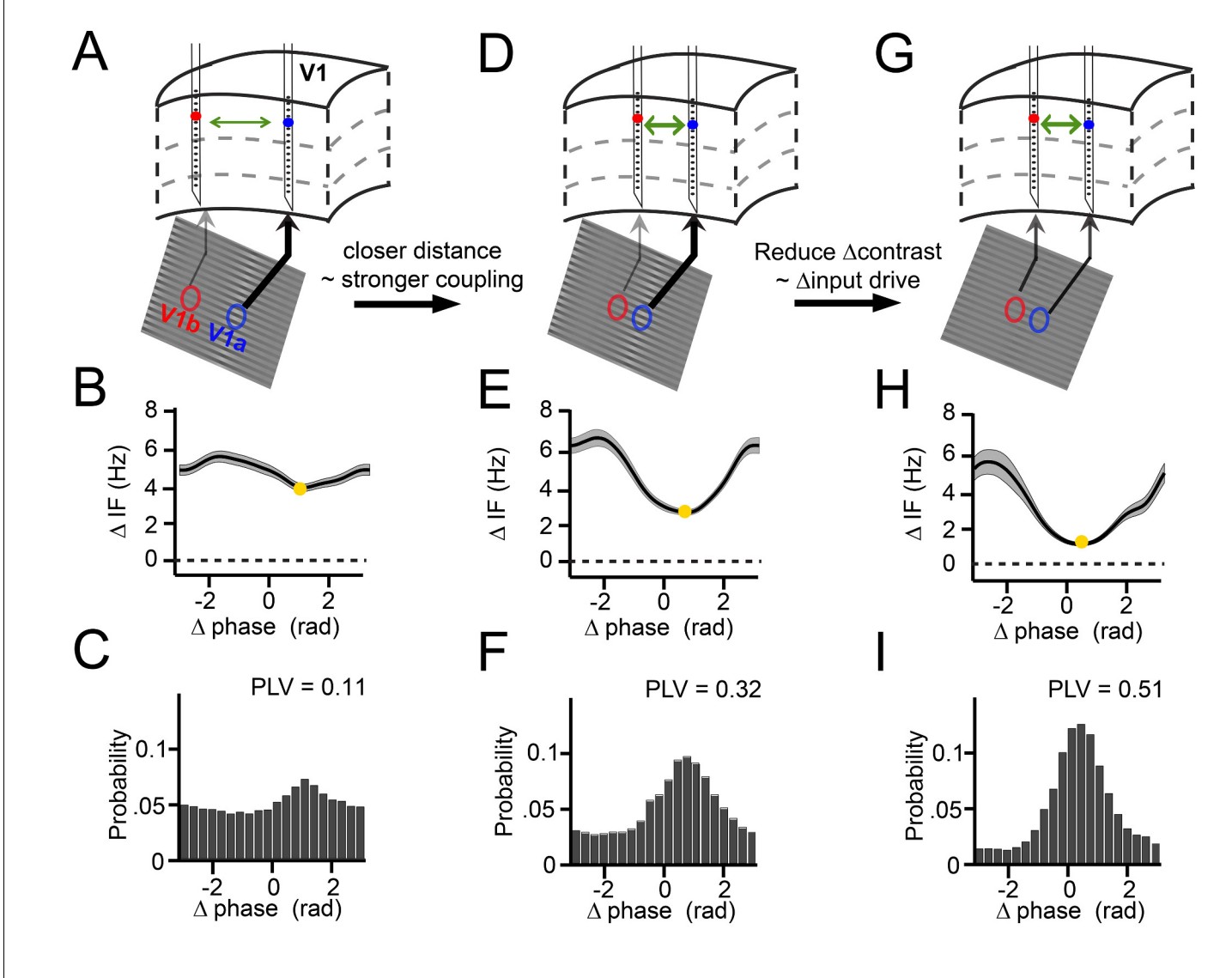

**Figure 3.** Illustration of V1 gamma-band dynamics. (A–C) Example 1 showing synchronization despite frequency difference (data from Monkey M1,~30 trials per condition). (A) Schematic figure of the contacts used from two laminar probes in V1. Below is a section of the stimulus grating with the corresponding RFs. The arrows' thickness indicates the strength of contrast-dependent input to the corresponding V1 location. (B) Instantaneous frequency difference (ΔIF), equivalent to the phase precession rate, as a function of phase difference. Yellow dot indicates the modulation minimum, equivalent to the preferred phase difference, shading is ±SE (C) The phase difference probability distribution and phase-locking value (PLV). (D–F) Example 2; probes were closer and the gamma peak frequency difference was similar. Conventions as in A-C. (G–I) Example 3; same distance, reduced frequency difference. Compare B, E, H; the RF distance determined IF modulation amplitude, whereas contrast difference determined mean gamma frequency difference. Note that the instantaneous phase difference at which the instantaneous frequency difference is minimal (yellow dot) is smaller for greater amplitudes of instantaneous frequency difference variation (compare E, H to B).

DOI: https://doi.org/10.7554/eLife.26642.007

The following figure supplements are available for figure 3:

**Figure supplement 1.** Instantaneous frequency modulations during gamma synchronization.

DOI: https://doi.org/10.7554/eLife.26642.008

**Figure supplement 2.** PING network simulations and intermittent synchronization.

DOI: https://doi.org/10.7554/eLife.26642.009

but some are over-represented – specifically phases where the faster cyclist is just in the lead. In our experiment, we measured the instantaneous gamma frequency, similar to looking on the speedometer of each bike, and the instantaneous phase difference, corresponding to the distance between the cyclists. This allowed us to understand the resultant probability distribution of phase differences (yielding the phase locking value and the average phase difference). We found that the average speed difference and the speed modulation strength defined the probability distribution.

## The theory of weakly coupled oscillators (TWCO): A framework for cortical gamma synchronization

We now show how the observed synchronization behavior can be accounted for within the mathematical framework of the theory of weakly coupled oscillators (*Ermentrout and Kleinfeld, 2001*; *Hoppensteadt and Izhikevich, 1998*; *Kopell and Ermentrout, 2002*; *Kuramoto, 1991*; *Pikovsky et al., 2002*; *Winfree, 1967*). Many oscillatory phenomena in the natural world represent dynamic systems with a limit-cycle attractor (*Winfree, 2001*). Although the underlying system might be complex (e.g. a neuron or neural population), the dynamics of the system can be reduced to a phase-variable if the interaction among oscillators is weak. If interaction strength is weak, amplitude changes are relatively small and play a minor role in the oscillatory dynamics. In this way, V1 neural populations can be approximated as oscillators, 'weakly coupled' by horizontal connections (*Figure 4A*). The manner in which mutually coupled oscillators adjust their phases, by phase-delay and phase-advancement, is described by the phase response curve, the PRC (*Brown et al., 2004*; *Canavier, 2015*; *Izhikevich, 2007*; *Kopell and Ermentrout, 2002*; *Schwemmer and Lewis, 2012*). The PRC is important, because if the PRC of a system can be described, the synchronization behavior can be understood at a more general level and hence predicted across various conditions.

According to the theory, the synchronization of two coupled oscillators can be predicted from the forces they exert on each other as a function of their instantaneous phase difference. The amount of force is here defined as interaction strength, which is modulated as a function of phase difference by an interaction function that is closely related to the PRC (for a detailed discussion of the relationship between the two functions, please see *TWCO predicts synchronization properties of V1 cortical gamma rhythms*). In addition, each oscillator has an intrinsic (natural) frequency and its own source of phase noise, making the oscillators stochastic. Hence, the phase precession of two oscillators is given by:

$$\dot{\theta} = \Delta\omega + \varepsilon G(\theta) + \eta \tag{1}$$

where $\dot{\theta}$ is the time derivative of the phase difference $\theta$ (the rate of phase precession), $\Delta\omega$ the detuning (the intrinsic frequency difference), $\varepsilon$ the interaction strength (scalar function), $G(\theta)$ the interaction function (mutual PRC), and $\eta$ the combined phase noise, where $\eta \sim N(0, \sqrt{2}\sigma^2)$, see (*Figure 4B*). Phase noise is defined here as variation that is unrelated to interaction, which occurs for neural oscillators due to inherent instabilities of the generative mechanism (*Atallah and Scanziani, 2009*; *Burns et al., 2010*). This type of variation is distinct from measurement noise, which is unrelated to the dynamics of the system. We express $\omega$, $\varepsilon$ and $\eta$ in units of Hz (1Hz = 2π*rad/s). The time derivative $\dot{\theta}$ is also expressed in Hz (instantaneous frequency, IF).

Note that here, detuning $\Delta\omega$ is the intrinsic or natural frequency difference between two oscillators, which is the frequency difference oscillators would have without any interaction. The *measured* detuning can differ from the intrinsic detuning $\Delta\omega$ if the oscillators exhibit synchronization. In model simulations or while solving analytical equations, intrinsic frequencies and frequency differences are known, whereas in empirical data the intrinsic detuning $\Delta\omega$ needs to be estimated from the measured detuning. Likewise, whereas $\varepsilon$ and $\eta$ are variables that can be set in analytical equations or simulations, they are not directly given in empirical data and need to be estimated. The issue of estimation is treated in the next section. Note that throughout the text, the symbols $\omega$, $\varepsilon$ and $\eta$ are used to refer to known variables in analytical or modeling contexts, and to estimates of those variables in the description of our empirical data.

Below, we discuss the results of solving *Equation 1* analytically (see Appendix for more information), which allowed us to study changes in the phase-difference probability distribution as a function of detuning $\Delta\omega$ and interaction strength $\varepsilon$. The phase-difference probability distribution was

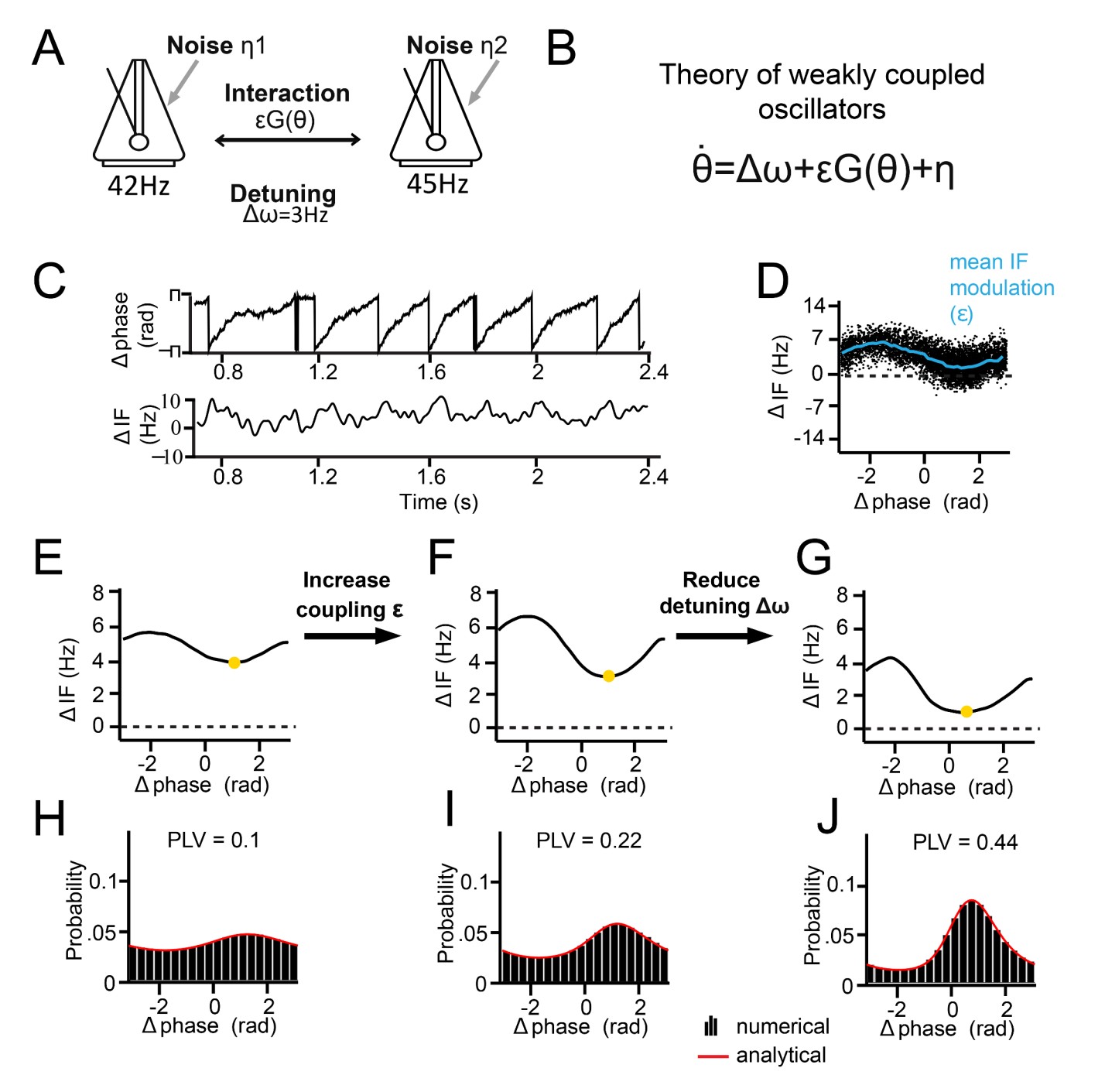

**Figure 4.** Theory of weakly coupled oscillators (TWCO). (A) Schematic illustration of the model. Two limit-cycle oscillators (here symbolized by metronomes) that mutually interact with strength ε and dependent on function G(θ). Each oscillator has its own intrinsic frequency ω and the difference is termed detuning Δω. Each oscillator additionally had phase noise η. (B) The single differential equation used for analysis. (C) Output example from numerical simulation of *Equation 1*. The phase precession is shown above and the ΔIF is shown below. Notice the ΔIF modulations over time. (D) ΔIF modulations averaged as a function of Δphase. (E–J) Equivalent behavior as in the examples shown in *Figure 3*. Top panels E-G show the modulation of the instantaneous frequency difference as a function of phase difference. Note that the instantaneous Δphase at which the ΔIF is minimal (yellow dot) is smaller when the interaction strength is larger (compare F, G to E). Bottom panels (H–J) show the phase difference probability distributions. Black bars are numerical simulation results, red lines indicate the analytical solutions. (E.H) Large detuning and low interaction strength. (F,I) Large detuning and strong interaction strength. (G,J) Small detuning and large interaction strength.
DOI: https://doi.org/10.7554/eLife.26642.010

characterized by the PLV and the mean (preferred) phase difference. The analytical solutions as a function of detuning Δω and interaction strength ε can be understood more easily by first considering the noise-free case. In the noise-free case (σ = 0), one can solve the equation for zero-points (equilibrium points), meaning that the phase precession is zero ($\dot{\theta} = 0$, i.e. zero frequency difference). To reach equilibrium, the detuning Δω and the interaction term εG(θ) need to be counterbalanced, and three cases can be considered. First, when detuning is smaller than the interaction strength ($|\Delta\omega|$ <=ε), there is a particular phase difference at which an equilibrium can be reached. At equilibrium, there is no phase precession and thus PLV equals 1 (full synchronization). Second, when interaction strength is zero (ε = 0), the asynchronous oscillators display continuous linear phase precession and have zero PLV, with the exception of zero detuning. Third, when detuning is larger than a nonzero interaction strength ($|\Delta\omega|$>ε, ε >0), oscillators exhibit nonlinear phase precession over time, characteristic for the intermittent synchronization regime (*Ermentrout and Rinzel, 1984b*; *Izhikevich, 2007*; *Pikovsky et al., 2002*, *Figure 4C*). The phase precession rate (instantaneous frequency difference) is determined by the detuning Δω, the modulation shape G(θ), and the modulation amplitude ε. Around the preferred phase-relation, the instantaneous frequency difference is reduced ('slow' precession in *Figure 4C*), whereas away from the preferred phase-relation, the instantaneous frequency is larger ('fast' precession in *Figure 4C*). For a given Δω and ε, a characteristic relationship can be predicted between ΔIF and Δphase (*Figure 4D*), indicative of the interaction function G(θ). Note that in the noiseless regime, a PLV between 0 and 1 can be obtained, varying between intermittent and full synchronization. However, including phase noise (σ > 0) has important effects on the synchronization behavior (*Izhikevich, 2007*; *Pikovsky et al., 2002*). The noise flattens the phase-relation distribution and can induce full cycles of phase precession (phase slips) that also lead to instantaneous frequency modulations. For noisy oscillators, the intermittent synchronization regime is the default regime for a large parameter range.

To show the applicability of the theory, we first reproduced the three empirical examples shown in *Figure 3* by numerical simulations of *Equation 1* and by varying detuning Δω and interaction strength ε. We assumed a sinusoidal G(θ) (see Kuramoto model, *Breakspear et al., 2010*; *Kuramoto, 1991*) and a phase variability of SD = 18 Hz (similar to our experimental data). As we did also with empirical data (see *Figures 5* and *6*), detuning was estimated here from the mean frequency difference at which the instantaneous frequency difference (ΔIF) modulations were centered, whereas the interaction strength was estimated from the amplitude of the modulations (*Figure 4D*). As shown in *Figure 4E–J*, our simulations showed the same relation between the instantaneous frequency difference modulations and the properties of the phase difference probability distribution as observed for V1 gamma (*Figure 3B–I*).

To test whether the same synchronization properties could be reproduced by simulation data from a more biologically plausible model, we constructed a model consisting of two mutually coupled pyramidal-interneuron gamma network (PING) networks (*Figure 3—figure supplement 2*). The PING network captures essential biophysical properties of cortical gamma rhythmicity (*Börgers et al., 2005*; *Fries, 2015*; *Tiesinga and Sejnowski, 2009*) and can be considered a biologically plausible instantiation of an oscillator in V1. As excitatory input drives gamma frequency (*Buia and Tiesinga, 2006*; *Jia et al., 2013b*; *Llinás et al., 1991*; *Ray and Maunsell, 2010*; *Roberts et al., 2013*), detuning was manipulated by independently varying the excitatory synaptic input strength to the two networks. The interaction strength was manipulated by changing the cross-network excitatory synaptic strengths. Using this more detailed model, we fully reproduced the synchronization properties obtained with the Kuramoto oscillator model (*Figure 3—figure supplement 2*). This shows that the latter model, despite its simplicity, captures essential aspects of neural synchronization.

## Estimating the underlying parameters and function of TWCO in observed data

To demonstrate the value of TWCO for understanding V1 gamma synchronization, we first assessed the ability of the theory to accurately predict monkey V1 recording data quantitatively (*Figure 5A*). Second, we tested whether we could reconstruct the Arnold tongue, which is a central prediction of the theory. The Arnold tongue describes the synchronization region in the parameter space of

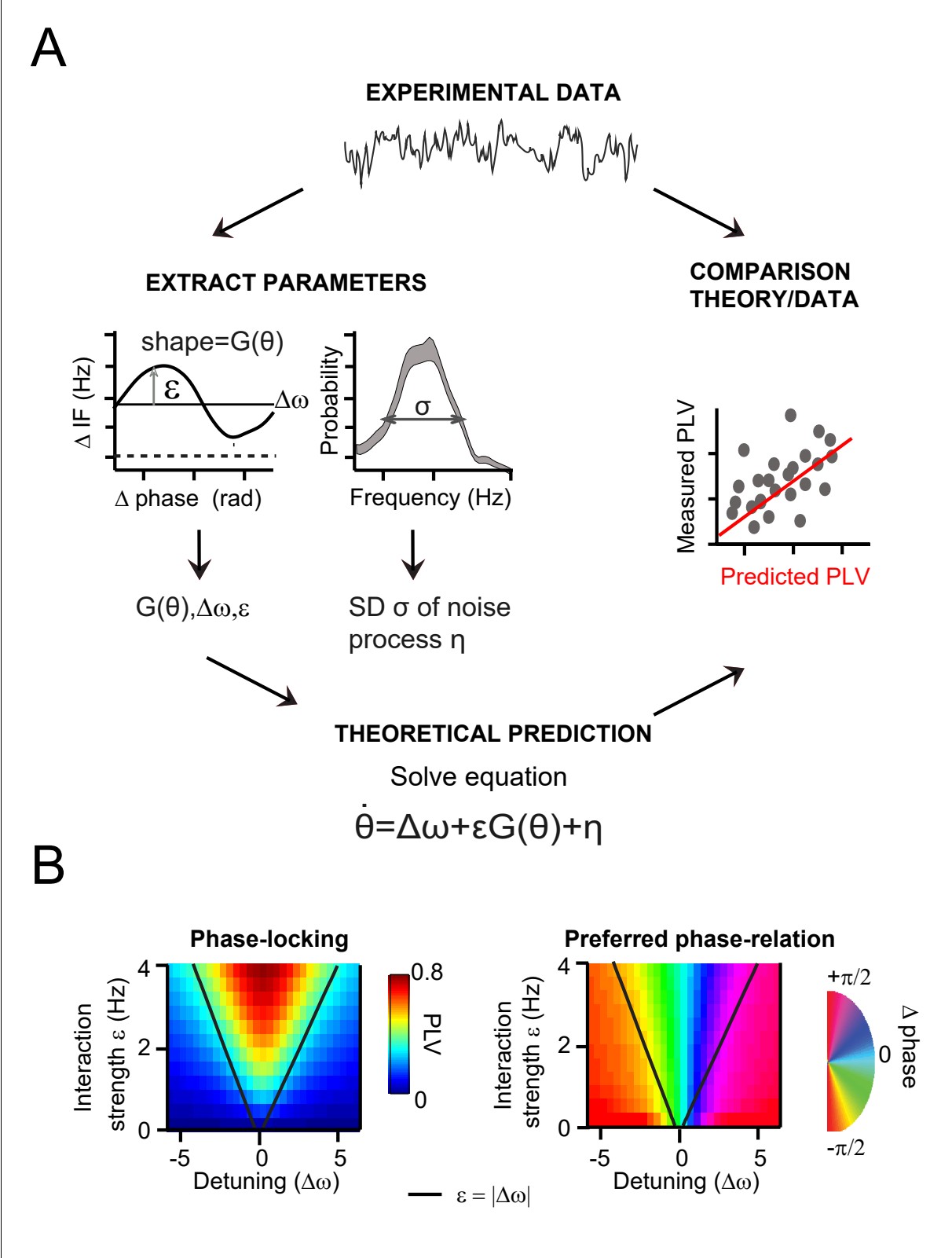

**Figure 5.** .General approach to derive and evaluate the theoretical predictions. (**A**) Schematic illustration of the main procedure to derive and evaluate the theoretical predictions for gamma PLV. From the experimental data (instantaneous frequency difference, top) we needed to estimate the function G (θ) and the parameters ε, Δω and σ to solve *Equation 1* (bottom), obtaining PLV predictions. We extracted (left) the function and parameters using observed $\overline{\Delta IF}(\theta)$(for G(θ),ε, Δω) and the gamma frequency distribution (for σ). We then solved *Equation 1* for each contact pair and compared directly
*Figure 5 continued on next page*

*Figure 5 continued*
the predicted and observed PLV (right, where each point represents one condition and contact pair) (B) The prediction of the Arnold tongue. In the parameter-space of ε and Δω, a characteristic inverted triangular-shaped synchronization region is as expected from TWCO. Left is the analytically derived PLV from *Equation 1* (where G(θ) being a sinusoid function and σ = 18 Hz). The black line represents the equality (ε=|Δω|), which sharply defines the Arnold tongue in the noise-free case. Right the mean phase difference is mapped showing a gradual change of phase-difference along the detuning dimension.
DOI: https://doi.org/10.7554/eLife.26642.011
The following figure supplement is available for figure 5:

**Figure supplement 1.** Testing the accuracy of the interaction function reconstruction.
DOI: https://doi.org/10.7554/eLife.26642.012

detuning and interaction strength (*Figure 5B*) and provides a general intuitive description of the gamma synchronization behavior.

To achieve the first goal, the theoretical parameters of *Equation (1)* need to be estimated. This equation can then be solved to predict the expected phase-difference probability distribution. Here, we were interested in two key properties of the distribution, the phase-locking value (PLV) and the mean phase difference. The theory predicts that the phase-difference-dependent modulation of the instantaneous frequency difference (ΔIF(θ)) is determined by the detuning Δω and the interaction term εG(θ). As shown above, we consistently observed modulations in ΔIF(θ) in our experimental datasets (*Figure 3*). Importantly, the time-averaged modulation of the instantaneous frequency $\Delta\overline{\mathrm{IF}}(\theta)$ directly relates to the deterministic term Δω+εG(θ), as noise is averaged out (see more in the Appendix). Based on this relation, the two parameters (Δω and ε) as well as the shape of function G(θ) were estimated from the experimentally observed modulation of ΔIF(θ) (*Figure 5A*, *Figure 5—figure supplement 1*). *Equation (1)* contains a white noise process η determined by variance $\sigma^2$ (mean = 0). The variance was determined by estimating the overall observed frequency variability in our gamma-band signal (taking SNR into account, see Appendix).

Based on these theoretical considerations, we estimated Δω and ε separately for each contact pair between probes in each experimental condition. The interaction strength ε was estimated by the modulation amplitude of the averaged modulation in the intrinsic frequency difference $\Delta\overline{\mathrm{IF}}(\theta)$. The detuning Δω was estimated by the average of the intrinsic frequency difference $\Delta\overline{\mathrm{IF}}(\theta)$ computed over the full range of instantaneous phase differences [-π π]. By contrast, we estimated a single G(θ) function and σ value from each monkey separately, therefore assuming stability of underlying PRCs and of the noise sources. The function G(θ) was estimated by the normalized $\Delta\overline{\mathrm{IF}}(\theta)$ modulation shapes. We validated the approach using phase-oscillator simulations (*Figure 5—figure supplement 1*). Note that the function G(θ) was estimated from data with absolute detuning of more than 4 Hz. This was done based on the observation that interaction functions became deformed when detuning was close to (see for more in Appendix). Further, it avoided smearing due to phase shifts occurring mainly within ±4 Hz. Given G(θ) and the value σ, the equation could be mathematically (analytically) solved for any values of detuning Δω and interaction strength ε. This means that for each contact pair and condition, we could derive precise predictions of differences in instantaneous frequency, phase relation, and phase locking (PLV) for comparison with the observed data.

A potential problem is that SNR influences both the PLV and the interaction strength estimate (problem of circularity). Further, the variables detuning and interaction might not be completely independent, due to factors like SNR. We therefore did not directly use the individual interaction strength values for comparison, but first binned contact pairs according to cortical distance (±0.25 mm). For each cortical distance, we then computed the averaged interaction strength. All contact pairs within a cortical distance bin were then assigned the same interaction strength. This step circumvented the problem of circularity and dependence of variables, but it also limited the maximum prediction accuracy that could be achieved.

To achieve the second goal, reconstructing the Arnold tongue, we mapped the observed PLV and mean phase differences as a function of detuning and interaction strength (using cortical distance binned as above) to obtain the Arnold tongue (*Pikovsky et al., 2002*). To demonstrate the expected shape of the synchronization region (*Figure 5B*), we mapped the analytically derived PLV

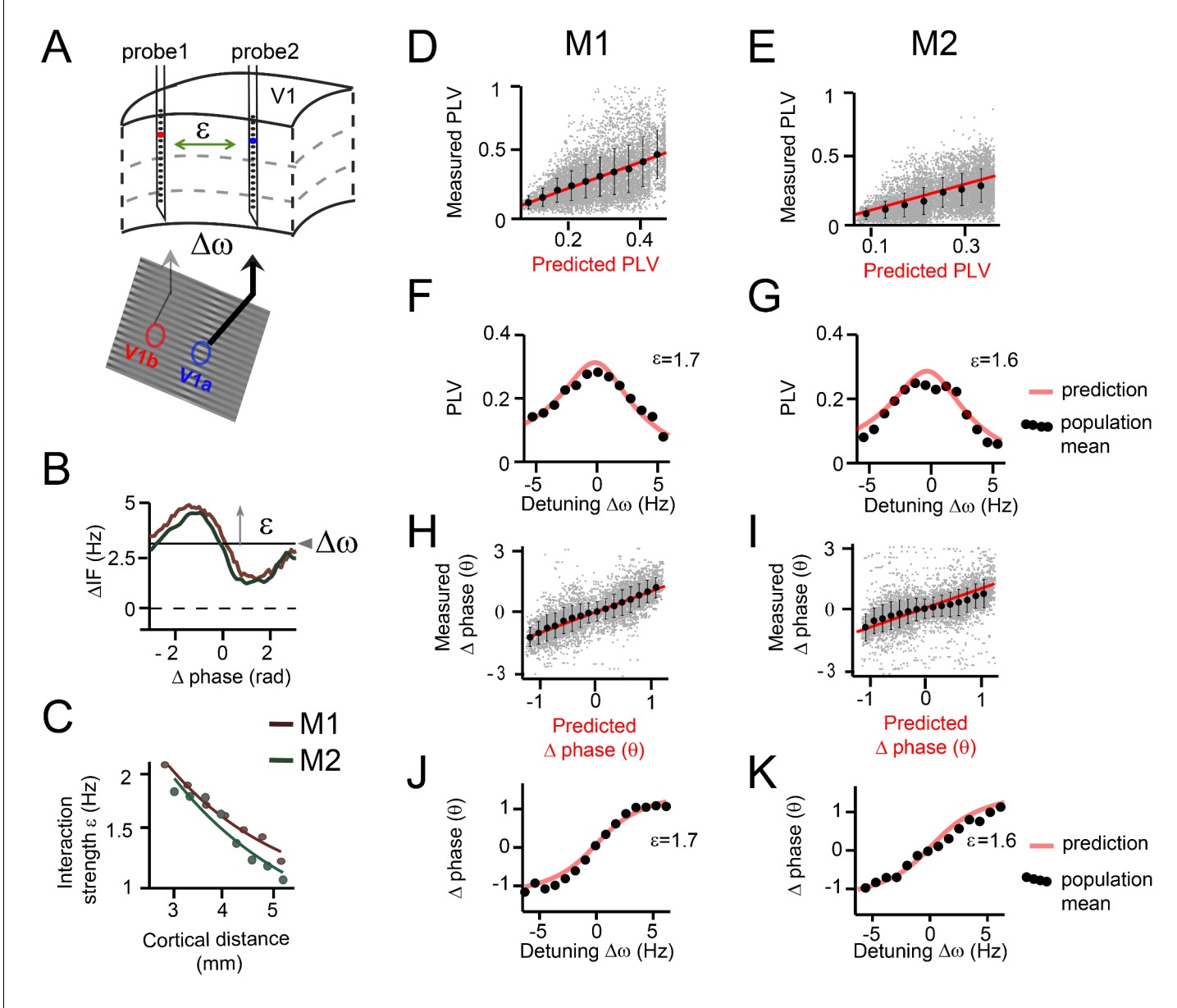

**Figure 6.** Predicting V1 gamma synchronization in monkeys M1 and M2. (A) Illustrative schema showing how detuning Δω and interaction strength ε of V1 gamma relate to local stimulus contrast and cortical distance respectively. (B) Example plots of averaged phase-dependent modulation of the instantaneous frequency difference (ΔIF) used for estimating ε and Δω for monkey M1 (brown) and M2 (green). The shape of the modulation indicates the G(θ). (C) Plots showing that the interaction strength ε decreased with cortical distance in both monkeys M1 and M2. (D, E) Each gray dot represents one single contact pair data and condition plotted as function of the observed PLV (y-axis) and the analytical predictions (x-axis). The red line shows unity line. The black dots represent the population means binned according to predicted PLV (+-SE). (F, G) The observed PLV population means (dots) and the analytical predictions (gray line) as a function of detuning Δω for one level of interaction strength (ε = 1.7 in M1; ε = 1.6 in M2). (H, I) Same as in (D, E), but now for mean (preferred) phase differences. (J, K) Similar to (F, G), but now for mean phase difference.
DOI: https://doi.org/10.7554/eLife.26642.013

and mean phase difference from TWCO *Equation (1)* in the Δω-ε parameter space. We observed a triangular synchronization region (*Figure 5B*) described as the Arnold tongue. This reflects the fact that stronger interaction strengths 'tolerate' larger detuning (|Δω|<=ε). Further, a clear phase gradient along the detuning dimension can be observed. The oscillator with a higher frequency led the oscillator with a lower frequency in terms of their phases.

## TWCO predicts synchronization properties of V1 cortical gamma rhythms

We then assessed whether the theory predicted the experimental gamma-band PLV values recorded from V1 (focusing on layers 2–4) using the estimation procedure as described above. We estimated for each contact pair and stimulus condition their detuning values (ranging from about −6 Hz to 6 Hz) as well as their interaction strength. The phase noise parameter and the interaction function G(θ) were estimated for the two monkeys separately.

The interaction function G(θ) was estimated as being approximately a sinusoidal function (*Figure 6B*) with relatively symmetric negative and positive components (*Akam et al., 2012*). This means that phase precession was accelerated (increase in frequency) or reduced (decrease in frequency) depending on the precise phase-difference. This type of interaction function allows for robust phase-locking for negative as well as positive detuning values (see symmetric Arnold tongue below). This is because negative detuning can be counterbalanced by the positive component of G(θ) and the positive detuning by the negative component of G(θ). It is worth noting that the interaction function G(θ) is not identical with the PRC. This is because the interaction function G(θ) is the convolution of the PRC with the coupling function (*Ermentrout, 1996*). In the present data, the exact form of the underlying (mainly synaptic) coupling function underlying V1 gamma synchronization was out of reach, and we only estimated here its overall strength ε. This contrasts with modeling data where synaptic coupling strengths are known and the coupling function can be computed. Nevertheless, the synaptic/electrical dynamics that underlie gamma rhythms are relatively fast, and we therefore expect that the interaction function G(θ) is closely related to the PRC. Hence, whenever we use the terms PRC and G(θ) in the context of our empirical data analysis, we keep their conceptual distinction in mind while considering them similar for practical purposes.

The phase noise parameter σ was found to be relatively large (M1: σ = 19 Hz, M2: σ = 20 Hz) indicating substantial frequency variability not explained by the interaction function (likely due to inherent noise and interactions with other cortical locations). The detuning Δω was positively correlated with the local contrast difference (linear regression, M1: $R^2$ = 0.28, M2: $R^2$ = 0.25, both $p < 10^{-10}$) and with MUA rate difference between probes (linear regression, M1: $R^2$ = 0.53, M2: $R^2$ = 0.36, both $p < 10^{-10}$) in line with *Ray and Maunsell, 2010*. The interaction strength ε was found to be inversely correlated with the cortical distance between probes (linear regression, M1: $R^2$ = 0.41, M2: $R^2$ = 0.29, both $p < 10^{-10}$, *Figure 6C*), in line with the known decrease of V1 horizontal connectivity with distance (*Stettler et al., 2002*).

To test further the idea that the interaction strength ε is a biologically meaningful measure of neural interaction more thoroughly, we repeated the analysis of interaction strength ε over cortical distance between probes with trial-shuffled data. A large interaction strength ε surviving the shuffling may reveal an influence of a stimulus-locked component on ε. This permutation analysis led to population-averaged IF modulation curves that were nearly flat, with values on average of ε = 0.31 Hz ± 0.002 in M1 and ε = 0.28 Hz ± 0.006 in M2. This is much lower than the ε values of 1–2 Hz observed without shuffling (*Figure 6C*). This may have been due to the fact we had only ~30 trials to shuffle per condition. This likely was not enough to obtain optimal randomization. Indeed, applying the same procedure to phase-oscillator simulations with 30 simulation trials also led to a remaining value of ε = 0.2 Hz ± 0.009. Furthermore, the higher the trial number, the closer the value got to zero (100 trials = 0.1 Hz ± 0.004, 500 trials = 0.05 Hz ± 0.002, 1000 trials = 0.03 Hz ± 0.004). Nevertheless, we cannot exclude that the small remaining non-zero value of ε after shuffling to some extent reflected a minor contribution of stimulus-dependent dynamics in our data. In an attempt to empirically test interaction strength in a case where no or weak anatomical connectivity is expected, we analyzed additional V1-V2 pair recordings with far-removed RFs in monkey M1 (Fig S9, G-I). The interaction strength we observed (ε = 0.3 Hz) was very small, not different from shuffled trials, in line with the expected weak connectivity between involved recordings sites. Altogether, these analyses support the conclusion that ε is a biologically meaningful measure of neural interaction. Having estimated detuning Δω, interaction strength ε, the interaction function G(θ), and the phase noise η, we were in a position to predict the properties of synchronization for each contact pair by solving the Kuramoto equation (*Figure 5A*).

We found that the gamma PLV variations over single contact pairs were significantly captured by the analytical predictions as a function of Δω and ε (model accuracy: M1: $R^2$ = 0.18, n = 7245, M2:

$R^2 = 0.32$, n = 7938, *Figure 6D,E*). This is particularly striking, given that the model predictions were derived out of first principles and single contact data were noisy. We also tested whether the model predicted variation of PLV evaluated for each single contact pair separately, where variation is induced mainly by detuning (model accuracy: M1: $R^2 = 0.27 \pm 0.0002$, n = 802, M2: $R^2 = 0.1 \pm 0.0001$, n = 882). The population means, defined as the averaged PLV values of contacts pairs with a similar detuning and cortical distance (bin size: ±0.35 Hz, ±0.3 mm), were very well predicted (model accuracy: M1: $R^2 = 0.83$, M2: $R^2 = 0.86$, both n = 638). To illustrate this, we plotted in *Figure 6F,G* the population means and the predictions for different detuning values for a single, medium interaction strength bin (M1: ε = 1.7, M2: ε = 1.6). The observed PLVs (dots) corresponded very well to the predictions (red line).

We also analyzed the mean phase difference (preferred phase-relation). A positive phase difference (phase X – phase Y) means that contact X leads (precedes in time) contact Y in terms of the phase of its oscillatory activity. Note that the temporal differences were smaller than the time scale of a full cycle, justifying the use of phase differences to indicate temporal ordering. The phase difference ranged nearly between –pi/2 to pi/2 in both M1 and M2. Again, single contact pair data was substantially captured by the analytical predictions as a function of Δω and ε (model accuracy: M1: $R^2 = 0.56$, n = 7245, M2: $R^2 = 0.3$, n=7938 *Figure 6H,I*). Furthermore, we tested whether the model predicted variability of phase difference evaluated for each single contact pair separately. This variability mainly represents variability induced specifically by detuning (model accuracy: M1: $R^2 = 0.52 \pm 0.0002$, n = 802, M2: $R^2 = 0.44 \pm 0.0004$, n = 882). The observed population means for different Δω and ε values followed the analytical predictions precisely (model accuracy: M1: $R^2 = 0.92$, M2: $R^2 = 0.88$, both n = 638). In *Figure 6J, K*, we plotted the population means and the predictions, but this time as a function of a range of detuning values for a medium interaction strength, further illustrating a good correspondence. The gamma rhythm with the higher frequency in a pair had the leading phase and the mean phase difference increased with increased detuning. To our knowledge, this is the first demonstration that phase locking values and preferred phase differences in primate cortex can be quantitatively predicted based on theoretical principles and limited knowledge of the system.

To further test the ability of TWCO to predict observed neural synchronization behavior, we plotted the observed CSD-CSD gamma PLVs in V1 as a function of Δω and ε for both M1 and M2. In this manner, we tested whether we would observe an Arnold tongue in the V1 data, which is a synchronization region with the shape of an inverted triangle defined by its regulative parameters Δω and ε and a core prediction of TWCO (See *Figure 5B*). *Figure 7A* shows the observed PLV (color-coded) plotted as a function of Δω and ε, revealing a structure that fitted the predicted Arnold tongue in both monkeys. As predicted, conditions of high interaction strength and low detuning showed strong gamma synchronization, whereas conditions of low interaction strength and high detuning yielded weak gamma synchronization. Notably, a model consisting of two coupled PING networks, in which interaction strength was manipulated by changing synaptic connectivity between the networks, and detuning by imposing differential excitatory drive, also yielded the Arnold tongue (*Figure 7—figure supplement 1*).

Using the estimated parameters, we also predicted the borders of the Arnold tongue analytically (black lines), which captured the outline of the observed Arnold tongue well. Due to intrinsic frequency variability (phase noise), the PLV values were not expected to decrease as sharply as expected from noiseless coupled oscillators (see *Figure 5B*). Further, in both monkeys (*Figure 7A* bottom), the map of mean phase difference showed a clear phase gradient across the detuning dimension as expected from the TWCO (*Figure 5B*). The results show that gamma rhythms with a higher frequency in a pair had the leading phase. Furthermore, for a given detuning, stronger interaction strength led to a reduction of the phase difference (see also yellow dots in *Figure 3B,E,H* and *Figure 4E,F,G*).

As an additional test of the robustness of our findings and their applicability to neural spiking data, we replicated our analysis in spike-CSD coupling measurements (see for more in the Appendix). We computed the PLV and mean phase difference between multi-unit activity (MUA) recorded from a contact of one probe and the CSD recorded from a contact of another probe. MUA activity was smoothed with a Gaussian kernel (σ = 4 ms) and demeaned to obtain a continuous spike density signal that was then analyzed similar to CSD signals. As shown in *Figure 7B*, we observed a similar

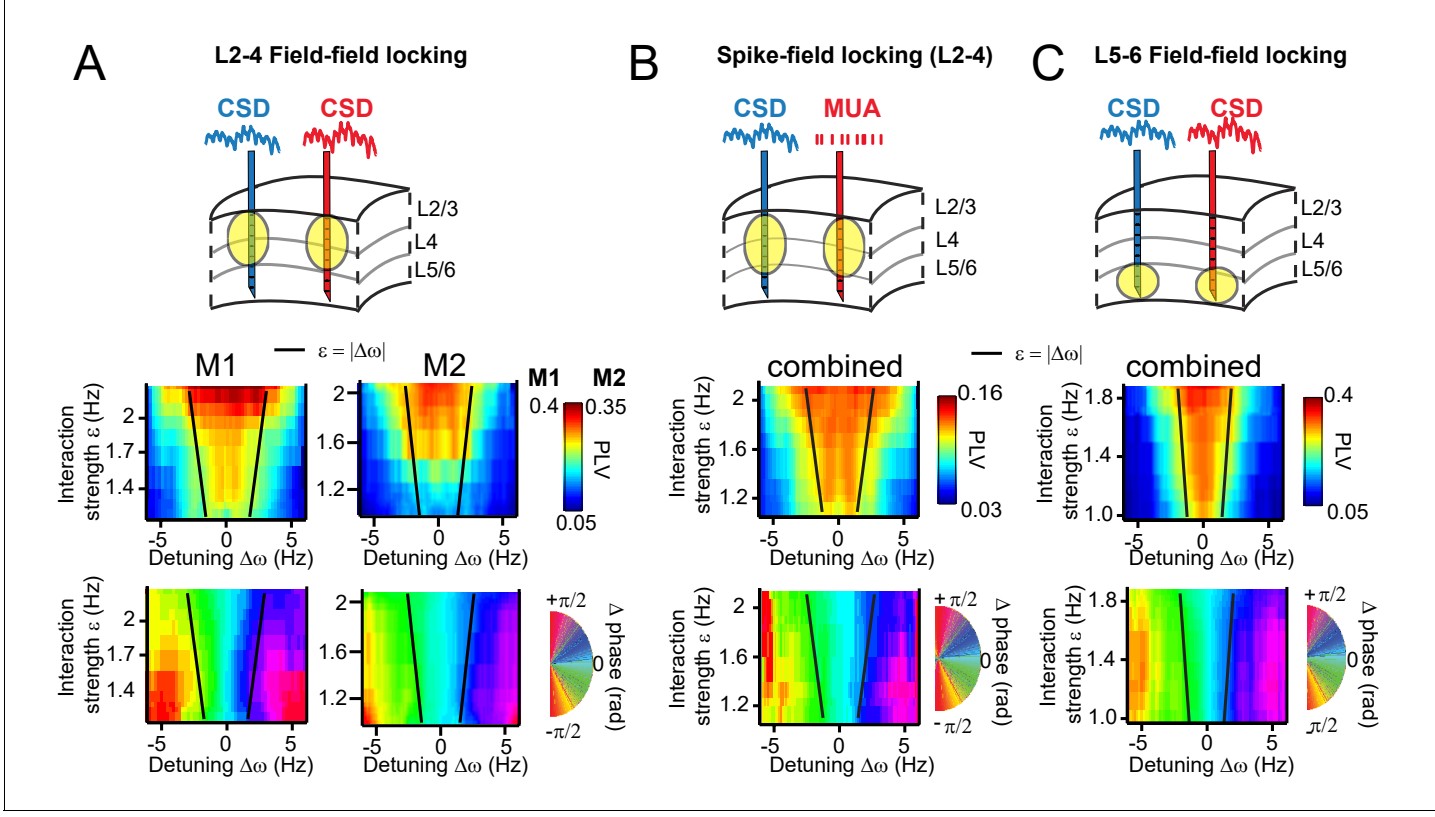

**Figure 7.** Arnold tongues. Combining different detuning Δω and interaction strengths ε, we observed a triangular region of high synchronization, the Arnold tongue. Black lines mark the predicted Arnold tongue borders as expected from the noise-free case (ε=|Δω|). (**A**) CSD-CSD PLV from V1 layers 2–4 are shown for both monkeys. Notice the inverted triangular shape of PLV values. Below, the mean phase difference is mapped in the same parameter space exhibiting a clear gradient with detuning. (**B**) Same analysis as in (A), but using MUA spikes from one contact and the CSD from the other contact. Here combined for both monkeys. (**C**) The same analysis as in (A), but using contacts from deep layer 5–6 in V1. We separated the analysis for L5/6 and L2/3, because we found strong coherence within each group, but weak coherence between the groups (see more in the Appendix).

DOI: https://doi.org/10.7554/eLife.26642.014

The following figure supplements are available for figure 7:

**Figure supplement 1.** Applying the theory of weakly coupled oscillators to coupled PING networks.

DOI: https://doi.org/10.7554/eLife.26642.015

**Figure supplement 2.** Arnold tongue mapping for CSD-MUA and MUA-MUA signals.

DOI: https://doi.org/10.7554/eLife.26642.016

**Figure supplement 3.** Phase-dependent instantaneous frequency modulation (and hence interaction strength ε) decreases with receptive field distance (and hence anatomical connectivity strength).

DOI: https://doi.org/10.7554/eLife.26642.017

**Figure supplement 4.** Gamma amplitude modulation as a function of phase-difference.

DOI: https://doi.org/10.7554/eLife.26642.018

Arnold tongue structure for spike-CSD measurements. The same analysis using Spike-Spike measurements also resulted in a similar Arnold tongue structure (**Figure 7—figure supplement 2**).

We have thus far confined analysis to pairs in middle and superficial layers. We therefore further separately investigated interactions between deep layers (**Figure 7C**). CSD-CSD analysis between deep contacts (L5-6) confirmed a similar Arnold tongue structure showing that the Arnold tongue properties do apply across the cortical layers. Our laminar probes reached also cortical area V2 lying beneath of V1 (**Figure 1—figure supplement 1**). We tested for V1-V2 pairs whether they exhibited similar phase-dependent instantaneous frequency modulations. We found that this was indeed the case (**Figure 7—figure supplement 3**).

The systematic variation of the phase difference between contact pairs by detuning indicates that detuning can affect the information flow between gamma rhythms (*Besserve et al., 2015*; *Buehlmann and Deco, 2010*; *Cannon et al., 2014*; *Lowet et al., 2016*). This is because spikes from a neural rhythm that leads another neural rhythm in time are more effective (*Buehlmann and Deco, 2010*; *Cannon et al., 2014*; *Fries, 2015*). To test this further, we mapped the main direction of Granger causal influence (see more in the Appendix) in the (CSD-CSD) gamma band (X→Y vs X←Y) as a function of detuning and interaction strength. We observed that a change in the sign of detuning and phase difference was linked to a change in the direction of strongest granger causality (*Figure 8*).

We observed one property of synchronization that was not accounted by the model equations. We found that gamma (instantaneous) amplitude (the absolute of analytical signal) varied weakly or moderately as a function of phase difference (*Figure 7—figure supplement 4*) in our experimental V1 data These amplitude variations were replicated also in simulation data of two mutually coupled PING spiking networks. The gamma amplitude variation became stronger with interaction strength. It has been shown before that increased mutual entrainment of synchronizing local gamma rhythms can enhance their amplitudes (*Womelsdorf et al., 2007*). However, TWCO remains highly predictive even in conditions of weak-to-moderate amplitude variations as long these variations do not strongly change the phase trajectory (*Izhikevich, 2007*; *Kopell and Ermentrout, 2002*; *Pikovsky et al., 2002*).

In *Figure 9*, we summarize schematically our main findings of how gamma synchronization between cortical locations is determined by their interaction strength and detuning and how it relates to the theory of weakly coupled oscillators, exemplified by the Arnold tongue. We propose that anatomical coupling is an important factor defining the interaction strength, however by itself is not sufficient to fully predict the amount of functional gamma-band interactions. Critical in addition is the amount of detuning that can functionally couple or decouple anatomically connected cortical locations. The crucial combined contribution of detuning $\Delta\omega$ and anatomical connectivity (related to $\varepsilon$) to synchronization is illustrated in three specific cases (*Figure 9A–C*), two of which yielding very low synchronization (with $\Delta\omega$, $\varepsilon$ coordinates falling just outside the Arnold tongue), and one of which yielding strong synchronization (with a $\Delta\omega$, $\varepsilon$ coordinate falling inside the Arnold tongue) (*Figure 9D–F*). Furthermore, *Figure 9A–C* (see arrows) illustrate that in the case of mutually anatomically coupled cortical locations, detuning influences the temporal relationship and possibly the direction of information flow between synchronized gamma rhythmic neural assemblies.

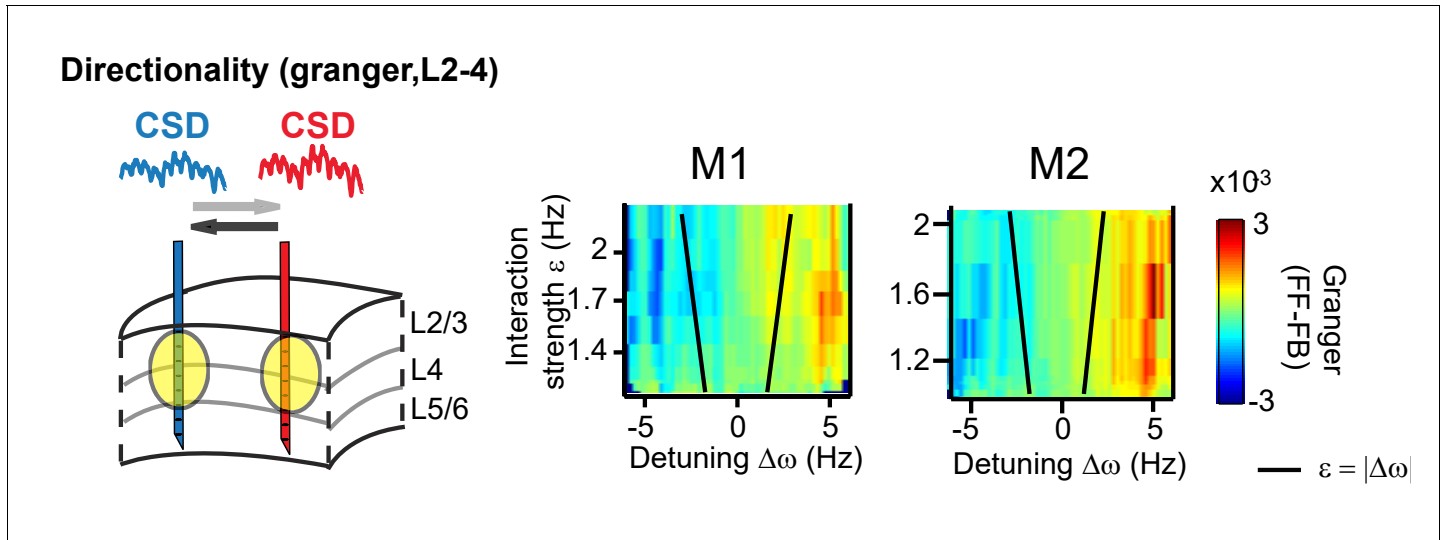

**Figure 8.** Same analysis as in *Figure 7*, but applying (non-stationary) granger causality directionality measure (X→Y vs X←Y). In line with phase-difference maps, the directionality influence flips as a function of detuning for monkey M1 (left) and M2 (right).
DOI: https://doi.org/10.7554/eLife.26642.019

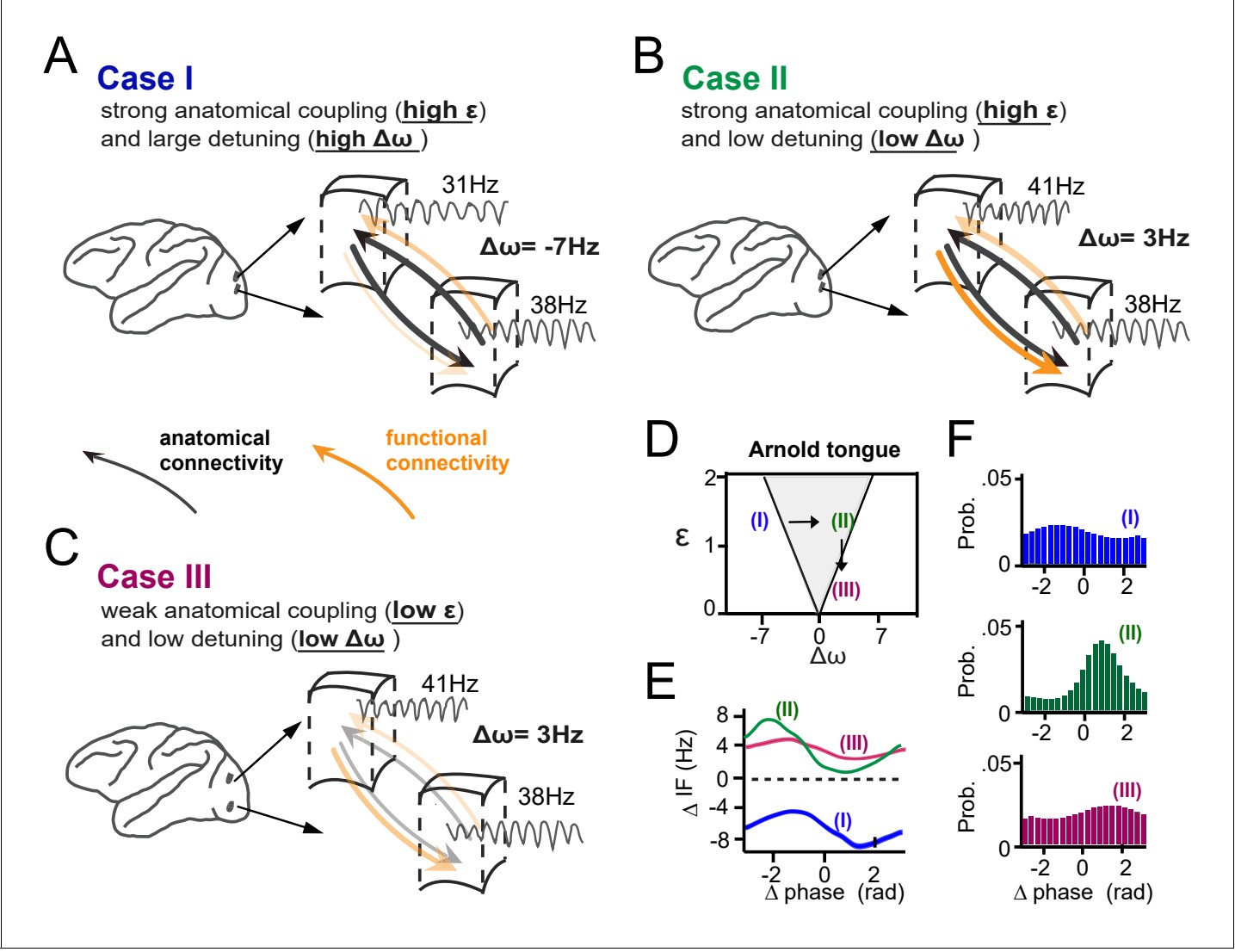

**Figure 9.** Summary of the main findings. (**A-C**) Three cases of cortical gamma-band interactions are used for illustration (**A**) In case I, two cortical locations have strong anatomical connections (black thick arrows, high interaction strength ε) and a large detuning Δω. This results in low functional interaction (orange arrows). (**B**) In case II, anatomical connections are as high as in (A), but detuning is low. This leads to strong functional interactions. The location with higher frequency functionally dominates the location with lower frequency. (**C**) In case III, there is the same low detuning as in (B), but with low anatomical connectivity. This results again in low functional interaction. (**D**) The three cases represented in relation to the Arnold tongue. Only case II is within the Arnold tongue. Moving out of the Arnold tongue by a change in Δω (from II to I) or a change in ε (from II to III) strongly reduces synchronization. (**E**) The instantaneous frequency difference modulations (ΔIF(θ)) as a function of phase-difference for the three examples. (**F**) The corresponding phase-difference probability distributions.

DOI: https://doi.org/10.7554/eLife.26642.020

## Discussion

The present study shows that gamma synchronization in awake monkey V1 adheres to theoretical principles of weakly coupled oscillators (*Ermentrout and Kleinfeld, 2001*; *Hoppensteadt and Izhikevich, 1998*; *Kopell and Ermentrout, 2002*; *Kuramoto, 1991*; *Pikovsky et al., 2002*; *Winfree, 1967*), thereby providing insight into the synchronization regime of gamma rhythms and its principles. Given the generality of the synchronization principles, they are likely to also apply to other brain regions and frequency bands.

## Intermittent synchronization: the role of non-stationary frequency modulations

Our findings reveal the importance of phase-dependent frequency modulations for synchronizing V1 gamma rhythms. These modulations show that a fixed and common frequency is not required for phase coordination. To the contrary, stronger non-stationary frequency modulations led to stronger synchronization, and thus to more reliable phase coordination. Frequency modulations arise naturally in the intermittent synchronization regime (*Ermentrout and Rinzel, 1984b*; *Izhikevich, 2007*; *Pikovsky et al., 2002*), when oscillators cannot remain in a stable equilibrium due to detuning and noise. Given the variable nature of gamma rhythms in vivo (*Atallah and Scanziani, 2009*; *Burns et al., 2010*; *Ray and Maunsell, 2010*; *Roberts et al., 2013*), intermittent synchronization is the most likely regime for their phase coordination. Although complete synchronization is not achieved in this regime, phase coordination remains sufficiently robust to influence the strength and directionality of information flow, by rendering particular phase-relations more likely than others (*Battaglia et al., 2012*; *Buehlmann and Deco, 2010*; *Fries, 2015*; *Maris et al., 2016*). The observation of non-stationary frequency modulations also has methodological implications. Gamma rhythms are often studied with stationary methods, for example spectral coherence or stationary Granger measures, yet our findings are not in line with the (weak-sense) stationarity assumption (*Lachaux et al., 1999*; *Lowet et al., 2016*). Time-resolved non-stationary methods are therefore more appropriate to study the dynamics underling gamma synchronization (*Bonizzi et al., 2014*; *Huang, 2005*; *Lachaux et al., 1999*).

## The Arnold tongue and the regulative parameters of gamma synchronization

Previous studies have established diversity in the phase-locking (*Eckhorn et al., 2001*; *Gray and Singer, 1989*; *Ray and Maunsell, 2010*) and in the phase-relations (*Maris et al., 2016*; *Vinck et al., 2010*) of gamma rhythms in the primate visual cortex. However, how this observed diversity in phase-relation and phase-locking is regulated was not well established. Here, we show that mainly two parameters determined gamma synchronization: the detuning $\Delta\omega$ and the interaction strength ε. This was highlighted in the mapping of the Arnold tongue, offering a graphical understanding of how these parameters shape gamma-band synchronization. Detuning represents a desynchronization force, whereas the interaction strength represents a synchronization force. In our experiment, the former was modulated by input drive differences associated with different local contrasts, and the latter by changes in connectivity strength associated with horizontal cortical distances between electrodes. Their interplay defined the resultant phase-locking strength and the preferred phase-relation between gamma rhythms. The observed role of detuning is in agreement with a previous study in the rat hippocampus (*Akam et al., 2012*), in which optogenetic entrainment strength and phase of gamma rhythms were dependent on the frequency-detuning. The results also agree with theoretical concepts of oscillatory interactions (*Ermentrout and Kopell, 1984a*; *Hoppensteadt and Izhikevich, 1998*; *Sancristóbal et al., 2014*; *Tiesinga and Sejnowski, 2010*). We suggest that small detuning values (mainly $<\Delta10$ Hz) reported in the present study and much larger shifts in the gamma frequency-range (25–50 Hz to 65–120 Hz) as reported in the rat hippocampus (*Colgin et al., 2009*) represent different but complementary mechanisms for controlling gamma synchronization. On the one hand, only a small difference in gamma frequency will leave a possibility for synchronization while a large difference will preclude synchronization. So, large shifts in detuning open or close opportunities for synchronization. On the other hand, at small levels of detuning that offer opportunities for synchronization, small changes in instantaneous frequency will modulate the exact strength and direction of the gamma-mediated information flow. Hence, instantaneous frequency modulations, which define the interaction strength, reflect the overall ability of two cortical locations to engage in gamma-band synchronization. These modulations are mediated by anatomical connectivity and further modified by oscillation amplitude. Hence, an important source of instantaneous V1 gamma frequency modulations is the underlying network (intermittent) synchronization process, which means that variations in gamma frequencies do not argue against a functional role of gamma synchronization (see *Bosman et al., 2009*; *Burns et al., 2011*; *2010*; *Roberts et al., 2013*). Furthermore, we show that the shape of the instantaneous frequency modulations reflects the underlying interaction function G(θ), which in our recording data likely is closely related to the PRC

(*Hoppensteadt and Izhikevich, 1998*; *Kopell and Ermentrout, 2002*; *Kuramoto, 1991*; *Pikovsky et al., 2002*; *Winfree, 1967*). The interaction function describes how the oscillators advance or delay each other's phase development to coordinate their phase-relation. We observed approximately symmetric sinusoidal-like functions in V1 gamma that resemble the basic function of the widely-used Kuramoto-model (*Breakspear et al., 2010*). This is in agreement with the biphasic PRC of gamma rhythms observed in the rat hippocampus (*Akam et al., 2012*) and fits with our observed symmetric Arnold tongues (*Izhikevich, 2007*; *Kopell and Ermentrout, 2002*; *Pikovsky et al., 2002*). Importantly, here we estimated the bidirectional interaction function G(θ). This function can be symmetric despite the presence of asymmetric individual (unidirectional) PRCs (*Cannon and Kopell, 2015*; *Wang et al., 2013*), as long as the rhythms interact approximately equally strongly, which is a plausible assumption between V1 locations.

The interaction functions we estimated here might be smoother than they really are due to limitations of our analysis arising from noise, averaging, and steps taken to reduce volume conduction. Future studies are required to characterize in more detail the (unidirectional/bi-directional) gamma-band interaction functions. Unidirectionally connected neural groups, for example between certain cortical areas, might have asymmetric interaction functions and an asymmetric Arnold tongue. In this situation, a frequency difference between cortical areas (*Bosman et al., 2012*; *Cannon et al., 2014*) might be favorable for optimal information transmission.

We found small-to-moderate phase-dependent variations of oscillation amplitude, which were not accounted for by the model equations. They were observed both in V1 data and PING simulations, indicating they are of biological origin. Future work is necessary to better understand their relevance. In addition, we assumed that synchronization between V1 locations emerged due to mutual horizontal interactions, yet common input fluctuations might further shape V1 gamma synchronization (*Wang et al., 2000*; *Wiesenfeld and Moss, 1995*; *Zhou et al., 2013*), especially for neurons with similar receptive fields. Although we did not investigate the possible effects of common input, the observation that gamma synchronization occurred between V1 locations with distinct receptive fields and with a dependence on cortical distance as expected from anatomical connectivity (*Gail et al., 2000*; *Gieselmann and Thiele, 2008*; *Palanca and DeAngelis, 2005*; *Ray and Maunsell, 2010*; *Stettler et al., 2002*) indicates that cross-columnar gamma-band synchronization depends strongly on direct mutual horizontal interactions (*Veit et al., 2017*).

## Role of V1 gamma synchronization for visual processing and broader relevance

In our experiment, detuning was dependent on the local contrast difference (*Ray and Maunsell, 2010*; *Roberts et al., 2013*), known to change neural excitation in V1 (*Sclar et al., 1990*), while the interaction strength was dependent on the underlying horizontal connectivity strength, here varied by cortical distance (*Stettler et al., 2002*). Gamma synchronization is therefore informative about the sensory input (*Besserve et al., 2015*) and about the underlying structure of connectivity. Indeed, the frequency of gamma rhythms is modulated by various sensory stimuli (*Fries, 2015*) and by cognitive manipulations (*Bosman et al., 2012*; *Buzsáki and Wang, 2012*; *Fries, 2015*) suggesting that frequency control is a potential avenue for modulating functional gamma-band coordination and information transfer (*Besserve et al., 2015*; *Buehlmann and Deco, 2010*; *Lowet et al., 2016*). Further, as phase lag is dependent on detuning, detuning may influence the direction of information flow among mutually coupled oscillators. This is in line with granger causality analysis in our paper (*Figure 8*), but also with network simulations published by us and others (*Besserve et al., 2015*; *Buehlmann and Deco, 2010*; *Cannon et al., 2014*; *Lowet et al., 2016*) showing that detuning will shape the information flow between model networks as measured by information theoretical tools (e.g. transfer entropy). Nevertheless, much more work is needed to explore the influence of detuning on directionality of information flow, and the results in the present paper are only suggestive.

The effect of detuning on synchronization was strongly modulated by interaction strength, which we demonstrated to relate strongly to the strength of horizontal connectivity. Horizontal connectivity in V1 is not only local, but also exhibits remarkable tuning to visual features, orientation being a prime example (*Stettler et al., 2002*). Hence, innate and learned connectivity patterns likely affect the interaction strength and hence the synchronization patterns of gamma rhythms within V1. These properties suggest V1 gamma as a functional mechanism for early vision (*Eckhorn et al., 2001*; *Gray and Singer, 1989*) by temporally coordinating local neural activity as a function of sensory

input and connectivity. In agreement with previous studies (*Eckhorn et al., 2001*; *Palanca and DeAngelis, 2005*; *Ray and Maunsell, 2010*), V1 gamma synchronization was found to be mainly local and limited to a narrow range of frequency differences. It is therefore not likely that gamma within V1 'binds' whole perceptual objects. Instead, it is more likely to bind features locally at the level of surround receptive fields. Furthermore, recent studies on the gamma-band response during natural viewing (*Brunet et al., 2015*; *Hermes et al., 2015*) have found variable levels of synchronization power for different natural images. In accordance with these observations, the revealed Arnold tongue of V1 gamma implies that natural image parts with high input/detuning variability (heterogeneity) will induce no or weak synchronization, whereas parts with low input/detuning variability (homogeneity) will induce stronger synchronization. This is also in line with proposals linking gamma synchronization with surround suppression/normalization (*Gieselmann and Thiele, 2008*; *Ray et al., 2013*) and predictive coding (*Vinck and Bosman, 2016*). Our findings and theoretical interpretation shed new light onto the operation of gamma synchronization in the brain and will permit new and more detailed descriptions of the mechanisms by which synchronization is regulated by cognitive and sensory inputs.

Finally, we propose that the mechanism we have described for gamma synchronization in V1 also holds outside the visual cortex. Gamma synchronization across cortical areas have been observed in spite of frequency differences (*Bosman et al., 2012*; *Gregoriou et al., 2009*), which is further supported by our additional analysis of V1-V2 interactions. Together, this suggests that similar principles likely operate for gamma-band inter-areal interactions. Further, the instantaneous gamma frequency fluctuations that we have shown to be instrumental in regulating synchronization, have also been observed in the rat hippocampus by *Atallah and Scanziani (2009)*. Their analysis suggested that these fluctuations, which reflected rapid phase shifts due to changes in excitation-inhibition balance, might be critical for gamma-mediated information flow. Likewise, *Nguyen et al. (2009)* observed instantaneous frequency modulations during ripples in rodent hippocampus, revealing dynamics that may be indicative of processes related to learning and memory. These findings support our proposal that cycle-by-cycle modulations in frequency that regulate gamma synchronization also happen in other frequency bands and in other brain regions or structures. Nevertheless, future studies are required to test to what extent weakly coupled oscillator principles apply to different frequency bands across brain regions. Importantly, as long as the instantaneous phase of a neural rhythm can be determined, the methods used in this study can be applied. Instantaneous phase extraction has been for example applied to theta rhythms (*Belluscio et al., 2012*; *Buzsáki, 2002*) or alpha rhythms (*Lakatos et al., 2005*; *Samaha and Postle, 2015*; *Schwabedal et al., 2016*). In future studies, optogenetic tools (*Boyden et al., 2005*; *Fenno et al., 2011*; *Zhang et al., 2007*) will be highly useful to modify oscillation properties like detuning in a precise manner. Variation of sensory or cognitive variables can also be a powerful and natural way of modulating network states if enough is known about the system (e.g., *Bosman et al., 2012*). Interaction strength could be estimated from anatomical knowledge or manipulated by optogenetics (e.g. by targeting cell-types involved in a specific type of anatomical connectivity and varying oscillation amplitude). Aside of emerging new technological possibilities for network state modulation, a tight combination of experimental and dynamic systems theory will be critical for fruitful analysis and interpretation of neural oscillatory data.

In summary, the present paper offers the first predictive theory of synchronization, which we suggest can be used to assess the mechanisms of synchronization in various frequency bands, and to assess their contribution to diverse forms of cognition.

## Materials and methods

### Species used and surgical procedures

Two adult male rhesus monkeys were used in this study. A chamber was implanted above early visual cortex, positioned over V1/V2. A head post was implanted to head-fix the monkeys during the experiment. All the procedures were in accordance with the European council directive 2010/63/EU, the Dutch 'experiments on animal acts' (1997) and approved by the Radboud University ethical committee on experiments with animals (Dier Experimenten Commissie, DEC).

## Recording methods

V1 recordings were made with 2 or 3 Plexon U-probes (Plexon Inc.) consisting of 16 contacts (150 µm inter-contact spacing). We recorded the local field potential (LFP) and multi-unit spiking activity (MUA). For the main analysis, we used the current-source density (CSD, (*Vaknin et al., 1988*)) to reduce volume conduction. We aligned the neural data from the different laminar probes according to their cortical depth and excluded contacts coming from deep V2. Layer assignment was based on the stimulus-onset CSD profile (*Schroeder et al., 1991a*) and the inter-laminar coherence pattern (*Maier, 2010b*). Receptive field (RF) mapping was achieved by presenting at fast rate high-contrast black and white squares pseudorandomly on a 10 × 10 grid (*Roberts et al., 2013*). For RF mapping we used CSD signals and spikes.

## Task and visual stimuli

The monkeys were trained for head-fixation and were placed in a Faraday-isolated darkened booth at a distance of 57 cm from a computer screen. Stimuli were presented on a Samsung TFT screen (SyncMaster 940bf, 38°x30° 60 Hz). During stimulation (2 s) and pre-stimulus time (1 s) the monkey maintained a central eye position (measured by infra-red camera, Arrington, 60 Hz sampling rate). The monkey's task was to passively gaze on a fixation point while a stimulus was shown. The monkey was rewarded for correct trials. The local stimulus contrast was manipulated in a full screen static square-wave grating (2 cycles/degree, presented at two opposite phases randomly interleaved). Contrast was varied smoothly over space such that different RFs had different contrast values. The direction of the contrast difference was parallel to the arrangement of RFs and orthogonal to the orientation of the grating. The stimulus was isoluminant with the pre-stimulus grey screen. We presented 9 different contrast modulation conditions (Table.S1). Cortex software (http://dally.nimh.nih.gov/index.html) was used for visual stimulation and behavioral control.

## Data analysis

We analyzed gamma rhythms in the visual stimulation period (0.2 s - 2 s). We discarded the first 200 ms to avoid stimulus-onset transients. To investigate dynamical changes in the gamma phase and frequency over time, we estimated the instantaneous gamma phase and frequency using the singular spectrum decomposition of the signal (SSD [*Bonizzi et al., 2014*]) combined with Hilbert-Transform or wavelet-decomposition. The phase-locking value (PLV) was estimated as the mean resultant vector length (*Lachaux et al., 1999*) and the preferred phase-relation as the mean resultant vector angle. For experimental data, we estimated the signal-to-noise ratio (SNR) to reduce the influence of measurement noise on estimates. Phase flipping due to CSD computation was corrected.

## Theoretical and computational modeling

Using the theory of weakly coupled oscillators, we investigated the phase-locking as well as the mean phase difference of two mutually coupled noisy phase-oscillators with variable frequency difference (detuning) and interaction strength. The stochastic differential equation was solved analytically (*Pikovsky et al., 2002*). The analytical results correctly predicted the numerical simulations.

## Statistics

The accuracy of the theoretical predictions for the experimental data was quantified as the explained variance $R^2$.

## Data availability

Experimental data sets, modeling and analysis tools are available to all interested researchers upon request from the corresponding author. For singular spectrum decomposition visit *https://project.dke.maastrichtuniversity.nl/ssd/*.

## Acknowledgements

We thank N Kopell, W Singer, A Bastos, P Fries, C Micheli, F Smulders, J.v.d. Eerden, J Karel, P Bonizzi, A Hadjipapas, AA.v.d. Berg for discussion. Supported by NWO VICI grant 453-04-002 to PDW and NWO VENI grant 451-09-025 to MJR. All data are stored at the Department of Psychology

and Neuroscience, Maastricht University, The Netherlands. We thank the Radboud University Nijmegen for hosting our experiments, and staff of the Central Animal Facility (CDL) for expert assistance.

## Additional information

### Funding

| Funder | Grant reference number | Author |
|---|---|---|
| Nederlandse Organisatie voor Wetenschappelijk Onderzoek | 451-09-025 | Mark Jonathan Roberts |
| Nederlandse Organisatie voor Wetenschappelijk Onderzoek | 453-04-002 | Peter De Weerd |

The funders had no role in study design, data collection and interpretation, or the decision to submit the work for publication.

### Author contributions
Eric Lowet, Conceptualization, Formal analysis, Investigation, Methodology, Writing—review and editing; Mark J Roberts, Conceptualization, Investigation, Supervision, Writing—review and editing; Alina Peter, Formal analysis, Methodology, Writing—review and editing; Bart Gips, Methodology, Writing—review and editing; Peter De Weerd, Supervision, Writing—review and editing

### Author ORCIDs
Eric Lowet http://orcid.org/0000-0002-9793-0639

### Ethics
Animal experimentation: All the procedures were in accordance with the European council directive 2010/63/EU, the Dutch 'experiments on animal acts' (1997) and approved by the Radboud University ethical committee on experiments with animals (Dier-Experimenten-Commissie, DEC).

### Decision letter and Author response
Decision letter https://doi.org/10.7554/eLife.26642.025
Author response https://doi.org/10.7554/eLife.26642.026

## Additional files

### Supplementary files
• Transparent reporting form
DOI: https://doi.org/10.7554/eLife.26642.021

### Major datasets
The following dataset was generated:

| Author(s) | Year | Dataset title | Dataset URL | Database, license, and accessibility information |
|---|---|---|---|---|
| Eric Lowet, Mark J Roberts, Alina Peter, Bart Gips, Peter De Weerd | 2017 | Data from: A quantitative theory of gamma synchronization in macaque V1 | http://dx.doi.org/10.5061/dryad.jv42j | Available at Dryad Digital Repository under a CC0 Public Domain Dedication |

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

# Appendix

DOI: https://doi.org/10.7554/eLife.26642.022

## Surgical procedures

Two adult male rhesus monkeys (Macaca mulatta) were used in this experiment. Two chambers were implanted above early visual cortex, one positioned over V1/V2 and the second over V4. For the experiment reported here we used data from the V1/V2 chamber only. A head post was implanted to head-fix the monkey during the experiment. All the procedures were in accordance with the European council directive 2010/63/*EU*, the Dutch 'experiments on animal acts' (1997) and approved by the Radboud University ethical committee on experiments with animals (Dier-Experimenten-Commissie, DEC).

## Recording techniques

V1 recordings were made with Plexon U-probes (Plexon Inc.) consisting of 16 contacts (10 μm diameter, 0.5–1 mΩ impedance, and 150 μm inter-contact spacing). Three probes were inserted through a sharp guide tube, which was lowered through granulation tissue to just above the level of the dura surface. The probes were arranged in a linear manner separated from each other by ~2–3 mm. The probes were then advanced by separate microdrives (Nan Instruments LTD.). The probes were connected to headstages of high input impedance, and data were acquired via the Plexon 'Multichannel Acquisition system' (MAP, Plexon Inc.). The measured extracellular signal was filtered online between 150 Hz and 8 kHz to extract spiking activity and filtered between 0.7 Hz and 300 Hz to obtain the' local field potential' (LFP). The signal was amplified and digitized with 1 kHz for the LFP and 40 kHz for the spike signal. The data was converted from Plexon to Matlab file format and cut into trials from fixation onset to stimulus offset using the fieldtrip toolbox (*Oostenveld et al., 2011*). For the LFP data, the line noise was removed using the fieldtrip toolbox dft filter, which fits a sine and a cosine at 50, 100 and 150 Hertz and subtracts these components from the data. We collected 7 recording sessions in monkey M1 and 6 sessions in M2. Each recording session had on average ~590 trials in M1 and ~718 trials in M2.

## Current source density (CSD)

First for extrapolating the CSD to the outermost contacts of our probes, at the top and bottom of the probe, a replica of the LFP of respectively the first and last contact was appended (*Vaknin et al., 1988*). The LFP was then smoothed with a Gaussian (zero-phase) filter of a SD of 1.2 and range of 5 (effectively weighting signals around the centre electrode by 24% in the centre, 20% immediate neighbours, 12% 2 contacts away, 5% 3 contacts away). Then the standard CSD algorithm was applied for each contact position x, our inter-contact spacing h of 150 μm and a conductivity C of 0.3 S/m:

$$CSD(x) = -C * \frac{LFP(x-h) - 2LFP(x) + LFP(x+h)}{h^2} \tag{1}$$

We used CSD signals for the main analysis to reduce effects of volume conduction (see also section *MUA-CSD and MUA-MUA analysis*).

## Receptive field mapping

Receptive fields (RFs) were mapped using both spiking and LFP information as described in (*Roberts et al., 2013*). Briefly, monkeys fixated centrally while high-contrast black and white squares of sizes 0.1-1degree were presented pseudorandomly on a 10 × 10 grid. The locations where the spiking or the LFP response exceeded the 75th percentile of the response distribution were defined as the RF. Other than in *Roberts et al. (2013)*, the LFP response was also used based on the envelope of the broadband gamma power (30–150 Hz) in the CSD, which we found to produce a localized result in line with spiking RFs. CSDs were computed as

described above, but with a smaller Gaussian filtering of SD 0.6 and a filter range of 2, meaning that only the two neighbouring electrodes of the centre electrode had some iAcademic Pressnfluence on the RF estimate of a given contact. This was done to avoid mislocalization of RF shifts in size or position that are indicative of a shift to a different column or to V2 (*Gattass et al., 1981*) (see *Figure 1—figure supplement 1B*, rightmost plots for an example of CSD and spiking RFs with such a shift). To obtain estimates of cortical distance (in mm) between the probes we took advantage of the well-known retinotopy of V1. We measured the distance between RF centres and calculated the cortical distance by converting differences in visual degrees using a cortical magnification factor (CMF, [*Schwartz, 1980*; *Sereno et al., 1995*]). The CMF was estimated individually for each monkey where we used the measured the physical distance between the laminar probes (fixed to the holder) before insertion into cortex (M1:~2.7 mm/deg, M2:~2.5 mm/deg).

## Laminar alignment

We inserted the laminar probes on each recording day. The exact laminar positions of the probes (*Figure 1—figure supplement 1*) differed within and between sessions and hence we depth-aligned the probes based on their stimulus-evoked response and inter-laminar coherence characteristics (*Maier, 2010b*). For depth-alignment (to assign each contact a particular cortical depth value) we used the following procedure:

1. We computed the CSD-VEP response. The different sink-source profiles were aligned using a parallel-tempering technique (*Frenkel and Smit, 2001*). This is an iterative procedure that minimizes the squared error between all probes, shifting the position of one probe by one position on each iteration. Central to the parallel tempering algorithm is the parallel start of the procedure at multiple 'temperatures', each of which in our case starts with a different initial, random offset in the probes. Higher temperatures accept higher increases in error with a shift in the position of a probe. If a procedure running at a high temperature achieves a lower error than another temperature (overcoming a local minimum), it swaps the achieved shift vector with a lower temperature to find the new minimum around it. Similar to (*Godlove et al., 2014*) (using a genetic algorithm), we implemented a lenient maximum shift constraint between electrodes (allowing by shifts of 4 channels upwards and downwards, which for any two probes enforces a minimal overlap of 50%) to prevent trivial solutions. For our data, we used 3000 iterations at 4 different temperatures and different error tolerances per temperature (log spaced between zero and 1). The procedure showed asymptotic behaviour (no further decrease in error) at <= 1500 iterations. Note that the optimal number of iterations required for this algorithm will depend on the number of probes/sessions entered.
2. We then computed the within laminar probe LFP coherence matrix (*Carter et al., 1973*). It has been shown that there is sharp decrease in coherence around the L4/L5 border (*Maier, 2010b*). We chose this to refine the depth alignments of step 1 using the coherence matrix and again parallel tempering with the initial values defined by the output of step 1. An advantage of the coherence matrix is that it is a robust feature and insensitive to possible gain differences among contacts.
3. We manually checked for outliers of which none were found in this dataset.

## Layer assignment

Channels were labelled as supragranular, granular and infragranular based on the location of the initial sink-source reversal (as established by the position of the reversal in the aligned grand average) in relation with known anatomy. We consider the position of the sink-source reversal to correspond to the edge of layers 4 and 5 (*Mitzdorf and Singer, 1977*; *Schroeder et al., 1991a*). Specifically, given our intercontact spacing of 150 micrometres and about 500 micrometres width generally used per layer (*van Kerkoerle et al., 2014*; *Maier, 2010b*), channels from this border to 450 micrometre below it were labelled infragranular, channels up to 450 micrometre above as granular, and channels 600 above it and higher as supragranular. Data were averaged within supra- and granular layers or infragranular layers in agreement with the two separable sites of gamma-power synchronization as indicated in the text.

## Definition of the V1-White Matter-V2 borders

The depth probes often collected signals beyond the lower V1 layer 6 border and often reached the deep V2 infragranular layers. When the probes reached deep V2 the RFs shifted abruptly several degrees as expected form V1-V2 retinotopy (*Figure 1—figure supplement 1B*, rightmost plots) (*Gattass et al., 1981*). The white matter situated between the two areas appeared relatively thin, often comprising 1–2 contacts (150-300 microns).

To estimate the lower V1 Layer 6 boundary, we first used spiking RFs to determine the transition. We computed a RF centre distance measure, referenced to L4-L5 border, to determine at which contact the transition to deep V2 occurred. Before the transition, often 1 or 2 contacts did not show spike RFs at all and were thus likely to represent white matter. V1 Layer 6 border was then defined as the contact with the last low RF centre distance (threshold <0.5 deg). In probes with low spiking quality; we used CSD signals (filtered in the gamma range (30–150 Hz) for determining the V1 L6 border.

## Single-session RF and CSD evaluation

For each session and probe, the CSD from full-screen checkerboard flashes (37), the task and RF data were plotted side-by-side. CSDs from flashes and the grating onset were very similar in the initial response (data not shown). The task-data from a single, high-contrast condition was split in an early and a later half to detect any changes in depth over the session and also compared with flash CSDs before and after the task (where available). Recordings were stable in depth according to this measure. The RF mapping was used to detect changes in the size or location of RFs over depth and to ascertain that there were no gradual drifts in RF location, indicative of a probe not inserted fully orthogonal to cortex. In cases were noticeable shifts were observed, the affected deeper channels were removed from the analysis. The final cut-off between deep V1 and white matter/V2 was determined based on the distance from the layer 4/5 reversal (see *Layer assignment*). This border, 450 microns below the 4/5 reversal, was typically above the level where RF shifts were observed, leading to removal of further deep channels from the analysis.

## Visual stimulation paradigm

The monkeys were trained to accept head-fixation and were placed in a Faraday-isolated darkened booth at a distance of 57 cm from a computer screen. Stimuli were presented on a Samsung TFT screen (SyncMaster 940bf, 38° x 30° 60 Hz). The screen was calibrated to linearize luminance as function of RGB values. During stimulation and pre-stimulus time the monkey maintained eye position (measured by infra-red camera, Arrington 60 Hz sampling rate) within a square window of $2 \times 2°$. This window was relatively large to allow for noise associated with the camera, recording with a second high-speed high-resolution camera showed that eye position was generally held more stable than the window required. The monkey was rewarded if for keeping gaze within the eye window during the whole trial.

We aimed to manipulate gamma frequency differences between three recorded locations in V1 each separated by ~2–3 mm, corresponding to receptive fields (RF) separated by ~1 degree in visual space. The probes were arranged linearly either perpendicularly or parallel to the lunate sulcus, thus receptive fields were arranged respectively either horizontally or vertically. To manipulate gamma frequency differences, we manipulated local stimulus contrast differences in a large square-wave grating (2 cycles/degree, presented at two opposite phases randomly interleaved). Contrast was varied smoothly between the three locations. The direction of the contrast difference was parallel to the arrangement of RFs and orthogonal to the orientation of the grating. To avoid that the contrast manipulation would attract exogenous and endogenous attention (possibly appearing as an object or object boundary), we manipulated contrast differences in a repeating symmetric pattern over the entire screen. Additionally, the stimulus was isoluminant at all points and was isoluminant with the pre-stimulus grey screen. The contrast at the location of the centre RF, was constant over all conditions. We presented 8 levels of contrast difference and one stimulus where contrast was

the same at all points. The exact contrasts differed slightly between the two monkeys since we used different screens (of the same type) which had somewhat different luminance levels. Contrast levels are given in *Table 1*.

We aimed to align the stimulus so that receptive fields at the three cortical locations would align with the highest, lowest and midpoint of one cycle of the contrast variation. However, RFs did not always fall exactly as we wished and there was often some variability in RFs within each probe. To get the best alignment that we could on a given session, we placed the stimulus such that receptive fields from the upper portion of the central probe fell on the midpoint between the peak and trough of the contrast variation. We then selected a stimulus where the distance between the peak and trough best matched the distance between RFs from the flanking probes. In most cases this lead to a peak-to-trough distance of 2 degrees. In some cases, we used a distance of 1 or of 3 degrees. In some sessions we recorded with only two probes in V1. In those cases, the stimulus was aligned so that the midpoint was midway between the RFs of the two probes.

Most analysis was based on the measured gamma frequency rather than the stimulus contrast and so any mismatch between the stimulus contrast a particular RF received and the contrast we planned to present did not affect our conclusions. Where statistical analysis (see sections below '*Effects of visual contrast and eccentricity on gamma frequency*') was based on stimulus contrast we took the stimulus contrast which was present at the centre of the measured RF of each single electrode contact. For *Figure 1* and *Figure 1—figure supplement 2* the data is shown binned by stimulus contrast values for illustration.

## Effects of visual contrast and eccentricity on gamma frequency

Local stimulus contrast had a significant effect on the V1 gamma frequency (linear regression, M1: $R^2 = 0.38$, n = 1179 M2: $R^2 = 0.27$, n = 1134, both $p<10^{-10}$, slope: ~0.15 Hz/contrast, see *Figure 1—figure supplement 2*) in both monkey M1and M2 confirming previous studies of monkey and human visual cortex (*Hadjipapas et al., 2015*; *Hall et al., 2005*; *Jia et al., 2013a*; *Ray and Maunsell, 2010*; *Roberts et al., 2013*; *Self et al., 2016*). Stimulus contrast lead to a monotonic increase of the frequency, here measured as the mean of the instantaneous gamma frequency (similar results were obtained using the conventional frequency of the power spectral peak). Both LFP and CSD gamma gave the same result. The frequency increase was approximately linear in the range tested, however it might deviate from linearity if the whole contrast range is considered. Further, in comparison to prior studies (*Hadjipapas et al., 2015*; *Jia et al., 2013a*; *Ray and Maunsell, 2010*; *Roberts et al., 2013*), we used here whole-field gratings with local spatially varying contrast. The MUA spike rate also significantly increased with stimulus contrast (linear regression, M1: $R^2 = 0.14$, n = 1179, M2: $R^2 = 0.12$, n = 1134, both $p<10^{-10}$), which has been well established by previous work (*Contreras and Palmer, 2003*; *Sclar et al., 1990*). It suggests that the frequency change is due to a change of network excitation (*Tiesinga and Sejnowski, 2009*; *Traub et al., 1996*). We inserted laminar probes acutely into the visual cortex and the different probes, depending on their arrangement, recorded from cortical location coding for different visual eccentricities. There was also variation across sessions. It has been shown in previous work that the V1 gamma frequency is modulated by eccentricity (*Lima et al., 2010*; *van Pelt and Fries, 2013*). We confirmed these observations. The gamma frequency significantly decreased with visual eccentricity (linear regression, M1: $R^2 = 0.12$, n = 1179, M2: $R^2 = 0.15$, n = 1134, both $p<10^{-10}$). We also observed that the MUA spike rate decreased with visual eccentricity (linear regression, M1: $R^2 = 0.04$, n = 1179, M2: $R^2 = 0.08$, n = 1134, both $p<10^{-10}$) similarly to gamma frequency. Frequency differences (detuning) between all V1 pairs were here a function of both stimulus contrast, being the strongest factor, and visual eccentricity (multiple linear regression, M1: $\Delta$contrast, $R^2 = 0.28$, $\Delta$eccentricity, $R^2 = 0.09$, n = 7245; M2: $\Delta$contrast, $R^2 = 0.25$, $\Delta$eccentricity, $R^2 = 0.11$, n = 7938, all $p<10^{-10}$). We observed that the frequency difference was closely related to MUA spike rate difference among probes (linear regression, M1: $R^2 = 0.53$, n = 7245, M2: $R^2 = 0.36$, n = 7938, both $p<10^{-10}$) indicating that gamma frequency differences (and hence detuning) between locations are related to excitability differences. The

lower excitability in more eccentric locations could reflect network differences or that stimulus, with a spatial frequency of 2 cycles/degree, was better suited to more foveal sites.

## Estimation of instantaneous gamma phase, frequency and amplitude

For quantifying the phase-locking value and the preferred phase difference we relied on the reconstruction of the instantaneous phase (*Picinbono, 1997*). Methods based on the instantaneous phase deal better with non-stationary dynamics (than e.g. spectral coherence), which were present in the gamma-band signals investigated here. The main challenge is to decompose the often complex, multi-component measured LFP/CSD signal, into a well-defined gamma oscillatory component from which the instantaneous phase can be extracted (i.e., after a Hilbert-Transform or directly from a time-frequency representation (TFR),[*Le Van Quyen et al., 2001*]). We used a method based on the singular spectrum decomposition of the signal (SSD, see https:// project.dke.maastrichtuniversity.nl/ssd/) (*Bonizzi et al., 2014*). SSD is a recently proposed method for the decomposition of nonlinear and non-stationary time series (*Bonizzi et al., 2012*; *Bonizzi et al., 2014*) in a completely data-driven manner. The method originates from singular spectrum analysis (SSA), which is a nonparametric spectral estimation method used for analysis and prediction of time series. For a given signal x (t) we applied SSD for each trial separately to extract the gamma oscillatory components ($SSD_\gamma$). Here a short overview is presented. For more information see (*Bonizzi et al., 2014*). The following steps were implemented to retrieve the gamma oscillatory component $SSD_\gamma$ (*Bonizzi et al., 2012*), where each iteration reproduces one component. The iteration stopped when 10 components were extracted or only 1% residual variance remained.

1. The signal x(t) is embedded giving a trajectory matrix X:

$$X = \begin{bmatrix} x_1 & x_2 & \cdots & x_{N-M+1} \\ x_2 & x_3 & \cdots & x_{N-M+2} \\ \vdots & \vdots & \vdots & \vdots \\ x_M & x_{M+1} & \cdots & x_N \end{bmatrix} \tag{2}$$

Particular to the SSD approach, the embedding dimension *M* is automatically estimated in a completely data-driven manner as 1.2*Fs/fmax, with *fmax* being the dominant frequency in the power spectral density (PSD) of *x(n)*, and *Fs* the sampling frequency. The factor 1.2 allows *M* to cover a time span 20% larger than the average period of the wanted component (to account for a variable period).

2. The singular value composition (SVD) of the trajectory matrix *X* is then computed:

$$X = UDV^T, \text{ with } U = (MxM) \text{ and } V = (NxN) \tag{3}$$

3. Out of the *M* principal components of *X*, an approximated version of *X* is obtained by selecting those principal components with a dominant frequency in the range [*f*max - δ*f*; *f*max + δ*f*], where the width of the dominant peak δ*f* is estimated by means of a Gaussian interpolation of the power spectral density of the time series *x(t)*. Then signal is then reconstructed by diagonal averaging. The reconstructed component signal is subtracted from the original signal and a new iteration of steps is started.

The SSD procedure results in a set of components representing rhythmic variation of the signal with different dominant frequencies. We were interested in the component which represented the gamma-band. We therefore selected the component which had the largest fraction of spectral power in gamma frequencies [25 Hz-60Hz] for each single trial. In the large majority of cases, there was a single dominant component representing gamma-band fluctuations in the LFP/CSD signals. To get an estimate of the percentage of outlier SSD trials, we counted trials with an instantaneous frequency variation exceeding 1.5 times the interquartile range from either the 25th or 75 percentile of the distribution. We found that according to this criterion, 1.46% in M1% and 1.25% in M2 of the SSD decomposed trials could be considered as outliers, with high frequency variations indicating that for those trials

the SSD decomposition was likely not optimal. As the percentage of outliers was small, we did not remove them from analysis.

For deriving the instantaneous phase of a SSD component, the Hilbert transform (HT) was applied using the Matlab implementation.

$$SSD\alpha_\gamma = SSD_\gamma + iHT(SSD_\gamma) \qquad (4)$$

where HT(SSD$_\gamma$) is the Hilbert-Transform of the selected SSD gamma component. The HT of a real-valued signal is added as imaginary component to the real-valued signal itself to obtain the analytic signal. SSD$\alpha_\gamma$ is the analytical signal of the SSD$_\gamma$. The instantaneous phase φ can then easily be derived from the analytic signal:

$$\varphi = \mathrm{Arg}(SSD\alpha_\gamma) \qquad (5)$$

Arg is the argument of the complex value SSDa$_\gamma$. The instantaneous frequency (IF) can be determined as the derivative of the instantaneous phase. The phases need to be unwrapped before applying the derivative. However, the IF might exhibit strong outliers if the signal is noisy. We used therefore a Savitzky-golay filter (**Schafer, 2011**) to smooth the phase trajectory (and hence the IF) using a polynomial fitting approach (kernel = 31 ms).

The HT is a standard approach for reconstruction of the instantaneous phase, however a problem of HT is its sensitivity to low SNR. We therefore used another approach for estimating instantaneous phase that is more robust against noise, but remains valid (**Lowet et al., 2016**). We approximated the instantaneous phase by using the time-frequency representation (TFR) of the signals using Morlet wavelets (**Le Van Quyen et al., 2001**). This approach was used mainly for estimating phase-locking strength (PLV). Morlet wavelet approach was defined as follows:

$$W_{SSD_\gamma}(t,\omega) = \int_{-\infty}^{+\infty} SSD_\gamma\, \Psi_{t,\omega}^*(\mathrm{x})\mathrm{dx} \qquad (6)$$

where $W_{SSD_\gamma}(t,\omega)$ is the wavelet coefficient of the gamma SSD component and $\Psi_{t,\omega}^*$ is the complex conjugate of the Morlet wavelets, both as a function of time t and frequency $\omega$. Morlet wavelets were defined as:

$$\psi_{(t,\omega)} = \sqrt{\omega}\mathrm{e}^{\mathrm{i}2\pi\omega(\mathrm{x}-\mathrm{t})}\mathrm{e}^{-\frac{(\mathrm{x}-\mathrm{t})^2}{2\omega^2}} \qquad (7)$$

Where σ defines the width of the wavelet which also defines the number of cycles (nc = 6fσ). Here we used 6 cycles. The argument of the complex wavelet coefficients gives the instantaneous phase for each frequency-time point:

$$\varphi_W(t,\omega) = \mathrm{Arg}(W_{SSD_\gamma}(t,\omega)) \qquad (8)$$

## Estimation of phase-locking strength and mean phase difference

The mean phase difference was defined as the mean circular phase difference between two oscillations (averaged in the complex domain), where θ = φ$_1$- φ$_2$:

$$\bar{\theta} = \mathrm{Arg}\left(\frac{1}{T}\sum_{t=1}^{T} e^{i\theta(t)}\right) \qquad (9)$$

with a range of [-π, π]. Arg is the argument function and θ is the instantaneous phase difference derived from the Hilbert transform. For estimating phase-locking we computed the phase-locking value (PLV, [**Lachaux et al., 1999**]) based on the instantaneous phase derived from the wavelet TFR. The PLV was computed by averaging the complex values with unit amplitude:

$$PLV = |\frac{1}{T}\sum_{t=1}^{T} e^{i\theta(t)}| \tag{10}$$

$$\theta(t) = \varphi_{W1}(t,\omega1) - \varphi_{W2}(t,\omega2) \tag{11}$$

where T is total number of time points (trials were concatenated). The frequencies $\omega_1$ and $\omega_2$ were chosen based on the frequency of the gamma spectral power peaks of the respective contacts. The PLV ranges from 1, corresponding to full phase consistency, to 0, corresponding to fully random. Importantly, the PLV measure allows that oscillations have different frequencies (a form of cross-frequency coupling measure, [*Lowet et al., 2016*]). Both, HT-PLV or wavelet TFR-PLV gave similar results. However, the wavelet TFR-PLV is more robust for SNR changes over different probes or sessions and we chose this as our preferred method for the main analysis. The main results were also not dependent on applying SSD and similar results could be obtained by combining filtering and HT or wavelet TFR on raw signals. For the MUA signals analysed (see below) we used wavelet TFR-PLV on the raw MUA signals.

## Correction for CSD-induced phase shifts

When applying CSD on a laminar probe the resultant signal from a given contact will likely show a constant (artificial) phase shift relative to the phase of the original LFP. This is because the CSD computes the difference among nearby LFP contacts which can change the polarity. For statistical analysis on single contact level these shifts are not problematic (as they are constant for a given contact pair) nor for the directionality measures, but it would give a scrambled picture for the Arnold tongue mapping, where all contact pairs are needed for analysis. To reduce the effect of the phase shifts, we normalized the phase-differences for each given contact pair to the condition having the smallest frequency difference (this corresponds to a parallel translation). Hence, for CSD the phase-difference is by definition 0 at frequency difference (detuning) zero. This was done because gamma oscillations had zero phase difference at zero frequency difference shown by 1) LFP-LFP analysis 2) confirmed by MUA-MUA analysis. An alternative correction of the CSD phase difference using the estimated time-lags from the PSI gave similar results.

## SNR estimation and SNR-correction

For experimental data it is important to consider (external) measurement noise. Measurement noise is noise that adds to the biological signal and is completely unrelated to the underlying dynamics. The amount of measurement noise is often expressed as the signal-to-noise ratio (SNR). Despite the fact that the SNR from invasive LFP or MUA measurements is higher than non-invasive EEG/MEG measures, the SNR is still a limiting factor and needs to be considered for a better interpretation of the data. At low SNR, the PLV is largely underestimated. For example, a SNR of 3 can reduce the PLV more than half. Further, it also important for separating effects of true gamma amplitude from effects by SNR. A further important motivation for considering SNR correction was to be able to compare experimental PLV to the analytical predictions from the coupled oscillator equations which are SNR free.

In the data the exact amount of biological signal and external noise is unclear and needs to be approximated. We approximated the gamma-band SNR by using the fact that most of the gamma power is induced by stimulation. We therefore compared gamma power during stimulation to gamma power during baseline period. The power spectra in the baseline period looked similar to 1/f indicating that the approximation is plausible. The gamma SNR was defined as follows:

$$SNR(\omega) = \frac{(Pow_{Stim}(\omega) - Pow_{Base}(\omega))}{Pow_{Base}(\omega)} \tag{12}$$

To obtain PLV values that are relatively SNR independent, we simulated artificial oscillatory synchronization data using phase-oscillator equations (*Lowet et al., 2016*) for different SNR

levels. We applied the exact same PLV estimation procedure as used for experimental data and quantified how SNR level does change the PLV estimate. The PLV estimates were compared to analytical derived expected PLV by solving the phase-oscillator equations. From these analyses we derived a SNR inverse function which gives a correction factor for the PLV measured at a particular SNR.

In addition, we performed the same procedure for the estimation of the interaction strength ε in experimental data which is also sensitive to SNR. At low SNR, the interaction strength ε is underestimated. Also here we computed a correction factor based on simulated data with different level of SNR.

## Theory of weakly coupled oscillators

Detailed reviews and mathematical descriptions of the theory, also its extensions and limitations, can be found in a number of publications (*Ermentrout and Kleinfeld, 2001*; *Hoppensteadt and Izhikevich, 1998*; *Kopell and Ermentrout, 2002*; *Kuramoto, 1991*; *Pikovsky et al., 2002*; *Winfree, 1967*). According to the theory of weakly coupled oscillators, the phase evolution of two given cortical V1 locations is reduced to:

$$\dot{\varphi}_1 = \omega_1 + \varepsilon_{21} \mathrm{H}_{21}(\varphi_1 - \varphi_2) + \eta_1 \tag{13}$$

$$\dot{\varphi}_2 = \omega_2 + \varepsilon_{12} \mathrm{H}_{12}(\varphi_2 - \varphi_1) + \eta_2 \tag{14}$$

where $\varphi_{1,2}$ is the phase, $\dot{\varphi}_{1,2}$ its temporal derivative, $\omega_{1,2}$ is the preferred frequency, $\varepsilon_{12}$ and $\varepsilon_{21}$ are the interaction strengths, $\mathrm{H}_{12}$ and $\mathrm{H}_{21}$ are the single PRCs and $\eta_{1,2}$ is a phase-noise term with $\eta_{1,2} \sim N(0, \sigma^2)$ N being the normal distribution. The two equations, as given in the main text **Equation 1**, can be further simplified to:

$$\dot{\theta} = \Delta\omega + \varepsilon G(\theta) + \eta \tag{15}$$

where $\theta = \varphi_1 - \varphi_2$ is the phase difference, $\Delta\omega = \omega_1 - \omega_2$ the detuning, $\varepsilon G(\theta) = \varepsilon_{21} H_{21}(\theta) - \varepsilon_{12} H_{12}(-\theta)$ the combined interaction term with ε being the interaction strength and G(θ) the mutual PRC (odd-parts) and η= η₁ −η₂ the phase noise with η ~N(0, $\sqrt{2\sigma^2}$).

## Analytical derivation of phase-locking and mean phase difference

**Equation 15** is a stochastic differential equation (Langevin equation) and was solved as described in (*Pikovsky et al., 2002*). **Equation 15** can be rewritten in the form of a Fokker-Planck equation that has been developed to give an analytical solution for the evolution of a probability distribution P of a particle influenced by a drag force (first term on the right side of the equation) and a random Gaussian noise process (second term). The drag force is here the combined systematic force of detuning Δω and the interaction function εG(θ):

$$\frac{\partial P}{\partial t} = \frac{\partial[(\Delta\omega + \varepsilon G(\theta))P]}{\partial \theta} + \sigma^2 \frac{\partial^2 P}{\partial \theta^2} \tag{16}$$

The stationary (time-independent) solution $\bar{P}(\theta)$ of the Fokker-Planck equation which is:

$$\bar{P}(\theta) = \frac{1}{C} \int\limits_{\theta}^{\theta+2\pi} \exp\left(\frac{V(\theta') - V(\theta)}{\sigma^2}\right) d\theta' \tag{17}$$

$$V(\theta) = \Delta\omega\theta + \varepsilon \int\limits_{0}^{\theta} G(x)dx \tag{18}$$

$$\bar{P}(\theta) = \frac{1}{C} \int_{\theta}^{\theta+2\pi} \exp\left(\frac{V(\theta') - V(\theta)}{\sigma^2}\right) d\theta' \qquad (19)$$

where C is a normalization constant defined by $\int_{0}^{2\pi} \bar{P}(\theta)d\theta = 1$. $V(\theta)$ represents the influence of systematic force as a function of phase difference. $\bar{P}(\theta)$ is the phase difference probability distribution and describes how likely a particular phase difference is to occur. A uniform distribution means that every phase difference is equally likely and the oscillator are hence asynchronous. If the distribution approximates a delta distribution (meaning only one phase difference has non-zero probability), then the oscillators are completely synchronized. All other distributions in between signify intermittent (partial) synchronization (also called cycle slipping or phase walk-through, [*Izhikevich, 2007*; *Pikovsky et al., 2002*]). To quantifying the narrowness of the distribution, we use the phase-locking value (the mean resultant vector length, [*Lachaux et al., 1999*]) defined here as:

$$PLV = \left| \int_{0}^{2\pi} \bar{P}(\theta) \, e^{i\theta} d\theta \right| \qquad (20)$$

Further, we were also interested in the mean phase difference, also described as the preferred phase difference, defined here as:

$$\bar{\theta} = arg\left( \int_{0}^{2\pi} \bar{P}(\theta) \, e^{i\theta} d\theta \right) \qquad (21)$$

A phase difference between oscillators in neural networks implies spike timing differences. It has been shown that spike-timing is an important characteristic in addition to spike synchrony (*Dan and Poo, 2004*; *Heitmann et al., 2013*; *London and Häusser, 2005*; *Markram et al., 2012*; *Masquelier et al., 2009*; *Tiesinga et al., 2008*).

## Biophysical modelling of coupled gamma-generating neural networks

To demonstrate that the results from the phase-oscillator equations are generalizable to more biophysically realistic neuronal network oscillations (see *Figure 3—figure supplement 1*), we simulated two coupled excitatory-inhibitory spiking neural networks generating pyramidal-interneuron gamma (PING, [*Tiesinga and Sejnowski, 2009*]) oscillations.

The neural voltage dynamics v were of the Izhikevich-type (*Izhikevich, 2003*) and defined as follows:

$$\frac{dv}{dt} = 0.04v^2 + 5v + 140 - u + I \qquad (22)$$

$$\frac{du}{dt} = a(bv - u) \qquad (23)$$

$$if \ v \geq 30mV, \ then \begin{cases} v \leftarrow c \\ u \leftarrow u + d \end{cases}$$

The coupled differential equations were numerically solved using the Euler method (1 ms step size). The networks were both composed of two types of neurons: 200 regular spiking neurons RS (a = 0.02, b = 0.2, c=-65mV, d = 8) and 50 fast-spiking interneuron FS (a = 0.1, b = 0.2, c=-65mV, d = 2). RS were excitatory neurons and FS inhibitory neurons (ratio 4:1). The neural networks were all-to-all synaptically connected. Synapses were modelled as exponential decaying functions, reset to 1 after the presynaptic neurons fired. Synaptic connection values

had a maximum synaptic connection strength (max syn). The synaptic strengths were chosen from a random uniform distribution defined between the 0 and the maximal connection strength.

Within a network, RS neurons projected excitatory synaptic AMPA (decay constant = 2 ms) connections onto FS neuron (max syn = 0.45) and among themselves (max syn = 0.05). FS neurons projected synaptic GABA-A (decay constant = 8 ms) connections onto RS neurons (max syn = −0.35) and among themselves (max syn = −0.2). For cross-connections between the networks, we included RS→FS connections (E→I, max syn(default)=0.015) and RS→RS connections (E→E, max syn(default)=0.007)1450011. We did not include inter-network FS→FS or FS→ RS connections to reflect that V1 horizontal connectivity is dominated by excitatory connections originating from pyramidal cells(*Angelucci and Bullier, 2003*; *Angelucci et al., 2002*; *Bosking et al., 1997*; *Boucsein et al., 2011*; *Stettler et al., 2002*).

The input drive to RS neurons was composed of a fixed input current to each neuron (=10), unique Gaussian input noise for a given neuron (SD ±3) and Gaussian input noise shared among neurons (SD ±1) of the same network. Thus each network received Gaussian input noise to RS neurons with the effect of inducing instantaneous frequency variation in the network over time (similar to intrinsic phase noise in the phase-oscillator model). For FS neurons, each received a fixed input current (=4) and Gaussian input noise (SD ±3). FS neurons received further excitatory drive from the RS neurons. For estimating the instantaneous phase, phase difference and frequency of the network oscillation we used a population signal defined as mean membrane voltage of all RS neurons of a given network. We simulated in total n = 697 conditions (17 coupling and 41 detuning conditions) to compare it to analytical predictions.

## Instantaneous frequency modulations by phase-difference

Synchronization counteracts the phase precession by either accelerating or decelerating the precession depending on the form of the phase-response curve (PRC). Hence, phase difference dependent frequency modulations are expected from synchronization theory. To quantify the phase difference dependent frequency modulation in simulation/experimental data, we first computed for each pair of oscillations their instantaneous phases and their derivative (instantaneous frequency). To estimate the modulation, we computed the mean instantaneous frequency for a given instantaneous phase difference. For this, we binned the instantaneous phase difference data into equal bin sizes (bin size = 0.1 rad), and for each bin we estimated the mean instantaneous frequency, here for contact 1.

$$\overline{IF}_1(\theta) = \frac{1}{T_\theta}\sum_{t_\theta=1}^{T_\theta} IF(t_\theta) \tag{24}$$

where IF the instantaneous frequency, $T_\theta$ is the maximal number of time points having phase difference θ, $t_\theta$ are individual time points with phase difference θ.

$$\Delta\overline{IF}(\theta) = \overline{IF}_1(\theta) - \overline{IF}_2(\theta) \tag{25}$$

## Estimation of detuning value $\Delta\varpi$

The intrinsic frequency, the frequency an oscillator would have without interactions with other oscillators, could not be directly measured experimentally. The simple mean (emergent) frequency difference between oscillations will change as a function of synchronization. The stronger the synchronization, the closer the (emergent) frequency difference will be become; up to the point they are complete synchronized (common frequency).

Yet, the intrinsic frequency can be approximated from the phase difference dependent instantaneous frequency fluctuations. If there are no interactions among oscillators, the measured frequency is equal to the intrinsic frequency. However, if the oscillators synchronize, the instantaneous frequency will fluctuate as a function of the phase difference. At the preferred phase difference, the IF difference between oscillators is minimal, whereas at the anti-preferred phase it is maximal. Importantly, if both the interaction strength and the PRC

are similar for both oscillators, then the mean of $\Delta\overline{IF}(\theta)$ will be equal to the detuning. Hence to derive the detuning, we first assumed that the interaction functions between oscillations were symmetric, which seems plausible, considering the isotropic horizontal connectivity properties in V1 (**Stettler et al., 2002**). The detuning value was then defined as follows:

$$\Delta\omega = \frac{1}{N}\sum_{n=1}^{N}\Delta\overline{IF}(n) \tag{26}$$

assuming that (1) $\varepsilon_{12} \approx \varepsilon_{21}$ and (2) $H_{12} \approx H_{21}$. The validity of the approach was tested using phase-oscillator model as well as the coupled PING network model. In the former, the true detuning was a given parameter and in the latter the detuning could be measured by decoupling the two PING networks. Both modelling types showed that the detuning could be robustly retrieved, if interaction strengths were approximately symmetric. The **Equation 26** can be adapted to deal with cases of interaction asymmetry ($\varepsilon_{12} \neq \varepsilon_{21}$) using the individual interaction strengths $\varepsilon_{12}$ and $\varepsilon_{21}$ for a weighted averaging. In our experimental V1 data we observed covariation and similarity of individual interaction strengths $\varepsilon_{12}$ and $\varepsilon_{21}$ that is in line with the isotropic connectivity structure of V1. We observed systematic deviations from symmetry in cases when the PING networks or V1 contacts had large amplitude differences. Oscillation amplitude can influence the interaction strength $\varepsilon$, because a network that can send a larger amount and more synchronized spikes to another network will have a stronger influence(**Fries, 2015**; **Tiesinga et al., 2004**; **Womelsdorf et al., 2007**).

## Estimation of interaction strength ε

A straightforward method is for each contact pair to estimate the modulation amplitude $\varepsilon$ as the (min-max)/2 of the modulation. Even though the method works in many cases, especially for the PING simulation data, it is not very robust against SNR and has a tendency of overestimating the interaction strength as tested with phase-oscillator simulations where the true interaction is known. We therefore used another approach (used in the main analysis) based on the Fourier transform (FFT) of the modulation function:

$$\mathcal{F}(\omega) = |FFT(|\Delta IF(\theta)|)| \tag{27}$$

where $\omega$ is frequency. The first Fourier coefficient is the mean offset of the modulation. Since we observed an approximately sinusoidal shape of the frequency modulation that was periodic over a phase differences of $2\pi$, the amplitude of the modulation is captured in the second Fourier coefficient. We also included the third Fourier coefficient to capture to some extent the asymmetries observed in the modulation shape. The higher Fourier coefficients should mainly represent noise. For estimating modulation strength $\varepsilon$ we summed the second and third Fourier coefficient and subtracted the estimated noise. This noise was assumed to be uniform across all Fourier components. It was therefore estimated as the mean amplitude of the second quadrant of N Fourier coefficients (N defined by number of phase bins of the modulation function).

$$\varepsilon = (\mathcal{F}(2) + \mathcal{F}(3)) - \frac{2}{N}\sum_{j=N/4}^{N/2}\mathcal{F}(j) \tag{28}$$

This gave much more robust estimates for lower SNR data and reduced the tendency of overestimation for lower interaction strengths. Rather it had a weak tendency of underestimation (especially if the modulation shape is more asymmetric). However, for both methods the estimation of interaction strength $\varepsilon$ systematically decreased with lower SNR.

For the experimental data, we scaled the $\varepsilon$ values for the analytical predictions to account for the SNR in macaque V1 data. For this, as described above, we estimated the known interaction strengths of the phase-oscillator data and added different level of external noise to mimic SNR seen in monkey data. This yielded a curve giving the accuracy of the interaction strength estimates as a function of SNR. The inverse of the curve gave us the rescaling values to compensate for SNR. Based on the estimated SNR of monkey M1 and M2 data, we

rescaled the estimate of ε to make it approximately SNR independent. For the main analysis conditions, with larger detuning were chosen to estimate the modulation function of instantaneous frequency by phase difference because data points were more equally distributed over different phases for these conditions. For each contact pair we used all conditions with a detuning value larger than 4 Hz and took the mean of those estimated ε values.

In our experimental data we always had cases of large detuning, owing to our experimental manipulation of contrast difference. In cases where only small detuning values are recorded, we suggest that estimations of ε can be based on the absolute instantaneous frequency differences. This will avoid cancellation of fluctuations around zero which would give severe underestimation of the true underlying interaction strength. The price will be a tendency of overestimation for very low interaction strengths.

## Estimation of the G($\theta$)

Given that the detuning value ($\Delta\omega$) and the interaction strength ($\varepsilon$) were estimated from the mean instantaneous frequency modulation by phase-difference ($\Delta\overline{IF}(\theta)$), then G($\theta$) can be simply estimated by following equation:

$$G(\theta) = \frac{\Delta\overline{IF}(\theta) - \Delta\omega}{\varepsilon}$$

(29)

This approach was tested using the phase-oscillator model (**Figure 5—figure supplement 1**), where G($\theta$) was a known function. Using the described approach any shape of G($\theta$) could be estimated from the simulation data assuming the oscillators were mutually connected with approximately similar interaction strengths (hence being symmetric).

The G($\theta$) describes how the rate of phase precession (equivalent to instantaneous frequency difference) between oscillator is altered as a function of phase-difference, whereas the single PRC H($\theta$) describes how the phase evolution of a single oscillator (=instantaneous frequency) is altered as a function of phase-difference. The H($\theta$) in the PING networks were asymmetric with a stronger positive (advancing) component and a weak negative (delaying) component, hence being more of the so-called PRC Type 1 (**Cannon and Kopell, 2015**; **Schwemmer and Lewis, 2012**). When PING networks were unidirectional coupled, they exhibited asymmetric Arnold tongues due to the asymmetric H($\theta$). Whether these applies to unidirectional coupled brain areas needs to be tested. For the main analysis, the PING networks were mutually connected with the same strength. The resultant G($\theta$) had therefore symmetric negative and positive components.

For making predictions of the PLV or mean phase-difference we assumed that the underlying interaction properties (the shape of the PRCs) did not change over the different conditions. We therefore used one estimation of G($\theta$) function for the whole dataset for each monkey 14500111450011or for all PING simulations. We assumed that the underlying interaction properties (the shape of the PRCs) did not change over the different conditions.

To obtain a G($\theta$) population estimation for a whole dataset we averaged single absolute |G($\theta$)| from all contact pairs that had a sufficient level of detuning (|$\Delta\omega$|>4 Hz). We took the absolute to make G($\theta$) independent of sign and so avoid cancelling each other out during averaging. Taking only conditions where |$\Delta\omega$|>4 Hz was necessary first to assure low synchronization and therefor to have a more uniform phase-difference probability distribution to ease the estimation of the instantaneous frequency difference for all phase-differences. Second, the minima of the G($\theta$) shifted in its mean (preferred) phase difference mainly within the range of −4 Hz to 4 Hz. This would lead to smearing of the obtained population estimation. Restricting to conditions of |$\Delta\omega$|>4 Hz ensured that the individual minima approximately overlapped.

While we used the population G($\theta$) for the main analysis, using a G($\theta$) from a single contact pair led to good prediction values of the whole dataset in many cases. However, there were also individual cases that deviated from the norm. Generally, estimation of G($\theta$) in contact pairs with low detuning that show very high level of synchronization is difficult as the oscillators remain constantly around their preferred phase-relation. This was the case

especially in PING simulation with low phase noise levels. In this case, perturbation techniques might be more appropriate. Further, in cases of strong amplitude differences between contacts, we observed asymmetries at the level of single PRC leading to different G(θ) properties. Further, we observed small to moderate amplitude modulations as a function of phase difference (*Figure 7—figure supplement 4*) that might have affected the shape of G(θ) of contact pairs with strong interaction values. The use of CSD for reducing volume conduction led to additional noise due to artificial phase shifts. Given all these considerations, single G(θ) could be noisy for experimental single contact pairs. The population average G(θ) was a better representation of the interaction properties of V1 horizontal connections.

The use of IF(θ) for estimating single PRC or G(θ) needs careful considerations. However, our work shows how much important information these modulations can contain about the underlying synchronization process. Our approach can be applied to other brain regions and frequency-bands to improve understanding of the underlying synchronization properties.

## Estimation of noise variance $\sigma^2$

The estimation of the phase noise variance σ of the noise process η(t) from data is not trivial due to measurement noise and general complexity of LFP/CSD signals. The phase noise is intrinsic to the oscillatory process (dynamic noise) and relevant for the understanding of the dynamics and therefore distinct to external measurement noise. Phase noise implies variability in the instantaneous frequency of oscillators (see *Equation 13-15)* and the overall variability of instantaneous frequency should scale with the noise variance $\sigma^2$. We approximated the noise variance $\sigma^2$ by determining in the phase-oscillator model what $\sigma^2$ value would produce the same observed instantaneous frequency difference distribution as observed in the PING or experimental V1 data. This approach was more robust than estimating the phase noise variance around the interaction function. It is important to note that the observed frequency variance is not the same as the (intrinsic) variance going into *Equation 13-15*. This is because synchronization also counteracts the intrinsic variability. The procedure involved two main steps:

1. Estimate the (population average) standard deviation of the observed instantaneous frequency difference distribution of SSD gamma.
2. Using phase-oscillator equations find the value for σ that can reproduce the observed standard deviation of the observed instantaneous frequency difference distribution giving the observed signal-noise-level.

## Analytical predictions for PING and experimental V1 gamma

Using the *Equations 17 and19* and the estimated G(θ) and noise variance σ from the data we could make predictions for any value of detuning Δω and interaction strength ε. The phase difference probability distribution $\bar{P}(\theta)$ was analytically predicted, from which we quantified the phase-locking strength (PLV, see *Equation 20*) and the mean phase difference (*Equation 21*). The predicted PLV and mean phase difference was compared to the observed PLV and mean phase difference from the data with the same detuning and interaction strength.

For PING networks we modulated the interaction strength ε by changing the inter-network connectivity strength and the detuning Δω by given different excitatory input drive to the two networks. For each simulation we had an estimate of interaction strength ε and detuning Δω.

For experimental V1 gamma, interaction strength ε was modulated by cortical distance and detuning Δω by local contrast and to a weaker extent eccentricity (see above). For each contact pair, we had their interaction strength ε and their detuning Δω.

## Mapping of the Arnold tongue

For PING data, we mapped the data corresponding to their detuning and internetwork synaptic connectivity strength. For data corresponding to particular connectivity strength, we estimated the interaction strength ε and used these values for the rescaling of the y-axis,

because the interaction strength ε was the parameter we wanted to compare to the theoretical model.

For experimental V1, we binned the contact pairs according to their detuning (±bin size = 0.35 Hz, bin steps = 0.2 Hz) and cortical distance (±bin size = 0.6 mm, bin steps = 0.3 mm). This average data binned according to detuning and cortical distance is what is termed population data. For data corresponding to a particular bin, we estimated the interaction strength ε and use these values for the rescaling of the y-axis. This was done to make sure that the interaction strength ε dimension was independent of the detuning Δω dimension, because binning directly using interaction strength ε and detuning had a potential risk of inducing dependencies between dimensions (e.g., due to SNR fluctuation) as both were based on estimation of ΔIF(θ).

## Evaluation of prediction accuracy of analytical model

We estimated the accuracy of the model predictions using the coefficient of determination, $R^2$, for phase-locking strength (PLV) and the mean phase difference. Notice that here we evaluate the model accuracy without optimizing the parameters to enhance fitting.

$$R^2 = 1 - \frac{SS_{res}}{SS_{Tot}}$$
(30)

$SS_{res}$ is sum of square of the prediction error, the residuals of the difference between observed data and the predicted data, and the $SS_{Tot}$ is sum of square of the demeaned observed data.

For the PING networks we observed that for both PLV ($R^2$ = 0.93, n = 697) and mean phase difference ($R^2$ = 0.94, n = 697) the model predictions explained a large significant part of the variance.

For experimental V1 gamma data we observed that the model predictions captured also a large significant part of the PLV (population level: M1:$R^2$ = 0.88, n = 638, M2: $R^2$ = 0.9, n = 638; level of single contact pairs: M1: $R^2$ = 0.18, n = 7245, M2: $R^2$ = 0.32, n = 7938) and mean phase difference variance in both monkeys (population level: M1: $R^2$ = 0.94, n = 638, M2: $R^2$ = 0.88, n = 638; level of single contact pairs: M1: $R^2$ = 0.56, n = 7245, M2: $R^2$ = 0.27, n = 7938). The population-level data represent the binned and averaged single-contact pairs and conditions according to detuning and cortical distance (see section *Mapping of the Arnold tongue*).

## Analysis of L2-L4 and L5-L6 gamma-band synchronization

For the main analysis of synchronization, we limited the analysis to data recorded from L2-L4 representing most the gamma power in V1 (*Buffalo et al., 2011*; *van Kerkoerle et al., 2014*; *Maier, 2010b*; *Roberts et al., 2013*; *Xing et al., 2009*). The lowest gamma power was observed around the L4-L5 border. We observed a second gamma peak around L5-L6 (*van Kerkoerle et al., 2014*; *Xing et al., 2012*) and gamma power going into deep V2. To distinguish L6 from deep V2 we used marked receptive fields shifts (as described above) as indicator for the transition from V1 to V2.

We did the exact same analysis for quantifying synchronization between pairs of L5-6 gamma as used for L2-4 gamma (*Figure 7C*). We could confirm the observation of an Arnold tongue in terms of PLV and mean phase difference also for the deep gamma showing that the observed synchronization properties can be generalized over different laminar compartments. We propose that calculating the PRC and Arnold tongue between various cortical locations would be a fruitful way to understand the connectivity between brain networks.

## MUA-CSD and MUA-MUA analysis

We also computed gamma PLV and the mean phase difference using multi-unit activity (MUA) by computing both CSD-MUA locking and MUA-MUA locking (*Figure 7B* and *Figure 7— figure supplement 2*). The MUA represent a local population spike rate signal and it is thought to reflect more the 'output' of the network, whereas LFP/CSD represent the synaptic

input of the network (**Buzsáki and Schomburg, 2015**; **Buzsáki et al., 2012**). Further, in the main analysis we estimated the synchronization gamma behaviour by using current-source density (CSD) signals derived from our V1 16-contact laminar probes. The important advantage of CSD compared to the local field potential (LFP) is the strong reduction of volume conduction which would substantially bias the PLV as well as the mean phase difference the closer the laminar probes are. The local second spatial derivation of nearby contact on the laminar probes for deriving CSD (**Einevoll et al., 2013**; **Vaknin et al., 1988**) reduced the effect of far electrical fields. However, application of CSD can likely not completely eliminate the influence of volume conduction of very near probes. Therefore, we used the more local MUA signal to test whether we can confirm a similar gamma synchronization behaviour as observed with CSD. A disadvantage of MUA signal in our recording data was its much lower SNR than LFP/CSD signals. We analysed the aggregate MUA signals of all L2-L4 contacts of a single laminar probes, converted the spikes into spike densities smoothed using a Gaussian filter (σ = 4 ms). In **Figure 7—figure supplement 2** the results of the MUA-CSD and MUA-MUA analysis are illustrated showing that the Arnold tongue in terms of PLV and mean phase-difference could be observed in MUA-CSD as well MUA-MUA signals. The results show that similar gamma synchronization behaviour in V1 can be observed at the level of CSD, representing mainly synaptic inputs, and spiking data, representing neural output. It also shows that volume conduction, already minimized for CSD signals, cannot be an influential determinant.

## Instantaneous amplitude modulation by phase difference

We also investigated whether instantaneous gamma amplitude changed as a function of phase difference (**Figure 7—figure supplement 4**). In both, the PING networks as well as V1 gamma oscillations, we observed small to moderate amplitude modulation (up to ~15% modulation from mean amplitude). The modulations observed in the PING model looked strikingly similar to the V1 gamma amplitude modulations (compare **Figure 7—figure supplement 4A/B** with **Figure 7—figure supplement 4C**). These modulations are not expected from the weakly coupled oscillator theory, but as mentioned above (section *Estimation of detuning value $\Delta\omega$*), oscillation amplitude can influence the interaction strength ε as more synchronized spikes are more effective in influencing receiving neurons. This might affect synchronization behaviour in terms of phase-locking and mean phase difference, especially if amplitude modulations become more substantial. Future work should explicitly address the effect of amplitude on synchronization (**Aronson et al., 1990**).1450011

## Granger causal analysis

We investigated Granger causal interactions (**Granger, 1969**) by fitting a full multivariate autoregressive model.

$$X_i(t) = \sum_{n=1}^{N}\sum_{k=1}^{K} \beta_{ink} X_n(t-k\Delta t) + \alpha_i + \eta_{i1}(t) \tag{31}$$

Where a the value of an each discrete time series $X_i$ at time bin t is predicted based on a linear combination of the past $K$ time intervals (i.e. the maximum lag of the model) of itself ($n=i$) and the past of the other simultaneously recorded signals ($n \neq i$) as well as a constant offset term ($\alpha_i$). Finally the residuals are captured in a noise term ($\epsilon_{i1}$). The coefficients ($\alpha_i, \beta_{ink}$) were fitted to the data by way of least squares regression.

Besides the full model in **Equation 31**, a second (restricted) model was fitted, where the past of signal $j$ was left out.

$$X_i(t) = \sum_{n=1,n\neq j}^{N}\sum_{k=1}^{K} \beta_{ink} X_n(t-k\Delta t) + \alpha_i + \eta_{i2}(t) \tag{32}$$

Signal $X_j$ is said to have a Granger causal interaction with $X_i$ if the full model fits statistically better than the restricted model, since this suggests that the past of signal $X_j$ affects the

present of signal $X_i$. The two fitting operations gave two residual noise signals, $\eta_{i1}(t)$ and $\eta_{i2}(t)$. If either of these models fitted the data significantly better, than the variance of its respective residual time series should be significantly lower than the other.

For all our analyses of Granger causal interactions of the SSD gamma components we used a maximum lag (K) of 10 time bins, that is, 10 ms. **Figure 7D** show the mean differences between the F-values calculated in a feedforward or feedback direction (i.e. $F_{j\rightarrow i} - F_{i\rightarrow j}$) for signals i and j as a function of detuning and coupling strength (i.e. cortical distance).

