## [Decision Letter]

Thank you for submitting your article "A quantitative theory of γ synchronization in macaque V1" for consideration by *eLife*. Your article has been reviewed by three peer reviewers, one of whom, Charles E Schroeder (Reviewer #1), is a member of our Board of Reviewing Editors, and the evaluation has been overseen by David Van Essen as the Senior Editor. The following individual involved in review of your submission has agreed to reveal their identity: Bard Ermentrout (Reviewer #2).

The reviewers have discussed the reviews with one another and the Reviewing Editor has drafted this decision to help you prepare a revised submission. We hope you will be able to submit the revised version within two months.

Summary:

This paper addressed the widely held view that, a stable, shared frequency over time is considered a condition for functional neural synchronization in the gamma range. The findings show that actually, the opposite is true. Instantaneous frequency modulations are critical for adjusting phase relations between different neural ensembles, and thus for synchronization of their oscillatory cycles; this is a key requirement for communication through oscillatory phase synchrony or coherence. The authors show that for local neuron populations oscillating at different gamma frequencies, if similar enough, these frequencies continually attracted and repulsed each other, which enabled preferred phase relations to be maintained in periods of minimized frequency difference. This dynamical interaction is predicted by the theory of weakly coupled oscillators, a fundamental mathematical principle that likely applies widely across brain regions and oscillation frequencies. The paper has a great many strengths, particularly, the outstanding effort at making this topic accessible to the general systems neuroscience audience. However, it could use improvement in several areas.

Essential revisions:

Along with the specific issues noted by the reviewers, there are a few broad concerns that the authors should address: Under reviewer 1 – 1) The issue of a negative control; 2) Negative detuning; and 3) Volume conduction. Reviewer 2's broad concerns were: 1) The authors seem to confuse the PRC and the coupling function throughout the paper; 2) Since they measure Phi, they can now compute the actual interaction function at both sites and this is even more informative than is G; 3) the way that G(φ) is plotted looks very sinusoidal possibly a consequence of the way the phase is extracted; and 4) Since the frequency varies rapidly between the two sites, it may covary between the two populations. Reviewer 3 also wanted to see a control analysis where the full analysis pipeline is applied to non-simultaneously recorded (and so un-synchronised) trials (either by shuffling trials or combining non-synchronously recorded probes), to ensure that the SSD step is not picking up some common effects (e.g. stimulus onset) that are inducing the delta-phase-delta-if relationship. Reviewer 3 also asked that the authors give some heuristic guidance for experimental designs to which these techniques can be applied. Presumably they need to systematically modulate both connection strength and frequency difference, as done here? Finally, reviewer 3 requested that the authors deposit the data in a suitable repository such as Dryad to reduce their future administrative burden and to ensure long-term stable access. https://opennessinitiative.org/

Reviewer #1:

This paper addressed the widely held view that, a stable, shared frequency over time is considered a condition for functional neural synchronization in the gamma range. The findings show that actually, the opposite is true. Instantaneous frequency modulations are critical for adjusting phase relations between different neural ensembles, and thus for synchronization of their oscillatory cycles; this is a key requirement for communication through oscillatory phase synchrony or coherence, and idea championed by Varela and von Stein and others, and more recently by Fries. The authors address a specific paradox: cell ensembles oscillating at different gamma range frequencies can somehow communicate. Recording with laminar multielectrodes placed in areas with different parafoveal receptive fields in monkey visual area V1, they show that for local populations running at different gamma frequencies, if similar enough, these frequencies continually attracted and repulsed each other, which enabled preferred phase relations to be maintained in periods of minimized frequency difference. This dynamical interaction is predicted by a physics theory – that of weakly coupled oscillators. This fundamental mathematical principle of synchronization through instantaneous frequency modulations likely applied widely across brain regions and oscillation frequencies. This paper has a great many strengths, particularly, the outstanding effort at making this topic accessible to the general systems neuroscience audience. We also really like the bicyclist analogy, which is also very Dutch. We do not see any serious flaws, however, the paper could use improvement in several areas. Comments on these are below.

First, a few broad issues.

1) Negative control. It would be nice if authors can show a case that can be considered reasonably as a negative control. The reason is that all figures show the cases of positive effects consistent with the theory. Evidence would be stronger if there are no effects like flat G(*θ*) when there is no interaction. That would also help in arguing the functional role of gamma synchronization later in Discussion. Figure 3—figure supplement 1 shows a negative dependence of interaction strength on cortical distance, which may suggest what a negative control case could be. However, the lowest values are still above 1 Hz. A more desirable control would be to use elevated gamma activity recorded simultaneously in 2 unrelated cortices. One control possibility might be recordings separated by much greater cortical distances. Assuming the authors do not have such cases, recordings during no stimulation or when animals are asleep may work. Even when control conditions are met, interaction strength still might not be zero. It is possible that there may always be some residual amount, as some interactions could remain even for null conditions. However, one might hope to find that interaction strength under control conditions is significantly smaller than values during test conditions. Also, even with a residual interaction strength, preferred phase could deviate from those during test conditions.

2) Negative detuning. Readers might surmise that for a give pair of sites, a pair of δ IFs, one positive and one negative values of same magnitudes, can be obtained, like IF1-IF2 or IF2-IF1, so that there could be negative detuning as shown in Figure 5. Also, 2 functions of G (*θ*) can be obtained, and those can be mirror images of each other across the line ΔIF = 0. Is that correct? It may help the authors' case if an example of negative IF is shown in Supplementary Information. If the answer is yes, then I could understand the left-right symmetry of Arnold tongue (as in Figure 5). However, then why are the cases in Figure 7 not left-right symmetric for physiological data?

3) Results section: "strongly reduce the influence of volume conduction by calculating current-source density (CSD)" Though the authors' have innovated extensively, the basic laminar electrode approach to data collection in awake monkeys is not new. There are quite a number of studies by a number of groups that used this method to analyze the areal and cellular generators of specific VEP, AEP and EEG components in monkeys between ~1990 and the present. It would be helpful to the broad readership of *eLife* if the present study were put in the context of that literature. Additionally, effects of volume conduction have been subject of debate over the last several years, and it would be helpful to readers to point to that. This all highlights the particular strength of the CSD approach relative to more typical LFP recordings. Note also, that there is a microscale phenomenon in volume conduction that might be relevant to this paper (Kajikawa J Neurophysiol. 2015).

Now, some more specific concerns

Introduction section: It would be helpful if the authors would be more descriptive introducing concepts of detuning and local stimulus contrast.

Results, first paragraph: Gamma power in the deepest layer is remarkable, but is analyzed rather later as an afterthought. This might be brought forward in the analysis and done in parallel with the other layer compartments.

Figure 1 Supposedly the shading represents SEM? Please state.

Results, second paragraph: It would help to state in the manuscript how the specific components were chosen. According to the Supplementary material, they chose the components with the largest fraction of spectral power in gamma frequencies (25-60Hz). They also state that in most cases there was one clear component representing gamma-band fluctuations. What was the percentage of cases where there was no clear component? What happens in cases where there was no one clear component?

Figure 2: Shouldn't mean IF modulation be called mean ΔIF modulation? Do the number of dots in E reflect time samples across 33 trials presented in D? Number of observations used in this analysis should be clearly stated. Also, the relation should be quantified with statistics. Why is this based on so few trials? I understand that monkeys would usually go through several iterations of an experiment. Does this relation hold through different runs?

Results section: "The key to understanding how this dynamic relationship leads to synchronization is that phase relationships associated with lower frequency differences are maintained than phase relationships- associated with higher frequency differences" This is unclear to me. Yet, understanding this link is critical. What is meant by "maintained" here?

Results, paragraph two, penultimate sentence: typo – "are (better) maintained"

Results, third paragraph: – Re Figure 3/Figure 3—figure supplement 1: Tell us the IF for each of the V1 sites before giving us the ΔIF. Also are effects of ΔIF linear x the band?

In the same paragraph: How was the 1Hz modulation amplitude derived?

Figure 3: this description is at odds with the Figure It looks more like 2x that (~3.5 Hz); it only becomes obvious later that the description refers to 1/2 amplitude modulation. The IF modulation in H looks larger than in E (around 4 vs. 3 respectively). This should be clarified. Also, it would be good to quantify if there is any difference in IF modulation between these two cases.

Results: "The chosen examples were representative for the 805 recorded across-probe contact pairs in monkey M1 and 882 pairs in monkey M2." How was this quantified?

Figure 3 readability of figure would improve if frequency difference (or max and min values used to estimate it) were directly marked on the axes. It would also help to clearly indicate N which was used to obtain these results (same for C, F, I). These results are shown qualitatively. Maybe some quantification using permutations will be beneficial.

Results, fourth paragraph: typo "hence speed is analoguesous to"

The cyclist analogy is outstanding (and very Dutch)!

Subsection “The theory of weakly coupled oscillators (TWCO): A framework for cortical gamma synchronization”: This is a point that occurred at several points throughout the paper. The authors mention horizontal connectivity, but in the communication through synchrony idea (aka CTC of Fries), hierarchical connectivity (e.g., between given RF representations in V1 and V2) is really more relevant lateral connectivity between different eccentricities in V1. Yjis point should be addressed in the Introduction and, while it is mentioned, it should be clarified in the Discussion. Through the authors downplay the notion, the present findings do seem to fit with Prof Singer's Binding Hypothesis.

In the same section: The formula includes only the detuning (Δ*ω*) and not the frequency itself. I understand that the authors would like to concentrate on gamma (and they provide clear motivation for that) but it will be good to discuss how this result might be extended to different frequency bands.

Figure 4: Aren't the functions supposed to have the shape of a sine wave? Particularly, Figure 4 does not look like that: downward peak look like they have almost twice the width of upward peaks.

Figure 5 legend: typo "we to we"

Subsection “TWCO predicts synchronization properties of V1 cortical gamma rhythms”: "PLV variations over single contact pairs were substantially captured by the analytical predictions as a function of Δ*ω* and *ε* (model accuracy: M1: R^2^=0.18, n=7245, M2: R^2^= 0.32, n=7938 Figure 6”? It would be helpful to explain why R^2^ 0f 0.18 and 0.32 values are considered substantial.

In the sixth paragraph: Sig s6 is measured data not model result – Should be S5

In the seventh paragraph: Is there any reason for not including noise in the model?

In the same paragraph: "The results show that gamma rhythms with a higher frequency in a pair had the leading phase." This should be unpacked a little. Please state how leading and lagging phases are assigned.

In the tenth paragraph of the same section: "in the (CSD-CSD) gamma band (feedforward-feedback) as a function of detuning and interaction" This should be unpacked a lot. How are feedforward and feedback determined here?

Figure 7: This Figure is not well motivated, it’s not clear why f-f vs s-f for some and not others. Why are B&C monkeys combined and not in A&B? Does the causality give the same directionality between IF1-IF2 and IF2?

Figure 7: Analyses of MUA for Figure 7 needs more explanation. Not clear how gamma phase was defined for spike timing. Did the authors derive/filter spike density functions? And with what kernel/binsize? Also, it is better to shown raw and processed MUA, as shown for CSD in Figure 2.

Figure 7—figure supplement 4: Does "gamma amplitude" mean the absolute amplitude of gamma band after Hilbert, or the amplitude of real signal? Also, are those amplitudes reflecting the sums of gamma in 2 sites? Please clarify. I may be wrong, but every panel in the figure shows full cycle of amplitude with a peak near zero phase difference, that makes me think it is real amplitude. Is it worth plotting gamma amplitude against interaction strength, *ε*?

In those plots, peak position is slightly rightward. Does it relate to phase leading/lagging? Same question again as above: If IF1-IF2 and IF2-IF1 are counted equally, then should the curves be symmetric around Δ-phase = 0?

Supplementary Information – Correction for CSD-induced phase shifts: "we normalized the phase-differences for each given contact pair to the condition having the smallest frequency difference," by doing what? Is this parallel translations of points along axes?

In the final paragraph of the subsection: "in the case of mutually anatomically coupled cortical locations, detuning strongly influences the main direction of information flow" As presented, this is questionable.

While grand G(*θ*) was used to predict PLV, preferred phase difference could differ among pairs according to Figure 6. Were G(*θ*) values shifted along the axis of phase difference to align the downward peak of the curve to the preferred phase difference for each pair?

Subsection “The Arnold tongue and the regulative parameters of γ synchronization”. Can authors show the range of detuning? Were there cases that can show large deviations in detuning that break down interactions?

In the same subsection: while larger detuning values would decrease opportunities for synchronization across distance in V1, when they are tied strongly to stimulus information in a cell's RF, e.g., the Gabor stimuli used in Ray and Maunsell, there are still opportunities for the neurons in V1 (seeing low contrast) to synchronize with ones in V2 with overlapping RF (also seeing low contrast) and the same for V1/V2 interactions across the different contrast regions of the Gabor. This means basically frequency segregation of channels seeing different "pixels."

Supplementary information: Equation 18 in Supplementary Information duplicates Equation 17. I guess the label "18" was meant for the line: "V(*θ*)"?

Summary: Likely many of the concerns are simply a result of our misunderstandings. We really enjoyed reading the paper and must say, it really stimulated our thinking. It should be similarly provocative for many others in the field and will likely prove over time to be a fundamental contribution.

Reviewer #2:

I found this to be a very interesting paper and I was quite excited to see how well the theoretical predictions matched the results of the Arnold tongues. I was surprised that the interaction function was so close to a sine wave (a point that I will address shortly below) I think that the paper needs some revision and it appears that the authors are somewhat confused by some of the mathematical terminology in TWCO and I will address that below as well.

Here are my main comments

1) The authors seem to confuse the PRC and the coupling function throughout the paper. They are not the same thing. In the theory of weak coupling, there are three things:

1) The PRC or adjoint function, Z(t) which is the infinitesimal PRC and is hard to compute with real neurons, but east to compute with models (see for example the tutorial: http://www.math.pitt.edu/~bard/bardware/meth3/node15.html#SECTION00052000000000000000on weak coupling of a simple voltage gated model;

2) The interaction function, H(φ) which is defined as:

H(*θ*_2_-*θ*_1_) = (1/T) ∫0TZ(s+θ1). C_12_(u(s+*θ*_1_),u(s+*θ*_2_)) ds

That is, it is the convolution of the PRC (Z) with the coupling between the oscillators, here denoted by u(t) with 1 = postsynaptic and 2 = presynaptic

3) The odd part of the interaction function which in this paper is G(*θ*) arises as follows

*θ*_1_' = *ω_1_* + ε_12_ H(*θ*_2_ – *θ*_1_) Equation 1

*θ*_2_' = *ω_2_* + ε_21_ H(*θ*_1_ – *θ_2_*)

If you let φ = *θ*_2_ – *θ*_1_, the phase difference and if ε_12_=ε_21_, then

φ' = *Δω* + εG(φ) Equation 2

where G(φ) = H(-φ) – H(φ) is proportional to the odd part of H(φ)

Equation 2 is what the authors measure; it is not the PRC. I just want to make that clear.

2) Since they measure φ, they can now compute the actual interaction function at both sites and this is even more informative than is G. By measuring *θ*_1_' and *θ*_2_', and plotting this against φ, equation 1 shows that

*θ*_1_' = *ω_1_* + ε_12_ H(φ)

*θ*_2_' = *ω_2_* + ε_21_ H(-φ)

thus, in addition to G, they will be able to measure *ω_1_* + ε_12_ H(φ) and ε_21_ H(-φ). This will give them more information about the direction of coupling and also will give them the full value of H. Indeed, there is nothing here that requires the two H's to even be the same. Thus, I would suggest plotting *θ*_1_', *θ*_2_' vs φ as well as φ' vs φ.

3) G(φ) is plotted looks very sinusoidal. Is this a consequence of the way the phase is extracted? I suggest taking a model pair of oscillators, such as Hodgkin huxley and coupling them with synapse along with noise and heterogeneity in coupling and in frequency. Then use the method that is used for the data to extract G(φ) and H(φ) and then compare this to the actual determistic values found by setting the noise to zero and computing H and G as in the above. This would assure me that the method of using Hilbert transforms isn’t somehow removing all the Fourier modes beyond the lowest.

4) Since the frequency varies rapidly between the two sites, does this variability covary between the two populations? I ask this because one can take two uncoupled oscillators and apply a broadband correlated signal to them and they will produce a very nice peak in their phase-difference histogram. For example, see Nakao, Arai, Kuramoto PRE 2005, or other papers by Arai & Nakao, or for a neuroscience version, Zhou, Burton, et al. 2013 Frontiers in Comp Neuro.

Reviewer #3:

This study applies the well-known theoretical model of weakly coupled oscillators to experimental data from macaque V1. This is achieved through an elegant experimental design, which obtains simultaneous paired recordings over a large range of coupling strengths and frequency differences, to allow fitting of the oscillator model. While the idea is simple, the work is technically impressive and has broad important applications for interpretation and analysis of phase relations between brain areas. The writing is extremely clear; the authors have done an excellent job of conveying a technically complex piece of work in a way that should be accessible to a broad audience.

The results are impressive, but I would like to see a control analysis where the full analysis pipeline is applied to non-simultaneously recorded (and so un-synchronised) trials (either by shuffling trials or combining non-synchronously recorded probes). This would ensure that the SSD step is not picking up some common effects (e.g. stimulus onset) that is inducing the delta-phase-delta-if relationship e.g. in Figure 2. (I don't think full permutation inference is necessary for all results, but just some confirmation that indeed the actual experimental data pipeline does produce flat delta-phase / delta-if curves)

The authors propose that the method of fitting TWCO model to data and associated analyses such as Arnold tongue plots might be applicable to other brain regions or frequency band relationships. It would be nice if they could give some heuristic guidance for experimental designs to which these techniques can be applied. Presumably they need to systematically modulate both connection strength and frequency difference, as done here?

The statistical analysis and Figure 6 show the overall average over all electrode pairs and stimulus conditions. Presumably the experimental contrast already induces a range of PLVs for each specific recording pair, so perhaps they could be evaluated statistically as well (e.g. scatter with pair PLV R2 as a function of distance between the pairs, or something along similar lines).

The authors indicate that data and code are available on request. I would strongly urge them then to deposit the data in a suitable repository such as Dryad to reduce their future administrative burden and to ensure long term stable access. https://opennessinitiative.org/

---

## [Author Response]

Reviewer #1:[…] 1) Negative control. It would be nice if authors can show a case that can be considered reasonably as a negative control. The reason is that all figures show the cases of positive effects consistent with the theory. Evidence would be stronger if there are no effects like flat G(θ) when there is no interaction. That would also help in arguing the functional role of γ synchronization later in Discussion. Figure 3—figure supplement 1 shows a negative dependence of interaction strength on cortical distance, which may suggest what a negative control case could be. However, the lowest values are still above 1 Hz. A more desirable control would be to use elevated γ activity recorded simultaneously in 2 unrelated cortices. One control possibility might be recordings separated by much greater cortical distances. Assuming the authors do not have such cases, recordings during no stimulation or when animals are asleep may work. Even when control conditions are met, interaction strength still might not be zero. It is possible that there may always be some residual amount, as some interactions could remain even for null conditions. However, one might hope to find that interaction strength under control conditions is significantly smaller than values during test conditions. Also, even with a residual interaction strength, preferred phase could deviate from those during test conditions.

The interaction strength (phase-dependent IF modulations) reflects the strength of synchronization (even if rhythms are not phase-locked due to high detuning), a measure predicted within the framework of weakly coupled oscillators. If the measure does truly relate to interaction strength, it should depend on anatomical connectivity strength. We did not directly measure anatomical connectivity, but we used the fact that horizontal connectivity decreases as a function of cortical distance in V1. The observation that measured interaction strength decreased with cortical distance was for us a strong indication that the measure has biological relevance. We point this out more clearly now in the text (as suggested by you and reviewer 3) and also include the relevant plot in the main manuscript (Figure 6):

“The interaction strength ε was found to be inversely correlated with the cortical distance between probes (linear regression, M1: R^2^=0.41, M2: R^2^=0.29, both p <10^-10^, Figure 6), in line with the known decrease of V1 horizontal connectivity with distance (Stettler et al., 2002).”

This biological relevance is strengthened further by two analyses suggested by the review (see below).

We agree with the reviewer that a case of no (mutual) connectivity would be beneficial to check whether some interaction strength is remaining. Unfortunately, we did not have such data. However, we have data that get close to this case (Figure 7—figure supplement 3). The laminar probes not only covered the full range of V1 cortical depth, but also a significant part of V2, situated just beneath V1. The receptive fields (RF) of the ‘deep’ V2 were relatively far away from the V1 RF (~3.5°). We therefore analyzed V1-V2 pairs with separate RF positions, to measure the resultant IF modulations amplitude. We found that V1-V2 pairs had similar IF modulations as observed for V1-V1 pairs. The analysis revealed that locations with a large distance (in terms of RF distance) showed much less phase-dependent interactions getting close to zero (interaction strength ε= 0.3Hz). Whether some modulations remained for locations without any connectivity, can however not be tested with this data.

As an additional control, we did the same analysis with shuffled trials to test whether stimulus-locked dynamics had an influence on the IF modulations results. We found that shuffling trials did strongly reduce systematic IF modulations. We have included a report of these two additional analyses in the main text:

“To test further the idea that the interaction strength ε is a biologically meaningful measure of neural interaction more thoroughly, we repeated the analysis of interaction strength ε over cortical distance between probes with trial-shuffled data. […] Altogether, these analyses support the conclusion that ε is a biologically meaningful measure of neural interaction.”

2) Negative detuning. Readers might surmise that for a give pair of sites, a pair of δ IFs, one positive and one negative values of same magnitudes, can be obtained, like IF1-IF2 or IF2-IF1, so that there could be negative detuning as shown in Figure 5. Also, 2 functions of G (θ) can be obtained, and those can be mirror images of each other across the line ΔIF = 0. Is that correct? It may help the authors' case if an example of negative IF is shown in Supplementary Information. If the answer is yes, then I could understand the left-right symmetry of Arnold tongue (as in Figure 5). However, then why are the cases in Figure 7 not left-right symmetric for physiological data?

For each contact pair we arbitrarily labeled the contact points as IF1 and IF2. However, the assignment for a given pair remained the same across all stimulus contrast conditions, thus inducing a range of frequency differences. A single contact pair therefore had negative as well as positive detuning values. For illustration purposes we showed only positive detuning. We added now a clarifying sentence:

“We show in these examples positive frequency differences for illustration, but negative differences were also present for single contact pairs in our data, depending on the sign of the contrast difference (see Figure 3—figure supplement 1).”

We also added further clarifications in the figure legend of Figure 2:

“Note further that ΔIF variations are the result of IF variations occurring simultaneously at the two contact points that together constitute a contact pair.”

Later on in the same figure legend we added that in panel E:

“the ΔIF values tend to be positive, which is related to the sign of the contrast difference and resulting detuning. If for the same pair the contrast difference had been reversed, ΔIF values would have tended to be negative.”

Panel F was newly added in Figure 3—figure supplement 1, and shows ΔIF modulation curves for different detuning values ranging from negative to positive for both monkeys.

Regarding the symmetry of the Arnold tongue: The Arnold tongue of the phase-oscillator model is symmetric in Figure 5, because the underlying interaction function is symmetric (sinusoid with equal negative and positive parts) and because the values were analytically exactly derived, so there were no numerical estimation errors. In Figure 7 we see slight deviations from symmetry because of noise and limited amount of experimental data. Symmetry fits with the experimental observation that interaction function had a positive and negative component and each member of a pair had IF modulations indicating interactions are mutual.

3) Results section: "strongly reduce the influence of volume conduction by calculating current-source density (CSD)" Though the authors' have innovated extensively, the basic laminar electrode approach to data collection in awake monkeys is not new. There are quite a number of studies by a number of groups that used this method to analyze the areal and cellular generators of specific VEP, AEP and EEG components in monkeys between ~1990 and the present. It would be helpful to the broad readership of eLife if the present study were put in the context of that literature. Additionally, effects of volume conduction have been subject of debate over the last several years, and it would be helpful to readers to point to that. This all highlights the particular strength of the CSD approach relative to more typical LFP recordings. Note also, that there is a microscale phenomenon in volume conduction that might be relevant to this paper (Kajikawa J Neurophysiol. 2015).

We thank the reviewer for this remark and agree fully. We edited the text to more clearly indicate the value of using CSD in the context of existing literature:

“The close positions of recording sites may have led to a contribution of volume condition to synchronization measures. […]The success in reducing volume conduction using CSD favors its use over LFP for spectral analysis at high spatial resolution.”

We did not know the article from Kajikawa and Schroeder and we have read it with a lot of interest. We agree that a deeper understanding of LFP and CSD across layers is an important topic. In our comparison between deep and middle-superficial layers we have used CSD, which should have limited far-reaching cross-laminar contributions present for LFP. We cited the article in this context.

Now, some more specific concernsIntroduction section: It would be helpful if the authors would be more descriptive introducing concepts of detuning and local stimulus contrast.

We improved the text that introduces detuning and local stimulus contrast:

“The local contrast varied periodically over visual space such that different contrasts were presented to different cortical locations. The magnitude of contrast difference (ranging from 0% to ~43%, see Table 1) was manipulated by varying the sign and amplitude of the spatial variation in contrast.”

And:

“Note that here, detuning ∆ω is the intrinsic or natural frequency difference between two oscillators, which is the frequency difference oscillators would have without any interaction. The measured detuning can differ from the intrinsic detuning ∆ω if the oscillators exhibit synchronization.”

Results, first paragraph: gamma power in the deepest layer is remarkable, but is analyzed rather later as an afterthought. This might be brought forward in the analysis and done in parallel with the other layer compartments.

We agree that it is very interesting. It does fit observations from (Xing et al., 2012) and (van Kerkoerle et al., 2014) who found also gamma in the deepest layer (we cited them now in the text). We found however that a detailed laminar analysis would go beyond the scope of this study. The main analysis was focused on L2-4 contacts that represented the largest gamma source (‘compartment’, (Maier et al., 2010)). In Figure 7, we extended our analysis to deep layer gamma (putatively Layer 6) to show that it did not differ qualitatively from middle-superficial layers. We believe adding information about the deep layers in Figure 6, and give more detailed description of analysis in text would overload the paper and not add much conceptually. In addition, a more detailed and careful analysis is needed for understanding possible small differences among layers as well as the gamma-band interactions occurring within a cortical column. We feel this goes beyond the scope of this paper.

Figure 1 Supposedly the shading represents SEM? Please state.

Thanks. It is corrected in legend of Figure 1.

Results, second paragraph: It would help to state in the manuscript how the specific components were chosen. According to the Supplementary material, they chose the components with the largest fraction of spectral power in gamma frequencies (25-60Hz). They also state that in most cases there was one clear component representing gamma-band fluctuations. What was the percentage of cases where there was no clear component? What happens in cases where there was no one clear component?

For all trials we chose the SSD component with strongest power in the gamma range (25-60HZ). It is true that trials with low gamma power (or low SNR) can lead to unclear SSD components that cover the gamma range. We estimated the percentage of trials where non-optimal decomposition occurred by evaluating the number of trials with untypical high frequency variation. We estimated the percentage of outlier SSD trials by counting trials with frequency variation exceeding 1.5 times the interquartile range from either the 25^th^ or 75^th^ percentile of the distribution. The resulting estimates of outliers were for M1: 1.46% and for M2: 1.25% of the trials. We describe this now in subsection “Estimation of instantaneous gamma phase, frequency and amplitude” of the Appendix. We did not eliminate these outliers.

Figure 2: Shouldn't mean IF modulation be called mean ∆IF modulation? Do the number of dots in E reflect time samples across 33 trials presented in D? Number of observations used in this analysis should be clearly stated. Also, the relation should be quantified with statistics. Why is this based on so few trials? I understand that monkeys would usually go through several iterations of an experiment. Does this relation hold through different runs?

Thanks. We corrected ‘mean IF modulation’ to mean *∆*IF modulation in Figure 2

The number of dots in Figure 2 reflects indeed the number of time samples for each of the 33 trials included (1800 x 33 = 59400 points). This now clarified in the legend of Figure 2. We believe we give sufficient statistics in the remainder of the manuscript.

These are trials from one stimulus condition and from one session. We had in total 22 stimulus conditions from which 9 conditions were of relevance for this manuscript. The monkey worked in general for a few hours on this task.

Results section: "The key to understanding how this dynamic relationship leads to synchronization is that phase relationships associated with lower frequency differences are maintained than phase relationships- associated with higher frequency differences" This is unclear to me. Yet, understanding this link is critical. What is meant by "maintained" here?Results, paragraph two, penultimate sentence: typo – "are (better) maintained"

We agree that this sentence was not clear. We changed it to: “are maintained longer over time (slower precession) than”

Results, third paragraph: Re Figure 3/Figure 3—figure supplement 1: Tell us the IF for each of the V1 sites before giving us the ∆IF. Also are effects of ∆IF linear x the band?

We believe that Figure 2 clearly illustrates the process of how in a single trial we first derive gamma band signals for the different V1 sites (e.g. (see Figure 2). We believe it is intuitive that from these two signal an instantaneous phase difference can be computed that varies over time, that in turn can lead to a specific instantaneous frequency difference, that also varies over time (Figure 2). The mean *∆*IF modulation is critical for making the predictions that are then compared to the theoretical framework, and hence the writing, figures and analysis are focused on that. It is will-known that there are IF variations at single recording sites (e.g. Roberts et al., 2013), but these variations are not relevant for our analysis. Nevertheless, examples of single site V1 IF modulations can be found in the Appendix (Figure 3—figure supplement 1).

In the same paragraph: How was the 1Hz modulation amplitude derived?

It is approximately (max-min)/2. We derived it using a Fourier approach (using the second and third Fourier component). We added this information in paragraph three of the Results section, and referred for further explanations to the Appendix (section ‘Estimation of interaction strength ε’).

Re Figure 3: this description is at odds with the Figure It looks more like 2x that (~3.5 Hz); it only becomes obvious later that the description refers to 1/2 amplitude modulation.

We corrected this in the text.

The IF modulation in H looks larger than in E (around 4 vs. 3, respectively). This should be clarified. Also, it would be good to quantify if there is any difference in IF modulation between these two cases.

Notice that the SE shaded area might mislead in coming to a visual judgment. There is actually no difference in IF modulation. In the main text we state for Figure 3 that the amplitude modulation was 1.8Hz, which is 3.6Hz from peak to through. Later on we stated that for Figure 3:

“the magnitude of the instantaneous frequency modulation did not change”. We now have added between brackets explicitly the value of 1.8Hz in the text.

Results: "The chosen examples were representative for the 805 recorded across-probe contact pairs in monkey M1 and 882 pairs in monkey M2." How was this quantified?

We changed the sentence to indicate that the observations, as illustrated in the examples, were characteristic for the whole dataset as shown in the main statistical analysis.

Figure 3 readability of figure would improve if frequency difference (or max and min values used to estimate it) were directly marked on the axes. It would also help to clearly indicate N which was used to obtain these results (same for C, F, I). These results are shown qualitatively. Maybe some quantification using permutations will be beneficial.

Thanks for the suggestion. We tried to directly mark the frequency difference on the axis, but we found this figure then looked too dense. We would therefore rather like to keep it as is.

This figure was devised as an introductory/illustrative figure. Detailed quantifications are presented in Figure 6 and corresponding text.

Results, fourth paragraph: typo "hence speed is analoguesous to"

Thanks for pointing this out. Corrected.

BTW the cyclist analogy is outstanding (and very Dutch)!

Thanks. There might have been indeed an influence by Dutch culture

Subsection “The theory of weakly coupled oscillators (TWCO): A framework for cortical gamma synchronization”: This is a point that occurred at several points throughout the paper. The authors mention horizontal connectivity, but in the communication through synchrony idea (aka CTC of Fries), hierarchical connectivity (e.g., between given RF representations in V1 and V2) is really more relevant lateral connectivity between different eccentricities in V1. Yjis point should be addressed in the Introduction and, while it is mentioned, it should be clarified in the Discussion. Through the authors downplay the notion, the present findings do seem to fit with Prof Singer's Binding Hypothesis.

We agree with the reviewer that it is an important issue that needs more discussion. We are thankful to the reviewer to promote this discussion.

We question whether there is a clear and founded distinction between ‘binding theories’ and CTC, particularly at the level of synchronization principles involved. Even though anatomical connectivity between cortical areas is different from within a cortical area, we think that coupled oscillator concepts (detuning, IF modulations) likely apply also for between-area synchrony. Our ongoing analysis of V1-V2 interactions (based on data in Roberts et al., 2013) supports this view as well as published reports that synchrony between cortical areas is observed despite frequency differences (Bosman et al., 2012).

It was not our intention to downplay the Binding by synchrony (BBS) hypothesis as proposed by Prof. Singer. Our data does support a local type of BBS (however not a global type BBS), where feature integration can be supported by gamma synchrony at the level of surround receptive fields. We added a Line to express that we think that gamma do support feature binding at a local spatial level. It reads now:

“These properties suggest V1 gamma as a functional mechanism for early vision (Eckhorn et al., 2001; Gray and Singer, 1989) by temporally coordinating local neural activity as a function of sensory input and connectivity. […]It is therefore not likely that gamma within V1 ‘binds’ whole perceptual objects, but rather binds features locally at the level of surround receptive fields.”

We now explicitly address the extension of inter-areal interactions in the text:

“Gamma synchronization across cortical areas have been observed in spite of frequency differences (Bosman et al., 2012; Gregoriou et al., 2009), which is further supported by our additional analysis of V1-V2 interactions. Together, this suggests that similar principles likely operate for gamma-band inter-areal interactions.”

In the same section: The formula includes only the detuning (Δω) and not the frequency itself. I understand that the authors would like to concentrate on gamma (and they provide clear motivation for that) but it will be good to discuss how this result might be extended to different frequency bands.

We completely agree. We added a paragraph stating that the analysis approach can be extended to other cortical/subcortical areas and to different frequency-bands. This paragraph reads as follows:

“Importantly, as long as the instantaneous phase of a neural rhythm can be determined, the methods used in this study can be applied. Instantaneous phase extraction has been for example applied to theta rhythms (Belluscio et al., 2012; Buzsáki, 2002) or α rhythms (Lakatos et al., 2012; Samaha and Postle, 2015; Schwabedal et al., 2016).”

Figure 4: Aren't the functions supposed to have the shape of a sine wave? Particularly, Figure 4 does not look like that: downward peak look like they have almost twice the width of upward peaks.

The underlying interaction function was found to be approximatively a sinusoid. In our original Discussion, we used terms such as “sinusoidal-like”. We mention now more explicitly that our analysis might have led to a more smoothed picture of the interaction function than it really is on single trials (due to noise, averaging, CSD-related phase shifts).

At low detuning, when the IF modulations gets close to zero, the IF modulations get deformed and more asymmetric. We observed this in simulated phase-oscillator model (where the PRC is exactly known), in PING network simulations and in V1 data. At higher detuning the IF modulations converge more and more to the true interaction function shape. That is the reason we estimated the interaction function from IF modulations with sufficient large detuning (>4Hz). It was written in the Supplementary Materials, but we clarified it now also in the main text with the following text:

“Note that the function G(*θ*) was estimated from data with absolute detuning of more than 4Hz. This was done based on the observation that interaction functions became deformed when detuning was close to (see supplementary materials for further details). Further it avoided smearing due to phase shifts occurring mainly within ± 4Hz.”

Figure 5 legend: typo "we to we"

Thanks. Corrected

Subsection “TWCO predicts synchronization properties of V1 cortical gamma rhythms”: PLV variations over single contact pairs were substantially captured by the analytical predictions as a function of Δω and ε (model accuracy: M1: R^2^=0.18, n=7245, M2: R^2^= 0.32, n=7938 Figure 6”? It would be helpful to explain why R^2^ 0f 0.18 and 0.32 values are considered substantial.

We agree with the reviewer that the term is vague. However, the values need to be considered in the context that the model predictions are derived from first-principle (without direct fitting) as well as that the single contact PLV estimates are noisy. The much higher explained variance values for population means support the view that the model predictions capture an important part of the underlying biological variance.

To address this, we removed the term and replaced it with a sentence to give context to the statistical values. It reads:

‘We found that the gamma PLV variations over single contact pairs were significantly captured by the analytical predictions as a function of ∆ω and ε (model accuracy: M1: R^2^=0.18, n=7245, M2: R^2^= 0.32, n=7938 Figure 6), which is considerable taking into account that the model predictions were derived out of first principles and that the single contact data used were noisy.’

In the sixth paragraph: Sig s6 is measured data not model result – should be S5

Thanks. Corrected

In the seventh paragraph: Is there any reason for not including noise in the model?

As shown in Figure 5, we did include noise in the model. We think the reviewer is referring to the borders (black lines) in Figure 5, indicating the expected shape of the Arnold tongue (Δω=ε). In the noiseless case, beyond this point oscillators are not completely synchronized anymore and show phase precession. In the presence of noise as is the case in Figure 5, this sharp transition from complete synchronization to incomplete synchronization does not exist per se. However, this point (or border) is still useful to understand where the synchronization region lies and where PLV values sharply decreases:

“Using the estimated parameters, we also predicted the borders of the Arnold tongue analytically (black lines), which captured the outline of the observed Arnold tongue well. Due to intrinsic frequency variability (phase noise) the PLV values were not expected to decrease as sharply as expected from noiseless coupled oscillators (see Figure 5).”

In the same paragraph: "The results show that gamma rhythms with a higher frequency in a pair had the leading phase.” This should be unpacked a little. Please state how leading and lagging phases are assigned.

A leading phase was defined as the rhythm A having a positive phase difference with rhythm B (phase A –phase B). It is equivalent of rhythm A reaching the oscillation maxima or minima slightly early in time.

For PING networks it meant that neurons of rhythm A spiked earlier than rhythm B (for reviewer’s interest, but not mentioned in paper).

The temporal differences are smaller than the time scale of a full cycle justifying the temporal ordering based on phase differences.

This interpretation was further supported by granger analysis (Figure 8).

Phase-slope-index (PSI) analysis (not included in the manuscript) did additionally support our interpretation as did cross-correlation analysis.

We added:

“A positive phase difference (phase X – phase Y) means that contact X leads (precedes in time) contact Y in terms of the phase of its oscillatory activity. Note that the temporal differences were smaller than the time scale of a full cycle, justifying the use of phase differences to indicate temporal ordering.”

In the tenth paragraph of the same section: "in the (CSD-CSD) gamma band (feedforward-feedback) as a function of detuning and interaction” This should be unpacked a lot. How are feedforward and feedback determined here?

The term ‘feedback’ and ‘feedforward’ are indeed difficult terms in this context. We changed them with more descriptive labels (feedforward: X→Y, feedback: X ←Y).

We clarify this in the text and in the main Figure 8.

Figure 7: This Figure is not well motivated, it’s not clear why f-f vs s-f for some and not others. Why are B&C monkeys combined and not in A&B? Does the causality give the same directionality between IF1-IF2 and IF2?

The main finding of the figure is the L2-4 Field-field Arnold tongue mapping, shown in Figure 7, which is a result that combines the essence of the previous data figures. Our data analysis was focused on L2-4 contacts, where the main gamma-band source is localized. As in previous figures we show the data for the monkeys separately.

B is an extension showing that the same results can be found using spike-field locking. Our motivation was to show that the results can be reproduced by using spiking activity. To limit the number of subplots and in light of all the previous evidence in the paper that the monkeys show comparable results, we opted to show plots combined for both monkeys. The same applies for C.

To address the reviewer’s question on granger causality, we took panel Figure 7, and separated it into a figure for the two monkeys. The changes in phase difference with detuning (Figure 7, bottom row) were entirely in line with the changes in directed influence computed with the Granger Causality method (Figure 8).

Figure 7: Analyses of MUA for Figure 7 needs more explanation. Not clear how gamma phase was defined for spike timing. Did the authors derive/filter spike density functions? And with what kernel/binsize? Also, it is better to shown raw and processed MUA, as shown for CSD in Figure 2.

We did our analysis on aggregated smoothed (Gaussian kernel) MUA signals and treated them as continuous signals. That had the advantage to do the exact same analysis pipeline as used for CSD signals. The smoothing kernel is now explicitly mentioned in the main text:

“MUA activity was smoothed with a Gaussian kernel (σ=4ms) and demeaned to obtain a continuous spike density signal that was then analyzed to the same as CSD signals.”

The gamma-component of the derived smoothed MUA signals was weak compared to CSD signals, which also resulted in much lower PLV estimates. We feel that it is not informative to see raw MUA signals, because it is difficult to visually see the gamma cycles among the noise at single trial level.

Figure 7—figure supplement 4: Does "gamma amplitude" mean the absolute amplitude of gamma band after Hilbert, or the amplitude of real signal? Also, are those amplitudes reflecting the sums of gamma in 2 sites? Please clarify.

The gamma amplitude is the absolute value of the Hilbert transform of the two gamma signals in a pair. In the figure we plotted the mean gamma amplitude as a function of phase difference (population-averaged). We clarified this in the figure legend of Figure 7—figure supplement 4 as follows: “CSD gamma amplitude modulation from monkey M1 averaged over the two contacts for a given pair with modulation strength expressed in percent change from the mean (y-axis) and as a function of phase difference (x-axis).”

I may be wrong, but every panel in the figure shows full cycle of amplitude with a peak near zero phase difference, that makes me think it is real amplitude.

On the Y-axes of all the panels, we show gammaγ amplitude modulation (% of the mean) as a function of delta phase. This is shown for M1, M2 and he PING model. The different subplots represent different detuning and interaction strength levels.

Is it worth plotting gamma amplitude against interaction strength, ε?

We believe the relationship between gamma amplitude modulation and interaction strength is well illustrated in Figure 7—figure supplement 3.

In those plots, peak position is slightly rightward. Does it relate to phase leading/lagging?

The amplitude modulations were indeed not centered at zero phase-difference. It is likely to relate to detuning and the established preferred phase-relation.

Similar properties were observed for PING network simulations, which convinced us that these properties probably tell something about how networks interact. However, we believe that this should be investigated by its own systematically in a future study.

Same question again as above: If IF1-IF2 and IF2-IF1 are counted equally, then should the curves be symmetric around Δ-phase = 0?

The detuning values were not absolute (not rectified), so we selected values from one side of the Arnold tongue. Therefore, e.g. for averaging of pairs with detuning of 4Hz, we averaged over conditions where for IF1-IF2 or IF2-IF1 the second contact had frequency 4Hz higher than the first one. That is the reason the curves do not need to be symmetric

Supplementary Information – Correction for CSD-induced phase shifts: "we normalized the phase-differences for each given contact pair to the condition having the smallest frequency difference," by doing what? Is this parallel translations of points along axes?

Yes, it is a parallel translation. We subtract the phase difference from the condition of smallest frequency difference with the phase differences from all other conditions. It shifts all points along the axis and does not change any other distribution properties. This was clarified in the Appendix, as follows:

“To reduce the effect of the phase shifts, we normalized the phase-differences for each given contact pair to the condition having the smallest frequency difference (this corresponds to a parallel translation).”

In the final paragraph of the subsection: "in the case of mutually anatomically coupled cortical locations, detuning strongly influences the main direction of information flow" As presented, this is questionable.

We agree with the reviewer that the sentence should be written more carefully. We have done so and now provide also a bit more context, which now read as follows:

“in the case of mutually anatomically coupled cortical locations, detuning influences the temporal relationship and possibly the direction of information flow between synchronized gamma rhythmic neural assemblies”.

We argue that for mutually coupled cortical location, like between V1 locations, the phase-lag is dependent on detuning. Further, as shown for the granger analysis, but also for network simulations published by us and others (Besserve et al., 2015; Buehlmann and Deco, 2010; Cannon et al., 2014; Lowet et al., 2016a), it will shape the information flow between the networks as measured by information theoretical tools (e.g. transfer entropy). But we agree that much more work is needed and these results are only indicative.

While grand G(θ) was used to predict PLV, preferred phase difference could differ among pairs according to Figure 6. Were G(θ) values shifted along the axis of phase difference to align the downward peak of the curve to the preferred phase difference for each pair?

To average the grand G we selected conditions with sufficient detuning such that they had similar phase differences (was written in the Supplementary Materials) to avoid smearing. It is now mentioned in the text:

“Note that the function G(*θ*) was estimated from data with absolute detuning of more than 4Hz. This was done based on the observation that interaction functions became deformed when detuning was close to (see supplementary materials for further details). Further it avoided smearing due to phase shifts occurring mainly within ± 4Hz.”

Subsection “The Arnold tongue and the regulative parameters of gamma synchronization”. Can authors show the range of detuning? Were there cases that can show large deviations in detuning that break down interactions?

The range of detuning values can be seen from Figure 6 and Figure 7.

In Figure 7 conditions of high detuning (Δω=+-6Hz) and low interaction strength (ε=1Hz) led to PLV values close to zero (strong de-synchronization).

We added two descriptive sentences:

“As predicted, conditions of high interaction strength and low detuning showed strong gamma synchronization, whereas conditions of low interaction strength and high detuning yielded weak gamma synchronization.”

In the same subsection: while larger detuning values would decrease opportunities for synchronization across distance in V1, when they are tied strongly to stimulus information in a cell's RF, e.g., the Gabor stimuli used in Ray and Maunsell, there are still opportunities for the neurons in V1 (seeing low contrast) to synchronize with ones in V2 with overlapping RF (also seeing low contrast) and the same for V1/V2 interactions across the different contrast regions of the Gabor. This means basically frequency segregation of channels seeing different "pixels."

This is a statement we agree on. In fact, we are working on a modeling paper that deals with implications for V1 and V2 interactions and the shape of V2 RF.

In our Plos comp bio paper (Lowet et al., 2015), we argue that detuning can help in local feature integration/ figure-ground segregation. Image areas of low heterogeneity ‘bind’ because of low detuning and image regions of high heterogeneity decouple because of high detuning (e.g. around object borders). This we describe in, as quoted here:

“In accordance with these observations, the revealed Arnold tongue of V1 gamma implies that natural image parts with high input/detuning variability (heterogeneity) will induce no or weak synchronization, whereas parts with low input/detuning variability (homogeneity) will induce stronger synchronization.”

We believe that the self-organization of gamma rhythmic activity will have implication for how V2 neurons integrate V1 information and also the relation between V2 neural populations: “Gamma synchronization across cortical areas have been observed in spite of frequency differences (Bosman et al., 2012; Gregoriou et al., 2009), which is further supported by our additional analysis of V1-V2 interactions. Together, this suggests that similar principles likely operate for gamma-band inter-areal interactions.”

Future studies will need to investigate better how V1-V2 gamma-band interactions operate (in both directions) and the implications for visual computation.

Supplementary information: Equation 18 in Supplementary Information duplicates Equation 17. I guess the label "18" was meant for the line: "V(θ)"?

Thanks for pointing it out.

Reviewer #2:[…] 1) The authors seem to confuse the PRC and the coupling function throughout the paper. They are not the same thing. In the theory of weak coupling, there are three things:1) the PRC or adjoint function, Z(t) which is the infinitesimal PRC and is hard to compute with real neurons, but east to compute with models (see for example the tutorial: http://www.math.pitt.edu/~bard/bardware/meth3/node15.html#SECTION00052000000000000000on weak coupling of a simple voltage gated model;

We thank the reviewer for providing the link.

2) The interaction function, H(φ) which is defined as:H(θ_2_-θ_1_) = (1/T) ∫0TZ(s+θ1). C_12_(u(s+θ_1_),u(s+θ_2_)) dsThat is, it is the convolution of the PRC (Z) with the coupling between the oscillators, here denoted by u(t) with 1 = postsynaptic and 2 = presynaptic3) The odd part of the interaction function which in this paper is G(θ) arises as followsθ_1_' = ω_1_ + ε_12_ H(θ_2_ – θ_1_) Equation 1θ_2_' = ω_2_ + ε_21_ H(θ_1_ – θ_2_)If you let φ = θ_2_ – θ_1_, the phase difference and if ε_12_=ε_21_, thenφ' = Δω + εG(φ) Equation 2where G(φ) = H(-φ) – H(φ) is proportional to the odd part of H(φ)Equation 2 is what the authors measure; it is not the PRC. I just want to make that clear.

We agree and thank the reviewer for his comments. We indeed did not take into account a coupling function in the equations that capture synaptic dynamics. This implies that we cannot be sure to have measured the underlying PRC.

We corrected this and clarified the relationship between the measured interaction function G to the PRC and the coupling function. Given that the synaptic dynamics that underlie gamma rhythms are relatively fast, we think that the (odd-part) PRC will be likely similar to the measured interaction function G. We clarified this in our text:

“It is worth noting that the interaction function G(*θ*) is not identical with the PRC. […]Hence, whenever we use the terms PRC and G(*θ*) in the context of our empirical data analysis, we keep their conceptual distinction in mind while considering them similar for practical purposes.”

Since they measure φ, they can now compute the actual interaction function at both sites and this is even more informative than is G. By measuring θ_1_' and θ_2_', and plotting this against φ, equation 1 shows thatθ_1_' = ω_1_ + ε_12_ H(φ)θ_2_' = ω_2_ + ε_21_ H(-φ)thus, in addition to G, they will be able to measure ω_1_ + ε_12_ H(φ) and ε_21_ H(-φ). This will give them more information about the direction of coupling and also will give them the full value of H. Indeed, there is nothing here that requires the two H's to even be the same. Thus, I would suggest plotting θ_1_', θ_2_' vs. φ as well as φ' vs. φ.

We thank the author for pointing to this important issue. In this manuscript we have focused on the bidirectional interaction function G, being the central function of the equation used, for predicting experimental synchronization data.

Prior to submission, we have tried to obtain (trustable) unidirectional interaction functions H for experimental data. However, the difficulty is to estimate the intrinsic frequency *ω_1_* or *ω_2_* which are hidden variables. Without a good estimate of the intrinsic frequencies, it cannot be determined which part of the single H is negative or positive. Future work is required to resolve these issues.

As we show in supplementary materials we can however obtain an estimate of the general shape of single H and their amplitudes for V1 data (see examples in Figure 3—figure supplement 1)

We estimated the single H of PING networks, which was highly asymmetric (mainly positive). This was possible because for PING networks the intrinsic frequencies *ω_1_* and *ω_2_* can be obtained by simply decoupling the networks.

The detuning variable is also a hidden variable. Taking the assumption that both rhythms have similar interaction strengths, we found that the detuning is the mean of the measured interaction function. This is also true even if the underlying H is asymmetric. We tested the validity of this approach using phase-oscillator simulations as well as PING simulations where the true detuning value was known.

3) G(φ) is plotted looks very sinusoidal. Is this a consequence of the way the phase is extracted? I suggest taking a model pair of oscillators, such as Hodgkin huxley and coupling them with synapse along with noise and heterogeneity in coupling and in frequency. Then use the method that is used for the data to extract G(φ) and H(φ) and then compare this to the actual determistic values found by setting the noise to zero and computing H, G as in the above. This would assure me that the method of using Hilbert transforms isn’t somehow removing all the Fourier modes beyond the lowest.

To show that the obtained G is not trivial, we simulated phase oscillators with different types of true G. As we show in Figure 5—figure supplement 1 the resultant G reflected well the true G for different SNR.

We need however to state that the noise in our measurements might have smoothed our interaction function estimates. Also that we applied CSD might have led to further variability and averaging out of more transient components. We state this problem more explicitly now:

“The interaction functions we estimated here might be smoother than they really are due to limitations of our analysis arising from noise, averaging, and steps taken to reduce volume conduction. Future studies are required to characterize in more detail the (unidirectional/bi-directional) γ-band interaction functions.”

4). Since the frequency varies rapidly between the two sites, does this variability covary between the two populations? I ask this because one can take two uncoupled oscillators and apply a broadband correlated signal to them and they will produce a very nice peak in their phase-difference histogram. For example, see Nakao, Arai, Kuramoto PRE 2005, or other papers by Arai & Nakao, or for a neuroscience version, Zhou, Burton, et al. 2013 Frontiers in Comp Neuro.

It concerns the question whether synchronization is due to direct mutual connections between local cortical locations or due to coordination by common input modulation.

Testing the idea directly would require measuring the feedforward synaptic inputs originating from the retina-geniculate pathway and see whether input correlation strength among V1 locations does influence gamma-band synchronization. This unfortunately goes beyond the scope of this study.

However, we think that shared broadband input as explanation for synchronization patterns found here is unlikely for the following reasons:

Empirically (Bosman et al., 2009; Brunet et al., 2013; Lowet et al., 2016b; Ray and Maunsell, 2010; Roberts et al., 2013) gamma frequency fluctuations in V1 and between V1 and V2 have been shown to have positive correlation, however with a strong dominance in the lower frequency bands (delta-theta 1-5Hz band). Low frequency modulations appear unlikely to be good candidate for phase synchronization of gamma rhythms.

A plausible scenario is that two cortical locations share directly some common input. This would imply that neurons from the locations have partially overlapping receptive fields (RFs). However, cortical V1 locations with clearly distinct RF positions still showed substantial phase coordination.

Previous studies have suggested that gamma-band synchronization in V1 has a spatial scale that fits well the known spatial spread of horizontal connectivity (Eckhorn et al., 2001; Palanca and DeAngelis, 2005). This was confirmed by our own data. This fits also well with observations that gamma-band properties depends on surround receptive field activation (Gieselmann and Thiele, 2008). Further, recent work in mice have shown that V1 gamma-band synchronization depends on particular interneurons (SOM+) that are important for horizontal cortical interactions (Veit et al., 2017).

Furthermore, frequency correlation per se can be cause as well as consequence of synchronization. For example, PING networks with mutual interactions with unshared input variability will exhibit increased frequency correlation with stronger connectivity. That is the reason is very important to measure the correlation at the input level to properly test the idea.

We wrote on this important topic in the Discussion:

“In addition, we assumed that synchronization between V1 locations emerged due to mutual horizontal interactions, yet common input fluctuations might further shape V1 gamma synchronization (Wang et al., 2000; Wiesenfeld and Moss, 1995; Zhou et al., 2013). Although we did not investigate the possible effects of common input, the observation that gamma synchronization occurred between V1 locations with distinct receptive fields and with a dependence on cortical distance as expected from anatomical connectivity (Gail et al., 2000; Gieselmann and Thiele, 2008; Palanca and DeAngelis, 2005; Ray and Maunsell, 2010; Stettler et al., 2002) indicates that cross-columnar gamma-band synchronization depends strongly on direct mutual horizontal interactions (Veit et al., 2017).”

Reviewer #3:[…] The results are impressive, but I would like to see a control analysis where the full analysis pipeline is applied to non-simultaneously recorded (and so un-synchronised) trials (either by shuffling trials or combining non-synchronously recorded probes). This would ensure that the SSD step is not picking up some common effects (e.g. stimulus onset) that is inducing the delta-phase-delta-if relationship e.g. in Figure 2. (I don't think full permutation inference is necessary for all results, but just some confirmation that indeed the actual experimental data pipeline does produce flat delta-phase / delta-if curves)

First, we apologize that we forgot to mention the exact time window used for our analysis. Our trials consisted of 1sec baseline fixation and 2sec visual stimulation. For our gamma-band analysis we focused on the stimulation period 0.2sec-2sec. This was fixed in the text, which read as follows:

“For analysis the stimulation period (0.2-2sec) was used, not including the first 200ms to avoid stimulus-evoke transients.”

The authors propose that the method of fitting TWCO model to data and associated analyses such as Arnold tongue plots might be applicable to other brain regions or frequency band relationships. It would be nice if they could give some heuristic guidance for experimental designs to which these techniques can be applied. Presumably they need to systematically modulate both connection strength and frequency difference, as done here?

Thanks for this suggestion. We wrote a paragraph in the Discussion section about how detuning and interaction strength could be manipulated in future studies:

“Nevertheless, future studies are required to test to what extent weakly coupled oscillator principles apply to different frequency bands across brain regions. […] Aside of emerging new technological possibilities for network state modulation, a tight combination of experimental and dynamic systems theory will be critical for fruitful analysis and interpretation of neural oscillatory data.”

The statistical analysis and Figure 6 show the overall average over all electrode pairs and stimulus conditions. Presumably the experimental contrast already induces a range of PLVs for each specific recording pair, so perhaps they could be evaluated statistically as well (e.g. scatter with pair PLV R2 as a function of distance between the pairs, or something along similar lines).

We computed for each single contact pair separately how much variability over the 9 contrast conditions could be explained by the model predictions. The variation for each single contact pair represent variation by contrast conditions /detuning. The results are PLV R^2^ of M1: 0.27 ± 0.0002, n=802 and M2: 0.1 ± 0.0001, n=882. For phase differences we found that R^2^ of M1: 0.52 ± 0.0001, n=802 and M2: 0.44 ± 0.0004, n=882. The R2 values can differ from the main analysis, because the variability over single contacts (representing differences in cortical distance/interaction strength) is missing. Notice also that the values need to be interpreted in the context of noisy single contact pair estimation and that the model was not directly fitted to the data. We included the analysis in the text for PLV and for mean phase difference

The authors indicate that data and code are available on request. I would strongly urge them then to deposit the data in a suitable repository such as Dryad to reduce their future administrative burden and to ensure long term stable access.https://opennessinitiative.org/

We agree with the reviewers that data sharing is an important issue in neuroscience and beyond. We will upload (after the acceptance of the manuscript, on dryad as suggested) the single-trial CSD data of each monkey allowing the reader to assess the reported observations and to test new ones. This way the final version of the manuscript will contain a link to the project files. Full access to the (raw) data is given upon reasonable request, given the size and complexity of the data. To enhance other researchers’ understanding of the observed phenomena and TWCO, we also added Matlab codes for phase-oscillator simulations and PING network simulations. Further, we will add codes for Singular spectrum decomposition (SSD).